# Nonlinear Reconstruction for Operator Learning of PDEs with Discontinuities

**Samuel Lanthaler**
Computing and Mathematical Science
California Institute of Technology
Pasadena, CA, USA
`slanth@caltech.edu`

**Roberto Molinaro, Patrik Hadorn & Siddhartha Mishra**
Seminar for Applied Mathematics
ETH Zurich
Zurich, Switzerland
`{roberto.molinaro,siddhartha.mishra}@ethz.ch`

## Abstract

A large class of hyperbolic and advection-dominated PDEs can have solutions with discontinuities. This paper investigates, both theoretically and empirically, the operator learning of PDEs with discontinuous solutions. We rigorously prove, in terms of lower approximation bounds, that methods which entail a linear reconstruction step (e.g. DeepONet or PCA-Net) fail to efficiently approximate the solution operator of such PDEs. In contrast, we show that certain methods employing a nonlinear reconstruction mechanism can overcome these fundamental lower bounds and approximate the underlying operator efficiently. The latter class includes Fourier Neural Operators and a novel extension of DeepONet termed shift-DeepONet. Our theoretical findings are confirmed by empirical results for advection equation, inviscid Burgers' equation and compressible Euler equations of aerodynamics.

## 1 Introduction

Many interesting phenomena in physics and engineering are described by partial differential equations (PDEs) with *discontinuous* solutions. The most common types of such PDEs are nonlinear hyperbolic systems of conservation laws (Dafermos, 2005), such as the Euler equations of aerodynamics, the shallow-water equations of oceanography and MHD equations of plasma physics. It is well-known that solutions of these PDEs develop finite-time discontinuities such as *shock waves*, even when the initial and boundary data are smooth. Other examples include the propagation of waves with jumps in linear transport and wave equations, crack and fracture propagation in materials (Sun & Jin, 2012), moving interfaces in multiphase flows (Drew & Passman, 1998) and motion of very sharp gradients as propagating fronts and traveling wave solutions for reaction-diffusion equations (Smoller, 2012). Approximating such (propagating) discontinuities in PDEs is considered to be extremely challenging for traditional numerical methods (Hesthaven, 2018) as resolving them could require very small grid sizes. Although bespoke numerical methods such as high-resolution finite-volume methods, discontinuous Galerkin finite-element and spectral viscosity methods (Hesthaven, 2018) have successfully been used in this context, their very high computational cost prohibits their extensive use, particularly for *many-query* problems such as UQ, optimal control and (Bayesian) inverse problems (Lye et al., 2020), necessitating the design of fast machine learning-based surrogates.

As the task at hand in this context is to learn the underlying solution *operator* that maps input functions (initial and boundary data) to output functions (solution at a given time), recently developed *operator learning* methods can be employed in this infinite-dimensional setting (Higgins, 2021). These methods include *operator networks* (Chen & Chen, 1995) and their deep version, *DeepONet* (Lu et al., 2019; 2021), where two sets of neural networks (branch and trunk nets) are combined in a

*linear reconstruction procedure* to obtain an infinite-dimensional output. DeepONets have been very successfully used for different PDEs (Lu et al., 2021; Mao et al., 2020b; Cai et al., 2021; Lin et al., 2021). An alternative framework is provided by *neural operators* (Kovachki et al., 2021a), wherein the affine functions within DNN hidden layers are generalized to infinite-dimensions by replacing them with kernel integral operators as in (Li et al., 2020a; Kovachki et al., 2021a; Li et al., 2020b). A computationally efficient form of neural operators is the Fourier Neural Operator (FNO) (Li et al., 2021a), where a translation invariant kernel is evaluated in Fourier space, leading to many successful applications for PDEs (Li et al., 2021a;b; Pathak et al., 2022).

Currently available theoretical results for operator learning (e.g. Lanthaler et al. (2022); Kovachki et al. (2021a;b); De Ryck & Mishra (2022b); Deng et al. (2022)) leverage the *regularity* (or smoothness) of solutions of the PDE to prove that frameworks such as DeepONet, FNO and their variants approximate the underlying operator efficiently. Although such regularity holds for many elliptic and parabolic PDEs, it is obviously destroyed when discontinuities appear in the solutions of the PDEs such as in the hyperbolic PDEs mentioned above. Thus, a priori, it is unclear if existing operator learning frameworks can efficiently approximate PDEs with discontinuous solutions. This explains the paucity of theoretical and (to a lesser extent) empirical work on operator learning of PDEs with discontinuous solutions and provides the rationale for the current paper where,

- using a lower bound, we rigorously prove approximation error estimates to show that operator learning architectures such as DeepONet (Lu et al., 2021) and PCA-Net (Bhattacharya et al., 2021), which entail a *linear reconstruction* step, *fail* to efficiently approximate solution operators of prototypical PDEs with discontinuities. In particular, the approximation error only decays, at best, linearly in network size.

- We rigorously prove that using a *nonlinear reconstruction* procedure within an operator learning architecture can lead to the efficient approximation of prototypical PDEs with discontinuities. In particular, the approximation error can decay *exponentially in network size*, even after discontinuity formation. This result is shown for two types of architectures with *nonlinear reconstruction*, namely the widely used Fourier Neural Operator (FNO) of (Li et al., 2021a) and for a novel variant of DeepONet that we term as *shift-DeepONet*.

- We supplement the theoretical results with extensive experiments where FNO and shift-DeepONet are shown to consistently outperform DeepONet and other baselines for PDEs with discontinuous solutions such as linear advection, inviscid Burgers' equation, and both the one- and two-dimensional versions of the compressible Euler equations of gas dynamics.

## 2 METHODS

**Setting.** Given compact domains $D \subset \mathbb{R}^d$, $U \subset \mathbb{R}^{d'}$, we consider the approximation of operators $\mathcal{G} : \mathcal{X} \to \mathcal{Y}$, where $\mathcal{X} \subset L^2(D)$ and $\mathcal{Y} \subset L^2(U)$ are the input and output function spaces. In the following, we will focus on the case, where $\bar{u} \mapsto \mathcal{G}(\bar{u})$ maps initial data $\bar{u}$ to the solution at some time $t > 0$, of an underlying time-dependent PDE. We assume the input $\bar{u}$ to be sampled from a probability measure $\mu \in \mathrm{Prob}(\mathcal{X})$.

**DeepONet.** DeepONet (Lu et al., 2021) will be our prototype for operator learning frameworks with linear reconstruction. To define them, let $\boldsymbol{x} := (x_1, \ldots, x_m) \in D$ be a fixed set of *sensor points*. Given an input function $\bar{u} \in \mathcal{X}$, we encode it by the point values $\mathcal{E}(\bar{u}) = (\bar{u}(x_1), \ldots, \bar{u}(x_m)) \in \mathbb{R}^m$. DeepONet is formulated in terms of two neural networks: The first is the **branch-net $\boldsymbol{\beta}$**, which maps the point values $\mathcal{E}(\bar{u})$ to coefficients $\boldsymbol{\beta}(\mathcal{E}(\bar{u})) = (\beta_1(\mathcal{E}(\bar{u})), \ldots, \beta_p(\mathcal{E}(\bar{u})))$, resulting in a mapping

$$\boldsymbol{\beta} : \mathbb{R}^m \to \mathbb{R}^p, \quad \mathcal{E}(\bar{u}) \mapsto (\beta_1(\mathcal{E}(\bar{u})), \ldots, \beta_p(\mathcal{E}(\bar{u}))). \tag{2.1}$$

The second neural network is the so-called **trunk-net $\boldsymbol{\tau}(y) = (\tau_1(y), \ldots, \tau_p(y))$**, which is used to define a mapping

$$\boldsymbol{\tau} : U \to \mathbb{R}^p, \quad y \mapsto (\tau_1(y), \ldots, \tau_p(y)). \tag{2.2}$$

While the branch net provides the coefficients, the trunk net provides the "basis" functions in an expansion of the output function of the form

$$\mathcal{N}^{\mathrm{DON}}(\bar{u})(y) = \sum_{k=1}^{p} \beta_k(\bar{u}) \tau_k(y), \quad \bar{u} \in \mathcal{X}, \ y \in U, \tag{2.3}$$

with $\beta_k(\bar{u}) = \beta_k(\mathcal{E}(\bar{u}))$. The resulting mapping $\mathcal{N}^{\mathrm{DON}} : \mathcal{X} \to \mathcal{Y}$, $\bar{u} \mapsto \mathcal{N}^{\mathrm{DON}}(\bar{u})$ is a **DeepONet**.

Although DeepONet were shown to be universal in the class of measurable operators (Lanthaler et al., 2022), the following *fundamental lower bound* on the approximation error was also established,

***Proposition* 2.1** (Lanthaler et al. (2022, Thm. 3.4))**.** Let $\mathcal{X}$ be a separable Banach space, $\mathcal{Y}$ a separable Hilbert space, and let $\mu$ be a probability measure on $\mathcal{X}$. Let $\mathcal{G} : \mathcal{X} \to \mathcal{Y}$ be a Borel measurable operator with $\mathbb{E}_{\bar{u}\sim\mu}[\|\mathcal{G}(\bar{u})\|_{\mathcal{Y}}^2] < \infty$. Then the following lower approximation bound holds for any DeepONet $\mathcal{N}^{\mathrm{DON}}$ with trunk-/branch-net dimension $p$:

$$\mathscr{E}(\mathcal{N}^{\mathrm{DON}}) = \mathbb{E}_{\bar{u}\sim\mu}\left[\|\mathcal{N}^{\mathrm{DON}}(\bar{u}) - \mathcal{G}(\bar{u})\|_{\mathcal{Y}}^2\right]^{1/2} \geq \mathscr{E}_{\mathrm{opt}} := \sqrt{\sum_{j>p}\lambda_j}, \qquad (2.4)$$

where the optimal error $\mathscr{E}_{\mathrm{opt}}$ is written in terms of the eigenvalues $\lambda_1 \geq \lambda_2 \geq \ldots$ of the covariance operator $\Gamma_{\mathcal{G}_\#\mu} := \mathbb{E}_{u\sim\mathcal{G}_\#\mu}[(u \otimes u)]$ of the push-forward measure $\mathcal{G}_\#\mu$.

We refer to **SM** A for relevant background on the underlying principal component analysis (PCA) and covariance operators, as well as an example illustrating the connection between sharpness of gradients and the decay of the PCA eigenvalues $\lambda_j$ (**SM** A.1). The same lower bound (2.4) in fact holds for any operator approximation of the form $\mathcal{N}(\bar{u}) = \sum_{k=1}^p \beta_k(\bar{u})\tau_k$, where $\beta_k : \mathcal{X} \to \mathbb{R}$ are arbitrary functionals. In particular, this bound continues to hold for e.g. the PCA-Net architecture of Hesthaven & Ubbiali (2018); Bhattacharya et al. (2021). We will refer to any operator learning architecture of this form as a method with "linear reconstruction", since the output function $\mathcal{N}(\bar{u})$ is restricted to the linear $p$-dimensional space spanned by the $\tau_1, \ldots, \tau_p \in \mathcal{Y}$. In particular, *DeepONet are based on linear reconstruction*. To overcome the lower bound (2.4) which sets fundamental limitations on DeepONets, the basis $\tau$ therefore needs to additionally depend on the input $u$.

**shift-DeepONet.** The lower bound (2.4) shows that there are fundamental barriers to the expressive power of operator learning methods based on linear reconstruction. This is of particular relevance for problems in which the optimal lower bound $\mathscr{E}_{\mathrm{opt}}$ in (2.4) exhibits a *slow decay* in terms of the number of basis functions $p$, due to the slow decay of the eigenvalues $\lambda_j$ of the covariance operator. It is well-known that even linear advection- or transport-dominated problems can suffer from such a slow decay of the eigenvalues (Ohlberger & Rave, 2013; Dahmen et al., 2014; Taddei et al., 2015; Peherstorfer, 2020), which could hinder the application of linear-reconstruction based operator learning methods to this very important class of problems. In view of these observations, it is thus desirable to develop a *nonlinear* variant of DeepONet which can overcome such a lower bound in the context of transport-dominated problems. We propose such an extension below.

A **shift-DeepONet** $\mathcal{N}^{\mathrm{sDON}} : \mathcal{X} \to \mathcal{Y}$ is an operator of the form

$$\mathcal{N}^{\mathrm{sDON}}(\bar{u})(y) = \sum_{k=1}^p \beta_k(\bar{u})\tau_k\left(\mathcal{A}_k(\bar{u})y + \gamma_k(\bar{u})\right), \qquad (2.5)$$

where the input function $\bar{u}$ is encoded by evaluation at the sensor points $\mathcal{E}(\bar{u}) \in \mathbb{R}^m$. We retain the DeepONet **branch-** and **trunk-nets** $\boldsymbol{\beta}$, $\boldsymbol{\tau}$ defined in (2.1), (2.2), respectively, and we have introduced a **scale-net** $\mathcal{A} = (\mathcal{A}_k)_{k=1}^p$, consisting of matrix-valued functions

$$\mathcal{A}_k : \mathbb{R}^m \to \mathbb{R}^{d'\times d'}, \quad \mathcal{E}(\bar{u}) \mapsto \mathcal{A}_k(\bar{u}) := \mathcal{A}_k(\mathcal{E}(\bar{u})),$$

and a **shift-net** $\boldsymbol{\gamma} = (\gamma_k)_{k=1}^p$, with

$$\gamma_k : \mathbb{R}^m \to \mathbb{R}^{d'}, \quad \mathcal{E}(\bar{u}) \mapsto \gamma_k(\bar{u}) := \gamma_k(\mathcal{E}(\bar{u})),$$

All components of a shift-DeepONet are represented by deep neural networks, potentially with different activation functions.

***Remark* 2.2.** The form of shift-DeepONet (2.5) is very natural from a theoretical perspective. Practical experimentation indicates that an extended architecture based on a trunk-net $\boldsymbol{\tau}^* : \mathbb{R}^{d'\times p} \to \mathbb{R}^p$, *depending jointly on all values* $\mathcal{A}_1(\bar{u})y + \gamma_1(\bar{u}), \ldots, \mathcal{A}_p(\bar{u})y + \gamma_p(\bar{u})$ and defining a mapping

$$\mathcal{N}^{\mathrm{sDON}^*}(\bar{u})(y) := \sum_{k=1}^p \beta_k(\bar{u})\tau_k^*\left(\mathcal{A}(\bar{u})y + \gamma(\bar{u})\right), \qquad (2.6)$$

with concatenated input $\mathcal{A}(\bar{u})y + \gamma(\bar{u}) := \big(\mathcal{A}_1(\bar{u})y + \gamma_1(\bar{u}), \ldots, \mathcal{A}_p(\bar{u})y + \gamma_p(\bar{u})\big)$ achieves better accuracy. Our numerical results will be reported for (2.6). We emphasize that all theoretical results in this work apply to both architectures, (2.5) and (2.6).

Since shift-DeepONets reduce to DeepONets for the particular choice $\boldsymbol{A} \equiv \boldsymbol{1}$ and $\boldsymbol{\gamma} \equiv 0$, the universality of DeepONets (Theorem 3.1 of Lanthaler et al. (2022)) is clearly inherited by shift-DeepONets. However, as shift-DeepONets do not use a linear reconstruction (the trunk nets in (2.5) depend on the input through the scale and shift nets), the lower bound (2.4) does not directly apply, providing possible space for shift-DeepONet to efficiently approximate transport-dominated problems, especially in the presence of discontinuities.

**Fourier neural operators (FNO).** An FNO $\mathcal{N}^{\mathrm{FNO}}$ (Li et al., 2021a) is a composition

$$\mathcal{N}^{\mathrm{FNO}} : \mathcal{X} \mapsto \mathcal{Y} : \quad \mathcal{N}^{\mathrm{FNO}} = Q \circ \mathcal{L}_L \circ \cdots \circ \mathcal{L}_1 \circ R, \tag{2.7}$$

consisting of a "lifting operator" $\bar{u}(x) \mapsto R(\bar{u}(x), x)$, where $R$ is represented by a (shallow) neural network $R : \mathbb{R}^{d_u} \times \mathbb{R}^d \to \mathbb{R}^{d_v}$ with $d_u$ the number of components of the input function, $d$ the dimension of the domain and $d_v$ the "lifting dimension" (a hyperparameter), followed by $L$ hidden layers $\mathcal{L}_\ell : v^\ell(x) \mapsto v^{\ell+1}(x)$ of the form

$$v^{\ell+1}(x) = \sigma \left( W_\ell \cdot v^\ell(x) + b_\ell(x) + \left( K_\ell v^\ell \right)(x) \right),$$

with $W_\ell \in \mathbb{R}^{d_v \times d_v}$ a weight matrix (residual connection), $x \mapsto b_\ell(x) \in \mathbb{R}^{d_v}$ a bias function and with a convolution operator $K_\ell v^\ell(x) = \int_{\mathbb{T}^d} \kappa_\ell(x - y)v^\ell(y)\,dy$, expressed in terms of a (learnable) integral kernel $x \mapsto \kappa_\ell(x) \in \mathbb{R}^{d_v \times d_v}$. The output function is finally obtained by a linear projection layer $v^{L+1}(x) \mapsto \mathcal{N}^{\mathrm{FNO}}(\bar{u})(x) = Q \cdot v^{L+1}(x)$.

The convolution operators $K_\ell$ add the indispensable *non-local* dependence of the output on the input function. Given values on an equidistant Cartesian grid, the evaluation of $K_\ell v^\ell$ can be efficiently carried out in Fourier space based on the discrete Fourier transform (DFT), leading to a representation

$$K_\ell v^\ell = \mathcal{F}_N^{-1} \left( P_\ell(k) \cdot \mathcal{F}_N v^\ell(k) \right),$$

where $\mathcal{F}_N v^\ell(k)$ denotes the Fourier coefficients of the DFT of $v^\ell(x)$, computed based on the given $N$ grid values in each direction, $P_\ell(k) \in \mathbb{C}^{d_v \times d_v}$ is a complex Fourier multiplication matrix indexed by $k \in \mathbb{Z}^d$, and $\mathcal{F}_N^{-1}$ denotes the inverse DFT. In practice, only a finite number of Fourier modes can be computed, and hence we introduce a hyperparameter $k_{\max} \in \mathbb{N}$, such that the Fourier coefficients of $b_\ell(x)$ as well as the Fourier multipliers, $\widehat{b}_\ell(k) \equiv 0$ and $P_\ell(k) \equiv 0$, vanish whenever $|k|_\infty > k_{\max}$. In particular, with fixed $k_{\max}$ the DFT and its inverse can be efficiently computed in $O(((2k_{\max} + 1)N)^d)$ operations (i.e. linear in the total number of grid points). The output space of FNO (2.7) is manifestly nonlinear as it is not spanned by a fixed number of basis functions. Hence, FNO constitute a *nonlinear reconstruction* method.

## 3 THEORETICAL RESULTS.

**Context.** Our aim in this section is to rigorously prove that the nonlinear reconstruction methods (shift-DeepONet, FNO) efficiently approximate operators stemming from discontinuous solutions of PDEs whereas linear reconstruction methods (DeepONet, PCA-Net) fail to do so. To this end, we follow standard practice in numerical analysis of PDEs (Hesthaven, 2018) and choose two prototypical PDEs that are widely used to analyze numerical methods for transport-dominated PDEs. These are the linear transport or advection equation and the nonlinear inviscid Burgers' equation, which is the prototypical example for hyperbolic conservation laws. The exact operators and the corresponding approximation results with both linear and nonlinear reconstruction methods are described below. The computational complexity of the models is expressed in terms of hyperparameters such as the model size, which are described in detail in **SM** B.

**Linear Advection Equation.** We consider the one-dimensional linear advection equation

$$\partial_t u + a \partial_x u = 0, \quad u(\cdot, t = 0) = \bar{u} \tag{3.1}$$

on a $2\pi$-periodic domain $D = \mathbb{T}$, with constant speed $a \in \mathbb{R}$. The underlying *operator* is $\mathcal{G}_{\mathrm{adv}} : L^1(\mathbb{T}) \cap L^\infty(\mathbb{T}) \to L^1(\mathbb{T}) \cap L^\infty(\mathbb{T})$, $\bar{u} \mapsto \mathcal{G}_{\mathrm{adv}}(\bar{u}) := u(\cdot, T)$, obtained by solving the PDE (3.1) with initial data $\bar{u}$ up to any final time $t = T$. We note that $\mathcal{X} = L^1(\mathbb{T}) \cap L^\infty(\mathbb{T}) \subset L^2(\mathbb{T})$. As *input measure* $\mu \in \mathrm{Prob}(\mathcal{X})$, we consider random input functions $\bar{u} \sim \mu$ given by the square (box) wave of height $h$, width $w$ and centered at $\xi$,

$$\bar{u}(x) = h 1_{[-w/2, +w/2]}(x - \xi), \tag{3.2}$$

where $h \in [\underline{h}, \bar{h}]$, $w \in [\underline{w}, \bar{w}]$ $\xi \in [0, 2\pi]$ are independent and uniformly identically distributed. The constants $0 < \underline{h} \le \bar{h}$, $0 < \underline{w} \le \bar{w}$ are fixed.

**DeepONet fails at approximating $\mathcal{G}_{adv}$ efficiently.**    Our first rigorous result is the following lower bound on the error incurred by DeepONet (2.3) in approximating $\mathcal{G}_{\mathrm{adv}}$,

***Theorem 3.1.*** Let $p, m \in \mathbb{N}$. There exists a constant $C > 0$, independent of $m$, $p$ and $T$, such that for *any* DeepONet $\mathcal{N}^{\mathrm{DON}}$ (2.3), with $\sup_{\bar{u} \sim \mu} \|\mathcal{N}^{\mathrm{DON}}(\bar{u})\|_{L^\infty} \le M < \infty$, we have the lower bound

$$\mathscr{E} = \mathbb{E}_{\bar{u} \sim \mu} \left[ \|\mathcal{G}_{\mathrm{adv}}(\bar{u}) - \mathcal{N}^{\mathrm{DON}}(\bar{u})\|_{L^1} \right] \ge \frac{C}{\min(m, p)}.$$

Consequently, to achieve $\mathscr{E}(\mathcal{N}^{\mathrm{DON}}) \le \epsilon$ with DeepONet, we need $p$, $m \gtrsim \epsilon^{-1}$ trunk and branch net basis functions and sensor points, respectively, entailing that $\mathrm{size}(\mathcal{N}^{\mathrm{DON}}) \gtrsim pm \gtrsim \epsilon^{-2}$ (cp. **SM** B).

The detailed proof is presented in **SM** D.2. It relies on two facts. First, following Lanthaler et al. (2022), one observes that translation invariance of the problem implies that the Fourier basis is optimal for spanning the output space. As the underlying functions are discontinuous, the corresponding eigenvalues of the covariance operator for the push-forward measure decay, at most, quadratically in $p$. Consequently, the lower bound (2.4) leads to a linear decay of error in terms of the number of trunk net basis functions. Second, roughly speaking, the linear decay of error in terms of sensor points is a consequence of the fact that one needs sufficient number of sensor points to resolve the underlying discontinuous inputs.

**Shift-DeepONet approximates $\mathcal{G}_{adv}$ efficiently.**    Next and in contrast to the previous result on DeepONet, we have following efficient approximation result for shift-DeepONet (2.5),

***Theorem 3.2.*** There exists a constant $C > 0$, independent of $T$, such that for any $\epsilon > 0$ there exists a shift-DeepONet $\mathcal{N}^{\mathrm{sDON}}_\epsilon$ (2.5) such that

$$\mathscr{E} = \mathbb{E}_{\bar{u} \sim \mu} \left[ \|\mathcal{G}_{\mathrm{adv}}(\bar{u}) - \mathcal{N}^{\mathrm{sDON}}_\epsilon(\bar{u})\|_{L^1} \right] \le \epsilon, \tag{3.3}$$

with *uniformly bounded* $p \le C$, and with the number of sensor points $m \le C\epsilon^{-1}$. Furthermore, we have

$$\mathrm{width}(\mathcal{N}^{\mathrm{sDON}}_\epsilon) \le C, \quad \mathrm{depth}(\mathcal{N}^{\mathrm{sDON}}_\epsilon) \le C \log(\epsilon^{-1})^2, \quad \mathrm{size}(\mathcal{N}^{\mathrm{sDON}}_\epsilon) \le C\epsilon^{-1}.$$

The detailed proof, presented in **SM** D.3, is based on the fact that for each input, the exact solution can be completely determined in terms of three variables, i.e., the height $h$, width $w$ and shift $\xi$ of the box wave (3.2). Given an input $\bar{u}$, we explicitly construct neural networks for inferring each of these variables with high accuracy. These neural networks are then combined together to yield a shift-DeepONet that approximates $\mathcal{G}_{adv}$, with the desired complexity. The nonlinear dependence of the trunk net in shift-DeepONet (2.5) on the input is the key to encode the shift in the box-wave (3.2) and this demonstrates the necessity of nonlinear reconstruction in this context.

**FNO approximates $\mathcal{G}_{adv}$ efficiently.**    Finally, we state an efficient approximation result for $\mathcal{G}_{adv}$ with FNO (2.7) below, where the constant $C > 0$ is again independent of the final time $T$:

***Theorem 3.3.*** There exists $C > 0$, such that for any $\epsilon > 0$, there exists an FNO $\mathcal{N}^{\mathrm{FNO}}_\epsilon$ (2.7) with

$$\mathbb{E}_{\bar{u} \sim \mu} \left[ \|\mathcal{G}_{\mathrm{adv}}(\bar{u}) - \mathcal{N}^{\mathrm{FNO}}_\epsilon(\bar{u})\|_{L^1} \right] \le \epsilon,$$

with grid size $N \le C\epsilon^{-1}$, and with Fourier cut-off $k_{\max}$, lifting dimension $d_v$, depth and size:

$$k_{\max} = 1, \quad d_v \le C, \quad \mathrm{depth}(\mathcal{N}^{\mathrm{FNO}}_\epsilon) \le C \log(\epsilon^{-1})^2, \qquad \mathrm{size}(\mathcal{N}^{\mathrm{FNO}}_\epsilon) \le C \log(\epsilon^{-1})^2.$$

A priori, one recognizes that $\mathcal{G}_{adv}$ can be represented by Fourier multipliers (see **SM** D.4). Consequently, a single *linear* FNO layer would in principle suffice in approximating $\mathcal{G}_{adv}$. However, the size of this FNO would be *exponentially larger* than the bound in Theorem 3.3. To obtain a more efficient approximation, one needs to leverage the *nonlinear reconstruction* within FNO layers. This is provided in the proof, presented in **SM** D.4, where the underlying height, wave and shift of the box-wave inputs (3.2) are approximated with high accuracy by FNO layers. These are then combined together with a novel representation formula for the solution to yield the desired FNO.

**Comparison.** Observing the complexity bounds in Theorems 3.1, 3.2, 3.3, we note that the DeepONet size scales at least quadratically, size $\gtrsim \epsilon^{-2}$, in terms of the error in approximating $\mathcal{G}_{adv}$, whereas for shift-DeepONet and FNO, this scaling is only linear and logarithmic, respectively. Thus, we rigorously prove that for this problem, the nonlinear reconstruction methods (FNO and shift-DeepONet) can be more efficient than DeepONet and other methods based on linear reconstruction. Moreover, FNO is shown to have a smaller approximation error than even shift-DeepONet for similar model size. We provide two remarks on extensions of these results to the approximation of the time-evolution and to higher dimensions in **SM** C.

**Inviscid Burgers' equation.** Next, we consider the inviscid Burgers' equation in one-space dimension, which is considered the prototypical example of nonlinear hyperbolic conservation laws (Dafermos, 2005):

$$\partial_t u + \partial_x \left( \frac{1}{2} u^2 \right) = 0, \quad u(\,\cdot\,, t = 0) = \bar{u}, \tag{3.4}$$

on the $2\pi$-periodic domain $D = \mathbb{T}$. It is well-known that discontinuities in the form of shock waves can appear in finite-time even for smooth $\bar{u}$. Consequently, solutions of (3.4) are interpreted in the sense of distributions and entropy conditions are imposed to ensure uniqueness (Dafermos, 2005). Thus, the underlying solution operator is $\mathcal{G}_{\text{Burg}} : L^1(\mathbb{T}) \cap L^\infty(\mathbb{T}) \to L^1(\mathbb{T}) \cap L^\infty(\mathbb{T})$, $\bar{u} \mapsto \mathcal{G}_{\text{Burg}}(\bar{u}) := u(\,\cdot\,, T)$, with $u$ being the *entropy* solution of (3.4) at final time $T$. Given $\xi \sim \text{Unif}([0, 2\pi])$, we define the random field

$$\bar{u}(x) := -\sin(x - \xi), \tag{3.5}$$

and we define the input measure $\mu \in \text{Prob}(L^1(\mathbb{T}) \cap L^\infty(\mathbb{T}))$ as the law of $\bar{u}$. We emphasize that the difficulty in approximating the underlying operator $\mathcal{G}_{\text{Burg}}$ arises even though the input functions are smooth, in fact *analytic*. This is in contrast to the linear advection equation.

**DeepONet fails at approximating $\mathcal{G}_{\text{Burg}}$ efficiently.** First, we recall the following result, which follows directly from Lanthaler et al. (2022) (Theorem 4.19) and the lower bound (2.4),

**_Theorem 3.4._** Assume that $\mathcal{G}_{\text{Burg}} = u(\,\cdot\,, T)$, for $T > \pi$ and $u$ is the entropy solution of (3.4) with initial data $\bar{u} \sim \mu$. There exists a constant $C > 0$, such that the $L^2$-error for *any* DeepONet $\mathcal{N}^{\text{DON}}$ with $p$ trunk-/branch-net output functions is lower-bounded by

$$\mathscr{E}(\mathcal{N}^{\text{DON}}) = \mathbb{E}_{\bar{u} \sim \mu} \left[ \| \mathcal{G}_{\text{Burg}}(\bar{u}) - \mathcal{N}^{\text{DON}}(\bar{u}) \|_{L^1} \right] \geq \frac{C}{p}. \tag{3.6}$$

Consequently, achieving an error $\mathscr{E}(\mathcal{N}_\epsilon^{\text{DON}}) \lesssim \epsilon$ requires at least $\text{size}(\mathcal{N}_\epsilon^{\text{DON}}) \geq p \gtrsim \epsilon^{-1}$.

**shift-DeepONet approximate $\mathcal{G}_{\text{Burg}}$ efficiently.** In contrast to DeepONet, we have the following result for efficient approximation of $\mathcal{G}_{\text{Burg}}$ with shift-DeepONet,

**_Theorem 3.5._** Assume that $T > \pi$. There is a constant $C > 0$, such that for any $\epsilon > 0$, there exists a shift-DeepONet $\mathcal{N}_\epsilon^{\text{sDON}}$ such that

$$\mathscr{E}(\mathcal{N}_\epsilon^{\text{sDON}}) = \mathbb{E}_{\bar{u} \sim \mu} \left[ \| \mathcal{G}_{\text{Burg}}(\bar{u}) - \mathcal{N}_\epsilon^{\text{sDON}}(\bar{u}) \|_{L^1} \right] \leq \epsilon, \tag{3.7}$$

*with a uniformly bounded number $p \leq C$ of trunk/branch net functions, the number of sensor points can be chosen $m = 3$, and we have*

$$\text{width}(\mathcal{N}_\epsilon^{\text{sDON}}) \leq C, \quad \text{depth}(\mathcal{N}_\epsilon^{\text{sDON}}) \leq C \log(\epsilon^{-1})^2, \quad \text{size}(\mathcal{N}_\epsilon^{\text{sDON}}) \leq C \log(\epsilon^{-1})^2.$$

The proof, presented in **SM** D.5, relies on an explicit representation formula for $\mathcal{G}_{\text{Burg}}$, obtained using the method of characteristics (even after shock formation). Then, we leverage the analyticity of the underlying solutions away from the shock and use the nonlinear shift map in (2.5) to encode shock locations. Careful inspection of the proof implies that the constant $C$ is independent of $T > \pi$.

**FNO approximates** $\mathcal{G}_{\mathrm{Burg}}$ **efficiently** Finally we prove (in **SM** D.6) the following theorem,

**Theorem 3.6.** Assume that $T > \pi$, then there exists a constant $C$ (again independent of $T$), such that for any $\epsilon > 0$ and grid size $N \geq 3$, there exists an FNO $\mathcal{N}_\epsilon^{\mathrm{FNO}}$ (2.7), such that

$$\mathscr{E}(\mathcal{N}_\epsilon^{\mathrm{FNO}}) = \mathbb{E}_{\bar{u} \sim \mu} \left[ \|\mathcal{G}_{\mathrm{Burg}}(\bar{u}) - \mathcal{N}_\epsilon^{\mathrm{FNO}}(\bar{u})\|_{L^1} \right] \leq \epsilon,$$

and with Fourier cut-off $k_{\max}$, lifting dimension $d_v$ and depth satisfying,

$$k_{\max} = 1, \quad d_v \leq C, \quad \mathrm{depth}(\mathcal{N}_\epsilon^{\mathrm{FNO}}) \leq C \log(\epsilon^{-1})^2, \quad \mathrm{size}(\mathcal{N}_\epsilon^{\mathrm{FNO}}) \leq C \log(\epsilon^{-1})^2.$$

**Comparison.** A perusal of the bounds in Theorems 3.4, 3.5 and 3.6 reveals that *after shock formation*, the accuracy $\epsilon$ of the DeepONet approximation of $\mathcal{G}_{\mathrm{Burg}}$ scales at best as $\epsilon \sim n^{-1}$, in terms of the total number of degrees of freedom $n = \mathrm{size}(\mathcal{N}^{\mathrm{DON}})$ of the DeepONet. In contrast, shift-DeepONet and FNO based on a *nonlinear reconstruction* can achieve an exponential convergence rate $\epsilon \lesssim \exp(-cn^{1/2})$ in the total number of degrees of freedom $n = \mathrm{size}(\mathcal{N}^{\mathrm{sDON}}), \mathrm{size}(\mathcal{N}^{\mathrm{FNO}})$, even *after the formation of shocks*. This again highlights the expressive power of nonlinear reconstruction methods in approximating operators of PDEs with discontinuities.

|                        | ResNet  | FCNN    | DeepONet | Shift - DeepONet | FNO    |
|------------------------|---------|---------|----------|------------------|--------|
| **Advection Equation** | 14.8%   | 11.6%   | 7.95%    | 2.76%            | 0.71%  |
| **Burgers' Equation**  | 20.16%  | 23.23%  | 28.5%    | 7.83%            | 1.57%  |
| **Shocktube Problem**  | 4.47%   | 8.83%   | 4.22%    | 2.76%            | 1.56%  |
| **2D Riemann Problem** | 2.6%    | 0.19%   | 0.89%    | 0.12%            | 0.12%  |

Table 1: Relative median-$L^1$ error computed over 128 testing samples for different benchmarks with different models.

## 4 EXPERIMENTS

In this section, we illustrate how different operator learning frameworks can approximate solution operators of PDEs with discontinuities. To this end, we will compare DeepONet (2.3) (a prototypical operator learning method with linear reconstruction) with (the extended) shift-DeepONet (2.6) and FNO (2.7) (as nonlinear reconstruction methods). Moreover, two additional baselines (described in detail in **SM** E.1) are also used, namely the well-known *ResNet* architecture of He et al. (2016) and a fully convolutional neural network (FCNN) of Long et al. (2015). Below, we present results for the best performing hyperparameter configuration, obtained after a grid search, for each model while postponing the description of details for the training procedures, hyperparameter configurations and model parameters to **SM** E.1.

**Linear Advection.** We start with the linear advection equation (3.1) in the domain $[0, 1]$ with wave speed $a = 0.5$ and periodic boundary conditions. The initial data is given by (3.2) corresponding to square waves, with initial heights uniformly distributed between $\underline{h} = 0.2$ and $\overline{h} = 0.8$, widths between $\underline{w} = 0.05$ and $\overline{w} = 0.3$ and shifts between 0 and 0.5. We seek to approximate the solution operator $\mathcal{G}_{adv}$ at final time $T = 0.25$. The training and test samples are generated by sampling the initial data and the underlying exact solution, given by translating the initial data by 0.125, sampled on a very high-resolution grid of 2048 points (to keep the discontinuities sharp), see **SM** Figure 10 for examples of the input and output of $\mathcal{G}_{adv}$. The relative median test error for all the models are shown in Table 1. We observe from this table that DeepONet performs relatively poorly with a high test error of approximately 8%, although its outperforms the ResNet and FCNN baselines handily. As suggested by the theoretical results of the previous section, shift-DeepONet is significantly more accurate than DeepONet (and the baselines), with at least a two-fold gain in accuracy. Moreover, as predicted by the theory, FNO significantly outperforms even shift-DeepONet on this problem, with almost a five-fold gain in test accuracy and a thirteen-fold gain vis a vis DeepONet.

**Inviscid Burgers' Equation.** Next, we consider the inviscid Burgers' equation (3.4) in the domain $D = [0, 1]$ and with periodic boundary conditions. The initial data is sampled from a Gaussian Random field i.e., a Gaussian measure corresponding to the (periodization of) frequently used covariance kernel,

$$k(x, x') = \exp\left(\frac{-|x - x'|^2}{2\ell^2}\right),$$

with correlation length $\ell = 0.06$. The solution operator $\mathcal{G}_{Burg}$ corresponds to evaluating the entropy solution at time $T = 0.1$. We generate the output data with a high-resolution finite volume scheme, implemented within the *ALSVINN* code Lye (2020), at a spatial mesh resolution of 1024 points. Examples of input and output functions, shown in **SM** Figure 11, illustrate how the smooth yet oscillatory initial datum evolves into many discontinuities in the form of shock waves, separated by Lipschitz continuous rarefactions. Given this complex structure of the entropy solution, the underlying solution operator is hard to learn. The relative median test error for all the models is presented in Table 1 and shows that DeepOnet (and the baselines Resnet and FCNN) have an unacceptably high error between 20 and 30%. In fact, DeepONet performs worse than the two baselines. However, consistent with the theory of the previous section, this error is reduced more than three-fold with the nonlinear Shift-DeepONet. The error is reduced even further by FNO and in this case, FNO outperforms DeepOnet by a factor of almost 20 and learns the very complicated solution operator with an error of only 1.5%

**Compressible Euler Equations.** The motion of an inviscid gas is described by the Euler equations of aerodynamics. For definiteness, the Euler equations in two space dimensions are,

$$\mathbf{U}_t + \mathbf{F}(\mathbf{U})_x + \mathbf{G}(\mathbf{U})_y = 0, \quad \mathbf{U} = \begin{pmatrix} \rho \\ \rho u \\ \rho v \\ E \end{pmatrix}, \quad \mathbf{F}(\mathbf{U}) = \begin{pmatrix} \rho u \\ \rho u^2 + p \\ \rho uv \\ (E + p)u \end{pmatrix}, \quad \mathbf{G}(\mathbf{U}) = \begin{pmatrix} \rho v \\ \rho uv \\ \rho v^2 + p \\ (E + p)v \end{pmatrix},$$

(4.1)

with $\rho, u, v$ and $p$ denoting the fluid density, velocities along $x$-and $y$-axis and pressure. $E$ represents the total energy per unit volume

$$E = \frac{1}{2}\rho(u^2 + v^2) + \frac{p}{\gamma - 1}$$

where $\gamma = c_p/c_v$ is the gas constant which equals 1.4 for a diatomic gas considered here.

**Shock Tube.** We start by restricting the Euler equations (4.1) to the one-dimensional domain $D = [-5, 5]$ by setting $v = 0$ in (4.1). The initial data corresponds to a shock tube of the form,

$$\rho_0(x) = \begin{cases} \rho_L & x \leq x_0 \\ \rho_R & x > x_0 \end{cases} \quad u_0(x) = \begin{cases} u_L & x \leq x_0 \\ u_R & x > x_0 \end{cases} \quad p_0(x) = \begin{cases} p_L & x \leq x_0 \\ p_R & x > x_0 \end{cases}$$

(4.2)

parameterized by the left and right states $(\rho_L, u_L, p_L)$, $(\rho_R, u_R, p_R)$, and the location of the initial discontinuity $x_0$. As proposed in Lye et al. (2020), these parameters are, in turn, drawn from the measure; $\rho_L = 0.75 + 0.45G(z_1), \rho_R = 0.4 + 0.3G(z_2), u_L = 0.5 + 0.5G(z_3), u_R = 0, p_L = 2.5 + 1.6G(z_4), p_R = 0.375 + 0.325G(z_5), x_0 = 0.5G(z_6)$, with $z = [z_1, z_2, \ldots z_6] \sim U\left([0, 1]^6\right)$ and $G(z) := 2z - 1$. We seek to approximate the operator $\mathcal{G} : [\rho_0, \rho_0 u_0, E_0] \mapsto E(1.5)$. The training (and test) output are generated with *ALSVINN* code Lye (2020), using a finite volume scheme, with a spatial mesh resolution of 2048 points and examples of input-output pairs, presented in **SM** Figure 12 show that the initial jump discontinuities in density, velocity and pressure evolve into a complex pattern of (continuous) rarefactions, contact discontinuities and shock waves. The (relative) median test errors, presented in Table 1, reveal that shift-DeepONet and FNO significantly outperform DeepONet (and the other two baselines). FNO is also better than shift-DeepONet and approximates this complicated solution operator with a median error of $\approx 1.5\%$.

**Four-Quadrant Riemann Problem.** For the final numerical experiment, we consider the two-dimensional Euler equations (4.1) with initial data, corresponding to a well-known four-quadrant Riemann problem (Mishra & Tadmor, 2011) with $\mathbf{U}_0(x, y) = \mathbf{U}_{sw}$, if $x, y < 0$, $\mathbf{U}_0(x, y) = \mathbf{U}_{se}$, if $x < 0, y > 0$, $\mathbf{U}_0(x, y) = \mathbf{U}_{nw}$, if $x > 0, y < 0$ and $\mathbf{U}_0(x, y) = \mathbf{U}_{ne}$, if $x, y > 0$, with states given by $\rho_{0,ne} = \rho_{0,sw} = p_{0,ne} = p_{0,sw} = 1.1, \rho_{0,nw} = \rho_{0,se} = 0.5065$,

$p_{0,nw} = p_{0,se} = 0.35$ $[u_{0,ne}, u_{0,nw}, v_{0,ne}, v_{0,se}] = 0.35[G(z_1), G(z_2), G(z_3), G(z_4)]$ and $[u_{0,se}, u_{0,sw}, v_{0,nw}, v_{0,sw}] = 0.8939 + 0.35[G(z_5), G(z_6), G(z_7), G(z_48)]$, with $z = [z_1, z_2, \ldots z_8] \sim U([0,1]^8)$ and $G(z) = 2z - 1$. We seek to approximate the operator $\mathcal{G} : [\rho_0, \rho_0 u_0, \rho_0 v_0, E_0] \mapsto E(1.5)$. The training (and test) output are generated with the *ALSVINN* code, on a spatial mesh resolution of $256^2$ points and examples of input-output pairs, presented in **SM** Figure 13, show that the initial planar discontinuities in the state variable evolve into a very complex structure of the total energy at final time, with a mixture of curved and planar discontinuities, separated by smooth regions. The (relative) median test errors are presented in Table 1. We observe from this table that the errors with all models are significantly lower in this test case, possibly on account of the lower initial variance and coarser mesh resolution at which the reference solution is sampled. However, the same trend, vis a vis model performance, is observed i.e., DeepONet is significantly (more than seven-fold) worse than both shift-DeepONet and FNO. On the other hand, these two models approximate the underlying solution operator with a very low error of approximately $0.1\%$.

## 5 DISCUSSION

**Related Work.** Although the learning of operators arising from PDEs has attracted great interest in recent literature, there are very few attempts to extend the proposed architectures to PDEs with discontinuous solutions. Empirical results for some examples of discontinuities or sharp gradients were presented in Mao et al. (2020b) (compressible Navier-Stokes equations with DeepONets) Kissas et al. (2022) (Shallow-Water equations with an attention based framework) and in Seidman et al. (2022) (free-surface waves based on a "operator learning manifold hypothesis"). However, with the notable exception of Lanthaler et al. (2022) where the approximation of scalar conservation laws with DeepONets is analyzed, theoretical results for the operator approximation of PDEs are not available. Hence, this paper can be considered to be the first where a rigorous analysis of approximating operators arising in PDEs with discontinuous solutions has been presented, particularly for FNOs. On the other hand, there is considerably more work on the neural network approximation of parametric nonlinear hyperbolic PDEs such as the theoretical results of De Ryck & Mishra (2022a) and empirical results of Lye et al. (2020; 2021). Also related are results with physics informed neural networks or PINNs for nonlinear hyperbolic conservation laws such as De Ryck et al. (2022); Jagtap et al. (2022); Mao et al. (2020a). However, in this setting, the input measure is assumed to be supported on a finite-dimensional subset of the underlying infinite-dimensional input function space, making them too restrictive for operator learning as described in this paper.

**Conclusions.**

A priori, it could be difficult to approximate operators that arise in PDEs with discontinuities. Given this context, we have proved a rigorous lower bound to show that any operator learning architecture, based on linear reconstruction, may fail at approximating the underlying operator efficiently. In particular, this result holds for the popular DeepONet architecture.

On the other hand, we rigorously prove that the incorporation of nonlinear reconstruction mechanisms can break this lower bound and pave the way for efficient learning of operators arising from PDEs with discontinuities. We prove this result for an existing widely used architecture i.e., FNO, and a novel variant of DeepONet that we term as shift-DeepONet. For instance, we show that while the approximation error for DeepONets can decay, at best, linearly in terms of model size, the corresponding approximation errors for shift-DeepONet and FNO decays exponentially in terms of model size, *even in the presence or spontaneous formation of discontinuities.*

These theoretical results are backed by experimental results where we show that FNO and shift-DeepONet consistently beat DeepONet and other ML baselines by a wide margin, for a variety of PDEs with discontinuities.

Moreover, we also find theoretically (compare Theorems 3.2 and 3.3) that FNO is more efficient than even shift-DeepONet. This fact is also empirically confirmed in our experiments. The non-local as well as *nonlinear* structure of FNO is instrumental in ensuring its excellent performance in this context, see Theorem 3.3 and **SM** E.2 for further demonstration of the role of nonlinear reconstruction for FNOs.

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

# Supplementary Material for:
Nonlinear Reconstruction for operator learning of PDEs with discontinuities.

## A   PRINCIPAL COMPONENT ANALYSIS

Principal component analysis (PCA) provides a complete answer to the following problem (see e.g. Bhattacharya et al. (2021); Lanthaler et al. (2022) and references therein for relevant results in the infinite-dimensional context):

Given a probability measure $\nu \in \mathrm{Prob}(\mathcal{Y})$ on a Hilbert space $\mathcal{Y}$ and given $p \in \mathbb{N}$, we would like to characterize the optimal *linear* subspace $\widehat{V}_p \subset \mathcal{Y}$ of dimension $p$, which minimizes the average projection error

$$\mathbb{E}_{w \sim \nu} \left[ \|w - \Pi_{\widehat{V}_p} w\|_{\mathcal{Y}}^2 \right] = \min_{\dim(V_p) = p} \mathbb{E}_{w \sim \nu} \left[ \|w - \Pi_{V_p} w\|_{\mathcal{Y}}^2 \right], \tag{A.1}$$

where $\Pi_{V_p}$ denotes the orthogonal projection onto $V_p$, and the minimum is taken over all $p$-dimensional linear subspaces $V_p \subset \mathcal{X}$.

***Remark* A.1.** A characterization of the minimum in A.1 is of relevance to the present work, since the outputs of DeepONet, and other operator learning frameworks based on *linear reconstruction* $\mathcal{N}(u) = \sum_{k=1}^p \beta_k(u) \tau_k$, are restricted to the linear subspace $V_p := \mathrm{span}\{\tau_1, \ldots, \tau_p\}$. From this, it follows that (Lanthaler et al. (2022)):

$$\mathbb{E}_{u \sim \mu} \left[ \|\mathcal{G}(u) - \mathcal{N}(u)\|_{\mathcal{Y}}^2 \right] \geq \mathbb{E}_{u \sim \mu} \left[ \|\mathcal{G}(u) - \Pi_{V_p} \mathcal{G}(u)\|_{\mathcal{Y}}^2 \right] = \mathbb{E}_{w \sim \mathcal{G}_{\#} \mu} \left[ \|w - \Pi_{V_p} w\|_{\mathcal{Y}}^2 \right],$$

is lower bounded by the minimizer in (A.1) with $\nu = \mathcal{G}_{\#} \mu$ the push-forward measure of $\mu$ under $\mathcal{G}$.

To characterize minimizers of (A.1), one introduces the covariance operator $\Gamma_\nu : \mathcal{Y} \to \mathcal{Y}$, by $\Gamma_\nu := \mathbb{E}_{w \sim \nu} [w \otimes w]$, where $\otimes$ denotes the tensor product. By definition, $\Gamma_\nu$ satisfies the following relation

$$\langle v', \Gamma_\nu v \rangle_{\mathcal{Y}} = \mathbb{E}_{w \sim \nu} \left[ \langle v', w \rangle_{\mathcal{Y}} \langle w, v \rangle_{\mathcal{Y}} \right], \quad \forall v, v' \in \mathcal{Y}.$$

It is well-known that $\Gamma_\nu$ possesses a complete set of orthonormal eigenfunctions $\phi_1, \phi_2, \ldots$, with corresponding eigenvalues $\lambda_1 \geq \lambda_2 \geq \cdots \geq 0$. We then have the following result (see e.g. (Lanthaler et al., 2022, Thm. 3.8)):

***Theorem* A.2.** A subspace $\widehat{V}_p \subset \mathcal{Y}$ is a minimizer of (A.1) if, and only if, $\widehat{V}_p = \mathrm{span}\{\phi_1, \ldots, \phi_p\}$ can be written as the span of the first $p$ eigenfunctions of an orthonormal eigenbasis $\phi_1, \phi_2, \ldots$ of the covariance operator $\Gamma_\nu$, with decreasing eigenvalues $\lambda_1 \geq \lambda_2 \geq \ldots$. Furthermore, the minimum in (A.1) is given by

$$\mathscr{E}_{\mathrm{opt}}^2 = \min_{\dim(V_p) = p} \mathbb{E}_{w \sim \nu} \left[ \|w - \Pi_{V_p} w\|_{\mathcal{Y}}^2 \right] = \sum_{j > p} \lambda_j,$$

in terms of the decay of the eigenvalues of $\Gamma_\nu$.

### A.1   ILLUSTRATIVE EXAMPLE

To illustrate the close connection between the decay of PCA eigenvalues and the sharpness of gradients in that distribution, we consider the solution operator $\mathcal{G} : L^2(\mathbb{T}) \to L^2(\mathbb{T})$ of the advection equation $\partial_t u + a \partial_x u = 0$ with $a = 1$, mapping the initial data $\bar{u} \mapsto \mathcal{G}(\bar{u}) = u(t = 1)$ to the solution at time $t = 1$. We consider the input probability measure $\mu^{(0)} \in \mathcal{P}(L^2)$ which is defined as the law of random indicator functions $\bar{u} = 1_{[-w/2, w/2]}(x - \xi)$ on the periodic torus $\mathbb{T} = [0, 2\pi]$, where $w \in [\pi/4, 3\pi/4]$ and $\xi \in [0, 2\pi]$ are drawn uniformly at random and independently. For $\delta > 0$, we define a "smoothened" probability measure $\mu^{(\delta)}$ on input functions, whose law is obtained by mollifying random draws $\bar{u}$ from $\mu^{(0)}$ against a Gaussian mollifier $g_\delta(x)$ with Fourier coefficients $\widehat{g}_\delta(k) = \exp(-\delta^2 k^2)$, i.e.

$$\mu^{(\delta)} = \mathrm{law} \left( \bar{u} * g_\delta(y) \mid u \sim \mu^{(0)} \right).$$

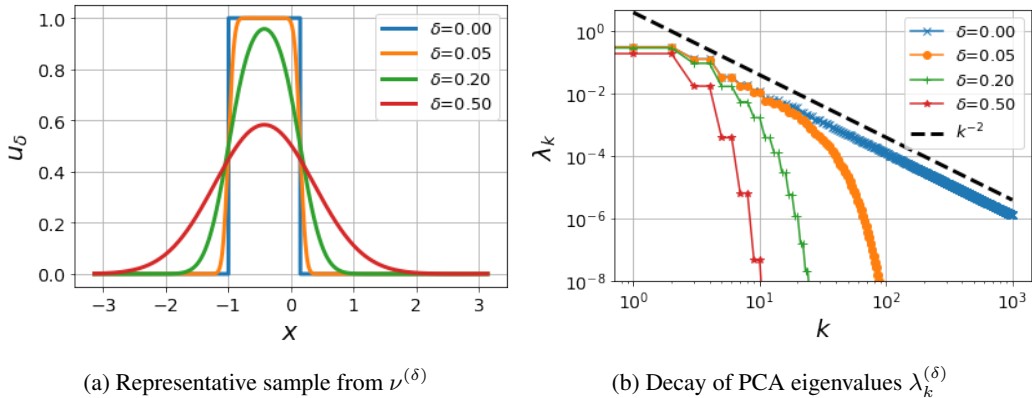

(a) Representative sample from $\nu^{(\delta)}$ · · · · · · · · · · (b) Decay of PCA eigenvalues $\lambda_k^{(\delta)}$

Figure 1: Sharper gradients (at length scale $\delta$) can cause a slow decay of the corresponding PCA eigenvalues $\lambda_k^{(\delta)}$.

We denote by $\nu^{(\delta)} = \mathcal{G}_\# \mu^{(\delta)}$ the push-forward measure under the solution operator $\mathcal{G}$. Observing that $\nu^{(\delta)}$ is a translation-invariant probability measure, and following (Lanthaler et al., 2022, Proof of Lemma 4.14), we note that the PCA eigenbasis is given by the standard Fourier basis, and the eigenvalue associated with the eigenfunction $e^{ikx}$ is given by

$$\widetilde{\lambda}_k^{(\delta)} = \frac{2\pi}{\Delta w} \int_{\pi/4}^{3\pi/4} |\widehat{\psi}_w^{(\delta)}(k)|^2 \, dw, \quad \psi_w^{(\delta)}(x) = 1_{[-w/2, w/2]} * g_\delta(x), \tag{A.2}$$

where $\widehat{\psi}_w^{(\delta)}(k) = (2\pi)^{-1} \int_0^{2\pi} \psi_w^{(\delta)}(x) e^{-ikx} \, dx$ denotes the $k$-th Fourier coefficient of $\psi_w^{(\delta)}$, and $\Delta w = \pi/2$ is a normalizing factor. We approximate $\widetilde{\lambda}_k^{(\delta)}$ numerically by a (trapezoidal) quadrature in $w$ with $N_w = 4001$ quadrature points. For each value of $w$, we approximate the Fourier coefficients $\widehat{\psi}_w^{(\delta)}(k)$ via the fast Fourier transform of $\psi_w^{(\delta)}(x)$ computed on a fine grid of $N = 10^4$ equidistant points. The approximate PCA eigenvalues are finally obtained by sorting the so computed eigenvalues in decreasing order and retaining only the first 1000, which yields a decreasing sequence $\lambda_1^{(\delta)} \geq \lambda_2^{(\delta)} \geq \cdots \geq \lambda_{1000}^{(\delta)}$.

Figure 1 compares and contrasts the decay of the PCA eigenvalues for the probability measures $\nu^{(\delta)}$ with gradients at different length scales $\delta \in \{0, 0.05, 0.2, 0.5\}$. We note that the theoretically predicted asymptotic $\lambda_k \sim C k^{-2}$ decay is recovered for $\delta = 0$.

## B    MEASURES OF COMPLEXITY FOR (SHIFT-)DEEPONET AND FNO

As pointed out in the main text, there are several hyperparameters which determine the complexity of DeepONet/shift-DeepONet and FNO, respectively. Table 2 summarizes quantities of major importance for (shift-)DeepONet and their (rough) analogues for FNO. These quantities are directly relevant to the expressive power and trainability of these operator learning architectures, and are described in further detail below.

| | (shift-)DeepONet | FNO |
|---|---|---|
| spatial resolution | $m$ | $\sim N^d$ |
| "intrinsic" function space dim. | $p$ | $\sim k_{\max}^d \cdot d_v$ |
| # trainable parameters | $\text{size}(\mathcal{N})$ | $\text{size}(\mathcal{N})$ |
| depth | $\text{depth}(\mathcal{N})$ | $\text{depth}(\mathcal{N})$ |
| width | $\text{width}(\mathcal{N})$ | $\sim k_{\max}^d \cdot d_v$ |

Figure 2: Approximate correspondence between measures of the complexity ($d$=dimension of the domain $D \subset \mathbb{R}^d$ of input/output functions).

**(shift-)DeepONet:** Quantities of interest include the number of sensor points $m$, the number of trunk-/branch-net functions $p$ and the width, depth and size of the operator network. We first recall the definition of the width and depth for DeepONet,

$$\text{width}(\mathcal{N}^{\text{DON}}) := \text{width}(\boldsymbol{\beta}) + \text{width}(\boldsymbol{\tau}),$$

$$\text{depth}(\mathcal{N}^{\text{DON}}) := \max\left\{\text{depth}(\boldsymbol{\beta}), \text{depth}(\boldsymbol{\tau})\right\},$$

where the width and depth of the conventional neural networks on the right-hand side are defined in terms of the maximum hidden layer width (number of neurons) and the number of hidden layers, respectively. To ensure a fair comparison between DeepONet, shift-DeepONet and FNO, we define the size of a DeepONet assuming a **fully connected (non-sparse) architecture**, as

$$\text{size}(\mathcal{N}^{\text{DON}}) := (m + p)\text{width}(\mathcal{N}^{\text{DON}}) + \text{width}(\mathcal{N}^{\text{DON}})^2\text{depth}(\mathcal{N}^{\text{DON}}),$$

where the second term measures the complexity of the hidden layers, and the first term takes into account the input and output layers. Furthermore, all architectures we consider have a width which scales at least as $\text{width}(\mathcal{N}^{\text{DON}}) \gtrsim \min(p, m)$, implying the following natural lower size bound,

$$\text{size}(\mathcal{N}^{\text{DON}}) \gtrsim (m + p)\min(p, m) + \min(p, m)^2\text{depth}(\mathcal{N}^{\text{DON}}). \tag{B.1}$$

We also introduce the analogous notions for shift-DeepONet:

$$\text{width}(\mathcal{N}^{\text{sDON}}) := \text{width}(\boldsymbol{\beta}) + \text{width}(\boldsymbol{\tau}) + \text{width}(\boldsymbol{\mathcal{A}}) + \text{width}(\boldsymbol{\gamma}),$$

$$\text{depth}(\mathcal{N}^{\text{sDON}}) := \max\left\{\text{depth}(\boldsymbol{\beta}), \text{depth}(\boldsymbol{\tau}), \text{depth}(\boldsymbol{\mathcal{A}}), \text{depth}(\boldsymbol{\gamma})\right\},$$

$$\text{size}(\mathcal{N}^{\text{sDON}}) := (m + p)\text{width}(\mathcal{N}^{\text{DON}}) + \text{width}(\mathcal{N}^{\text{DON}})^2\text{depth}(\mathcal{N}^{\text{DON}}).$$

**FNO:** Quantities of interest for FNO include the number of **grid points** in each direction $N$ (for a total of $O(N^d)$ grid points), the **Fourier cut-off** $k_{\max}$ (we retain a total of $O(k_{\max}^d)$ Fourier coefficients in the convolution operator and bias), and the **lifting dimension** $d_v$. We recall that the lifting dimension determines the number of components of the input/output functions of the hidden layers, and hence the "intrinsic" dimensionality of the corresponding function space in the hidden layers is proportional to $d_v$. The essential informational content for each of these $d_v$ components is encoded in their Fourier modes with wave numbers $|k| \leq k_{\max}$ (a total of $O(k_{\max}^d)$ Fourier modes per component), and hence the total intrinsic function space dimension of the hidden layers is arguably of order $\sim k_{\max}^d d_v$. The **width** of an FNO layer is defined in analogy with conventional neural networks as the maximal width of the weight matrices and Fourier multiplier matrices, which is of order $\sim k_{\max}^d \cdot d_v$. The **depth** is defined as the number of hidden layers $L$. Finally, the **size** is by definition the total number of tunable parameters in the architecture. By definition, the Fourier modes of the bias function $b_\ell(x)$ are restricted to wavenumbers $|k| \leq k_{\max}$ (giving a total number of $O(k_{\max}^d d_v)$ parameters), and the Fourier multiplier matrix is restricted to wave numbers $|k| \leq k_{\max}$ (giving $O(k_{\max}^d d_v^2)$ parameters). Apriori, it is easily seen that if the lifting dimension $d_v$ is larger than the number of components of the input/output functions, then (Kovachki et al., 2021b)

$$\text{size}(\mathcal{N}^{\text{FNO}}) \lesssim \left(d_v^2 + d_v^2 k_{\max}^d + d_v N^d\right)\text{depth}(\mathcal{N}^{\text{FNO}}),$$

where the first term in parentheses corresponds to $\text{size}(W_\ell) = d_v^2$, the second term accounts for $\text{size}(P_\ell) = O(d_v^2 k_{\max}^d)$ and the third term counts the degrees of freedom of the bias, $\text{size}(b_\ell(x_j)) = O(d_v N^d)$. The additional factor $\text{depth}(\mathcal{N}^{\text{FNO}})$ takes into account that there are $L = \text{depth}(\mathcal{N}^{\text{FNO}})$ such layers.

If the bias $b_\ell$ is constrained to have Fourier coefficients $\widehat{b}_\ell(k) \equiv 0$ for $|k| > k_{\max}$ (as we assumed in the main text), then the representation of $b_\ell$ only requires $\text{size}(\widehat{b}_\ell(k)) = O(d_v k_{\max}^d)$ degrees of freedom. This is of relevance in the regime $k_{\max} \ll N$, reducing the total FNO size from $O(d_v^2 N^d L)$ to

$$\text{size}(\mathcal{N}^{\text{FNO}}) \lesssim d_v^2 k_{\max}^d \text{depth}(\mathcal{N}^{\text{FNO}}). \tag{B.2}$$

Practically, this amounts to adding the bias in the hidden layers in Fourier $k$-space, rather than physical $x$-space.

## C  EXTENSIONS OF THEORETICAL RESULTS

In this section, we remark on two straight-forward extensions of our theoretical results in Section 3.

## C.1 TIME-EVOLUTION

We first consider the approximation of the time-evolution $t \mapsto u(\,\cdot\,, t)$ for solutions of the linear advection equation

$$\partial_t u + a \partial_x u = 0, \quad u(\,\cdot\,, t = 0) = \bar{u}, \tag{C.1}$$

with $\bar{u}$ drawn from the probability measure $\mu$ specified in (3.2) in the main text.

One way to apply neural operators to this time-evolution setting is by recursive application, where one fixes $\Delta t > 0$, and learns an approximation $\mathcal{N}(\bar{u}) \approx \mathcal{S}_{\Delta t}(\bar{u})$ for the given time-step, with $\bar{u} \mapsto \mathcal{S}_{\Delta t}(\bar{u})$ given by the data-to-solution mapping $\bar{u} \mapsto u(\,\cdot\,, \Delta t)$ of the PDE (C.1). Given $\bar{u}$, an approximation of the time-evolution $u(\,\cdot\,, t_j)$ at the discrete time-steps $t_j = j\Delta t$ for $j = 1, \dots, N_T$ with $N_T \Delta t = T$, is then obtained by iterative evaluations

$$u(\,\cdot\,, t_j) \approx \mathcal{N}_j(\bar{u}) := \underbrace{(\mathcal{N} \circ \cdots \circ \mathcal{N})}_{j \text{ times}}(\bar{u}).$$

In this context, Theorems 3.1, 3.2 and 3.3, can be extended to show that

- For **DeepONets**, we have a lower bound

$$\mathscr{E} = \sup_{j=1,;N_T} \mathbb{E}_{\bar{u} \sim \mu} \left[ \|\mathcal{S}_{t_j}(\bar{u}) - \mathcal{N}_j^{DON}(\bar{u})\|_{L^1} \right] \geq \frac{C}{\min(m, p)},$$

  with $C > 0$ independent of $T, \Delta t$. Consequently to achieve an error $\mathscr{E} < \epsilon$ requires $p, m \gtrsim \epsilon^{-1}$ trunk and branch net basis functions and sensor points, entailing a lower size bound $\text{size}(\mathcal{N}^{\text{DON}}) \gtrsim mp \gtrsim \epsilon^{-2}$.

- For **shift-DeepONets**, there exists a constant $C > 0$, independent of $\Delta t$ and $N_T$, such that for any $\epsilon > 0$ there exists a shift-DeepONet $\mathcal{N}^{\text{sDON}}$ such that

$$\mathscr{E} = \sup_{j=1,\dots,N_T} \mathbb{E}_{\bar{u} \sim \mu} \left[ \|\mathcal{S}_{t_j}(\bar{u}) - \mathcal{N}_j^{\text{sDON}}(\bar{u})\|_{L^1} \right] \leq \epsilon,$$

  with uniformly bounded $p \leq C$, with number of sensor points $m \leq C N_T / \epsilon$ and $\text{size}(\mathcal{N}^{\text{sDON}}) \leq C N_T / \epsilon$.

- For **FNO**, there exists $C > 0$ independent of $\Delta t$ and $N_T$, such that for any $\epsilon > 0$, there exists an FNO $\mathcal{N}^{\text{FNO}}$ with

$$\mathscr{E} = \sup_{j=1,\dots,N_T} \mathbb{E}_{\bar{u} \sim \mu} \left[ \|\mathcal{S}_{t_j}(\bar{u}) - \mathcal{N}_j^{\text{sDON}}(\bar{u})\|_{L^1} \right] \leq \epsilon,$$

  with grid size $N \leq C N_T / \epsilon$, Fourier cut-off $k_{\max} = 1$, lifting dimension $d_v \leq C$ and $\text{size}(\mathcal{N}^{\text{FNO}}) \leq C \log(N_T / \epsilon)^2$.

The additional factors of $N_T$ in these estimates stem from the fact that the errors over iterative time-steps accumulate, requiring an accuracy of order $\epsilon / N_T$ per time-step in order to achieve an cumulative error of at most $\epsilon$.

The above results imply that for any fixed choice of $\Delta t$ and $T = N_T \Delta t$, shift-DeepONets and FNO can approximate the time-evolution of (C.1) more efficiently than DeepONets. As an avenue for future work, it would be interesting to consider the approximation of the solution operator $\mathcal{S} : \bar{u} \mapsto u$, where the output function $u = u(x, t)$ depends on both position $x$, as well as on time $t$. But this is outside of the scope of the present work.

## C.2 EXTENSION TO HIGHER DIMENSIONS

While the analysis becomes considerably more cumbersome in higher dimensions, the main insights of this work also apply to problems on higher-dimensional domains, as we indicate for the linear advection example in the following: We consider the PDE

$$\partial_t u + \sum_{j=1}^{d} a_j \partial_{x_j} u = 0, \quad u(\,\cdot\,, t = 0) = \bar{u}, \tag{C.2}$$

where $\bar{u} = \bar{u}(x_1, \ldots, x_d)$ is a function defined on the $d$-dimensional torus $\mathbb{T}^d$, which we assume to be given by a random box wave of the form

$$\bar{u}(x_1, \ldots, x_d) = h \prod_{j=1}^{d} 1_{[-w_j/2, w_j/2]}(x_j - \xi_j). \tag{C.3}$$

Here $h \in [\underline{h}, \overline{h}]$, $w_j \in [\underline{w}, \overline{w}]$, and $\xi_j \in [0, 2\pi]$ are independent, uniform random variables. As in the 1-dimensional case, we consider the input probability measure $\mu$ defined as the law of this random box-wave in $d$-dimensions. Fixing $T > 0$, and by a slight abuse of notation, we will write the solution operator of (C.2) as $\mathcal{G}_{\mathrm{adv}}(\bar{u}) = u(\cdot, T)$.

**DeepONet:** Following an analysis analogous to the one-dimensional case, it can be shown that the $d$-dimensional Fourier basis provides an optimal PCA basis for the push-forward measure $\mathcal{G}_{\mathrm{adv}, \#}\mu$. Furthermore, an argument based on the decay of the Fourier coefficients of the box wave (C.3) implies that the PCA eigenvalues $\lambda_k$ satisfy the lower bound $\lambda_k \gtrsim k^{-2}$. In particular, this provides a similar lower bound on the number of required basis functions $p$ for the DeepONet approximation also in the $d$-dimensional case. Furthermore, an extension of the argument in the proof of Proposition D.10 also provides a similar lower bound in terms of $m$, i.e. if $\mathrm{ess\,sup}_{\bar{u} \sim \mu} \|\mathcal{G}_{\mathrm{adv}}(\bar{u})\|_{L^\infty} \le M$, then

$$\mathscr{E} = \mathbb{E}_{\bar{u} \sim \mu} \left[ \|\mathcal{G}_{\mathrm{adv}}(\bar{u}) - \mathcal{N}^{\mathrm{DON}}(\bar{u})\|_{L^1} \right] \ge \frac{C}{\min(m, p)},$$

for a constant $C = C(M) > 0$ that is independent of $p$ and $T$. Again this shows that a large number of basis functions is necessary to approximation $\mathcal{G}_{\mathrm{adv}}$ by a DeepONet. We note that we can only establish a lower bound $\gtrsim 1/m$ rather than $\gtrsim 1/m^{1/d}$, as one might have expected to appear from the "curse of dimensionality" associated with higher-dimensional problems.

**shift-DeepONet:** In contrast to the case of DeepONet, for shift-DeepONet it can be shown that there exists a constant $C = C(d) > 0$, such that for any $\epsilon > 0$, there exists a shift-DeepONet $\mathcal{N}^{\mathrm{sDON}}$ such that

$$\mathscr{E} = \mathbb{E}_{\bar{u} \sim \mu} \left[ \|\mathcal{G}_{\mathrm{adv}}(\bar{u}) - \mathcal{N}^{\mathrm{sDON}}(\bar{u})\|_{L^1} \right] \le \epsilon,$$

and with a *bounded number* of basis functions $p \le C$, a number of sensor points $m \le C\epsilon^{-1}$, and such that $\mathrm{size}(\mathcal{N}^{\mathrm{sDON}}) \le C\epsilon^{-1}$.

*Sketch of proof:* The idea underlying this construction is to first reduce the multi-$d$ problem to $d$ one-dimensional problems, to apply the known result in the one-dimensional case (along each coordinate) and to finally reconstruct the output wave-form from the solutions of the one-dimensional problems. To reduce to the one-dimensional case, we fix a coarse uniform grid $x_\ell^{\mathrm{coarse}}$, $\ell = 1, \ldots, N_{\mathrm{coarse}}$, on $\mathbb{T} = [0, 2\pi)$ with step size $< \underline{w}$ (smaller than the smallest possible box width in (C.3)), as well as a fine uniform grid $x_\ell^{\mathrm{fine}}$, $\ell = 1, \ldots, N_{\mathrm{fine}}$, on $\mathbb{T} = [0, 2\pi)$ with step size $\sim \epsilon$. In terms of these one-dimensional grids, and for any given coordinate direction $j \in \{1, \ldots, d\}$, we fix sensor points

$$\boldsymbol{x}_{\ell_1, \ldots, \ell_d}^{(j)} = (x_{\ell_1}^{\mathrm{coarse}}, \ldots, x_{\ell_{j-1}}^{\mathrm{coarse}} x_{\ell_j}^{\mathrm{fine}}, x_{\ell_{j+1}}^{\mathrm{coarse}}, \ldots, x_{\ell_d}^{\mathrm{coarse}}) \in \mathbb{T}^d,$$

where $\ell_j = 1, \ldots, N_{\mathrm{fine}}$, and where the other $\ell_k$ ($k \ne j$) run over the coarse index set $\ell_k = 1, \ldots, N_{\mathrm{coarse}}$. Since $N_{\mathrm{coarse}}$ is independent of $\epsilon$, while $N_{\mathrm{fine}} \sim \epsilon^{-1}$, the total number $m$ of sensor points

$$\left\{ \boldsymbol{x}_{\ell_1, \ldots, \ell_d}^{(j)} \,\middle|\, j \in \{1, \ldots, d\}, \ell_j \in \{1, \ldots, N_{\mathrm{fine}}\}, \ell_k \in \{1, \ldots, N_{\mathrm{coarse}}\} \text{ for } k \ne j \right\},$$

can be bounded by $m \le C\epsilon^{-1}$, where $C = C(d, \underline{w})$ is independent of $\epsilon$.

Importantly for the reduction to the one-dimensional case:

- the height $h$ of the $d$-dimensional box wave can be obtained by taking the maximum over the sensor values $\bar{u}(\boldsymbol{x}_{\ell_1, \ldots, \ell_d}^{(j)})$, with $\ell_1, \ldots, \ell_d$ belonging to a *coarse* subset of indices (cp. Step 1 in the proof in **SM** D.3; this requires a neural network of size $O(\epsilon^{-1})$),

- for any box wave of the form (C.3), and for any direction $j = 1, \ldots, d$, a summation over the coarse indices (i.e. sum over $\ell_k$ for $k \ne j$) of the encoded values $\bar{u}(\boldsymbol{x}_{\ell_1, \ldots, \ell_d})$ combined with a ReLU truncation, allows us to construct a DNN mapping

$$\bar{u} = h \prod_{k=1}^{d} 1_{[-w_k/2, w_k/2]}(x_k - \xi_k) \mapsto 1_{[-w_j/2, w_j/2]}(x_j - \xi_j),$$

where the output function is encoded by evaluation at the fine grid points $x_\ell^{\text{fine}}$, $\ell = 1, \ldots, N_{\text{fine}}$.

Given this reduction to the one-dimensional case, we can then apply our one-dimensional results to find suitable approximations of $h$, $w_j$, $\xi_j$, $(j = 1, \ldots, d)$ as in the one-dimensional case. Based on this, we construct an approximation of the box wave solution $u(\cdot, T) = h \prod_{j=1}^d 1_{[-w_j/2, w_j/2]}(x_j - \xi_j - a_j T)$ by defining the trunk net as a suitable approximation of the $d$-dimensional unit box $\tau(x) \approx \prod_{j=1}^d 1_{[-1,1]}(x_j)$, defining the scale-net to scale each coordinate direction by $\mathcal{A}(\bar{u}) \approx \text{diag}(1/w_1, \ldots, 1/w_d)$, setting the shift-net to be a shift by $\gamma(\bar{u}) \approx (\xi_j + a_j T)$, and finally defining the branch net to be equal to the box height $\beta(\bar{u}) = h$, so that

$$u(\cdot, T) = h \prod_{j=1}^d 1_{[-w_j/2, w_j/2]}(x_j - \xi_j - a_j T) \approx \beta(\bar{u}) \tau\left(\mathcal{A}(\bar{u}) \cdot x + \gamma(\bar{u})\right),$$

provides the desired approximation of the box wave solution. We will not provide the precise details and required estimates here.

**FNO:** Similarly, for FNO it can be shown that there exists a constant $C = C(d) > 0$, such that for any $\epsilon > 0$, there exists an FNO $\mathcal{N}^{\text{FNO}}$ such that

$$\mathscr{E} = \mathbb{E}_{\bar{u} \sim \mu}\left[\|\mathcal{G}_{\text{adv}}(\bar{u}) - \mathcal{N}^{\text{FNO}}(\bar{u})\|_{L^1}\right] \le \epsilon,$$

and with a *bounded* truncation parameter $k_{\max} = 1$, lifting dimension $d_v \le C$, a number of grid points $N \le C\epsilon^{-d}$, and such that $\text{size}(\mathcal{N}^{\text{FNO}}) \le C \log(\epsilon^{-1})^2$.

*Sketch of proof:* The idea is very similar to the case of shift-DeepONet, and follows by a reduction to the one-dimensional case. Note that in this case, a fine grid needs to be chosen in each coordinate direction (requiring many grid points, of order $\sim \epsilon^{-d}$), but crucially the number of weights and biases of the FNO architecture is *independent* of this discretization parameter. Hence this does not affect the overall size. In the case of FNOs, we furthermore note that the box wave

$$\bar{u}(x) = h \prod_{j=1}^d 1_{[-w_j/2, w_j/2]}(x_j - \xi_j),$$

can be uniquely reconstructed (by a *nonlinear* reconstruction procedure) from knowledge of it's Fourier coefficients

$$\mathcal{F}\bar{u}(k_1, \ldots, k_d), \quad \text{for } k_1, \ldots, k_{j-1}, k_{j+1}, \ldots, k_d = 0, \ k_j \in \{-1, 0, 1\}, \text{ and } j \in \{1, \ldots, d\},$$

following the same approach detailed in the one-dimensional case in **SM** D.4, below.

Therefore, given the Fourier coefficients $\mathcal{F}\bar{u}(k) = \mathcal{F}\bar{u}(k_1, \ldots, k_d)$, for $k = (k_1, \ldots, k_d)$ with $|k| \le 1$, we can first compute the corresponding Fourier coefficients of the solution $u(\cdot, T) = \bar{u}(\cdot - aT)$, by a simple phase shift $\mathcal{F}[u(\cdot, T)](k) = \mathcal{F}\bar{u}(k)e^{-i(k \cdot a)T}$, and then reconstruct $\bar{u}(\cdot - aT)$ from knowledge of these Fourier coefficients. Thus $k_{\max} = 1$ also suffices in this case (corresponding to $\sim 3d$ Fourier coefficients which are retained). The lifting dimension $d_v$ of the detailed construction outlined above scales linearly in the dimension of the domain $d$, but is independent of $\epsilon$.[1]

## D   MATHEMATICAL DETAILS

In this section, we provide detailed proofs of the Theorems in Section 3. We start with some preliminary results below,

### D.1   ReLU DNN BUILDING BLOCKS

In the present section we collect several basic constructions for ReLU neural networks, which will be used as building blocks in the following analysis. For the first result, we note that for any $\delta > 0$, the

---

[1] Similar to the one-dimensional case, additional shifted copies of these basis functions are needed in practice to account for the $2\pi$-periodicity of the output function.

following approximate step-function

$$\zeta_\delta(x) := \begin{cases} 0, & x < 0, \\ \frac{x}{\delta}, & 0 \le x \le \delta, \\ 1, & x > \delta, \end{cases}$$

can be represented by a neural network:

$$\zeta_\delta(x) = \sigma\left(\frac{x}{\delta}\right) - \sigma\left(\frac{x-\delta}{\delta}\right),$$

where $\sigma(x) = \max(x,0)$ denotes the ReLU activation function. Introducing an additional shift $\xi$, multiplying the output by $h$, and choosing $\delta > 0$ sufficiently small, we obtain the following result:

***Proposition* D.1** (Step function). Fix an interval $[a,b] \subset \mathbb{R}$, $\xi \in [a,b]$, $h \in \mathbb{R}$. Let $h\,1_{[x>\xi]}$ be a step function of height $h$. For any $\epsilon > 0$ and $p \in [1,\infty)$, there exist a ReLU neural network $\Phi_\epsilon : \mathbb{R} \to \mathbb{R}$, such that

$$\mathrm{depth}(\Phi_\epsilon) = 1, \quad \mathrm{width}(\Phi_\epsilon) = 2,$$

and

$$\|\Phi_\epsilon - h\,1_{[x>\xi]}\|_{L^p([a,b])} \le \epsilon.$$

The following proposition is an immediate consequence of the previous one, by considering the linear combination $\Phi_\delta(x-a) - \Phi_\delta(x-b)$ with a suitable choice of $\delta > 0$.

***Proposition* D.2** (Indicator function). Fix an interval $[a,b] \subset \mathbb{R}$. Let $1_{[a,b]}(x)$ be the indicator function of $[a,b]$. For any $\epsilon > 0$ and $p \in [1,\infty)$, there exist a ReLU neural network $\Phi_\epsilon : \mathbb{R} \to \mathbb{R}$, such that

$$\mathrm{depth}(\Phi_\epsilon) = 1, \quad \mathrm{width}(\Phi_\epsilon) = 4,$$

and

$$\|\Phi_\epsilon - 1_{[a,b]}\|_{L^p([a,b])} \le \epsilon.$$

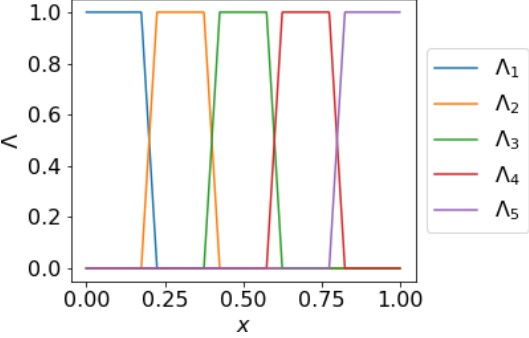

Figure 3: Illustration of partition of unity network for $J = 5$, $[a,b] = [0,1]$.

A useful mathematical technique to glue together local approximations of a given function rests on the use of a "partition of unity". In the following proposition we recall that partitions of unity can be constructed with ReLU neural networks (this construction has previously been used by Yarotsky (2017); cp. Figure 3):

***Proposition* D.3** (Partition of unity). Fix an interval $[a,b] \subset \mathbb{R}$. For $J \in \mathbb{N}$, let $\Delta x := (b-a)/J$, and let $x_j := a + j\Delta x$, $j = 0,\ldots,J$ be an equidistant grid on $[a,b]$. Then for any $\epsilon \in (0, \Delta x/2]$, there exists a ReLU neural network $\Lambda : \mathbb{R} \to \mathbb{R}^J$, $x \mapsto (\Lambda_1(x),\ldots,\Lambda_J(x))$, such that

$$\mathrm{width}(\Lambda) = 4J, \quad \mathrm{depth}(\Lambda) = 1,$$

each $\Lambda_j$ is piecewise linear, satisfies

$$\Lambda_j(x) = \begin{cases} 0, & (x \le x_{j-1} - \epsilon), \\ 1, & (x_{j-1} + \epsilon \le x \le x_j - \epsilon), \\ 0, & (x \ge x_j + \epsilon), \end{cases}$$

and interpolates linearly between the values 0 and 1 on the intervals $[x_{j-1} - \epsilon, x_{j-1} + \epsilon]$ and $[x_j - \epsilon, x_j + \epsilon]$. In particular, this implies that

- $\operatorname{supp}(\Lambda_j) \subset [x_{j-1} - \epsilon, x_j + \epsilon]$, for all $j = 1, \ldots, J$,

- $\Lambda_j(x) \ge 0$ for all $x \in \mathbb{R}$,

- The $\{\Lambda_j\}_{j=1,\ldots,J}$ form a partition of unity, i.e.

$$\sum_{j=1}^{J} \Lambda_j(x) = 1, \quad \forall\, x \in [a, b].$$

We also recall the well-known fact that the multiplication operator $(x, y) \mapsto xy$ can be efficiently approximated by ReLU neural networks (cp. Yarotsky (2017)):

***Proposition* D.4** (Multiplication, (Yarotsky, 2017, Prop. 3)). There exists a constant $C > 0$, such that for any $\epsilon \in (0, \frac{1}{2}]$, $M \ge 2$, there exists a neural network $\widehat{\times}_{\epsilon,M} : [-M, M] \times [-M, M] \to \mathbb{R}$, such that

$$\operatorname{width}(\widehat{\times}_{\epsilon,M}) \le C, \quad \operatorname{depth}(\widehat{\times}_{\epsilon,M}) \le C \log(M\epsilon^{-1}), \quad \operatorname{size}(\widehat{\times}_{\epsilon,M}) \le C \log(M\epsilon^{-1}),$$

and

$$\sup_{x,y \in [-M,M]} |\widehat{\times}_{\epsilon,M}(x, y) - xy| \le \epsilon.$$

We next state a general approximation result for the approximation of analytic functions by ReLU neural networks. To this end, we first recall

***Definition* D.5** (Analytic function and extension). A function $F : (\alpha, \beta) \to \mathbb{R}$ is **analytic**, if for any $x_0 \in (\alpha, \beta)$ there exists a radius $r > 0$, and a sequence $(a_k)_{k \in \mathbb{N}_0}$ such that $\sum_{k=0}^{\infty} |a_k| r^k < \infty$, and

$$F(x) = \sum_{k=0}^{\infty} a_k (x - x_0)^k, \quad \forall\, |x - x_0| < r.$$

If $f : [a, b] \to \mathbb{R}$ is a function, then we will say that $f$ has an **analytic extension**, if there exists $F : (\alpha, \beta) \to \mathbb{R}$, with $[a, b] \subset (\alpha, \beta)$, with $F(x) = f(x)$ for all $x \in [a, b]$ and such that $F$ is analytic.

We then have the following approximation bound, which extends the main result of Wang et al. (2018). In contrast to Wang et al. (2018), the following theorem applies to analytic functions without a globally convergent series expansion.

***Theorem* D.6.** Assume that $f : [a, b] \to \mathbb{R}$ has an analytic extension. Then there exist constants $C, \gamma > 0$, depending only on $f$, such that for any $L \in \mathbb{N}$, there exists a ReLU neural network $\Phi_L : \mathbb{R} \to \mathbb{R}$, with

$$\sup_{x \in [a,b]} |f(x) - \Phi_L(x)| \le C \exp(-\gamma L^{1/2}),$$

and such that

$$\operatorname{depth}(\Phi_L) \le CL, \quad \operatorname{width}(\Phi_L) \le C.$$

*Proof.* Since $f$ has an analytic extension, for any $x \in [a, b]$, there exists a radius $r_x > 0$, and an analytic function $F_x : [x - r_x, x + r_x] \to \mathbb{R}$, which extends $f$ locally. By the main result of (Wang et al., 2018, Thm. 6), there are constants $C_x, \gamma_x > 0$ depending only on $x \in [a, b]$, such that for any $L \in \mathbb{N}$, there exists a ReLU neural network $\Phi_{x,L} : \mathbb{R} \to \mathbb{R}$, such that

$$\sup_{|\xi - x| \le r_x} |F(\xi) - \Phi_{x,L}(\xi)| \le C_x \exp(-\gamma_x L^{1/2}),$$

and $\mathrm{depth}(\Phi_x) \le C_x L$, $\mathrm{width}(\Phi_x) \le C_x$. For $J \in \mathbb{N}$, set $\Delta x = (b - a)/J$ and consider the equidistant partition $x_j := a + j\Delta x$ of $[a, b]$. Since the compact interval $[a, b]$ can be covered by finitely many of the intervals $(x - r_x, x + r_x)$, then by choosing $\Delta x$ sufficiently small, we can find $x^{(j)} \in [a, b]$, such that $[x_{j-1}, x_j] \subset (x^{(j)} - r_{x^{(j)}}, x^{(j)} + r_{x^{(j)}})$ for each $j = 1, \ldots, J$.

By construction, this implies that for $\bar{C} := \max_{j=1,\ldots,J} C_{x^{(j)}}$, and $\bar{\gamma} := \min_{j=1,\ldots,J} \gamma_{x^{(j)}}$, we have that for any $L \in \mathbb{N}$, there exist neural networks $\Phi_{j,L} (= \Phi_{x^{(j)},L}) : \mathbb{R} \to \mathbb{R}$, such that

$$\sup_{x \in [x_{j-1}, x_j]} |f(x) - \Phi_{j,L}(x)| \le \bar{C} \exp(-\bar{\gamma} L^{1/2}),$$

and such that $\mathrm{depth}(\Phi_{j,L}) \le \bar{C} L$, $\mathrm{width}(\Phi_{j,L}) \le \bar{C}$.

Let now $\Lambda : \mathbb{R} \to \mathbb{R}^J$ be the partition of unity network from Proposition D.3, and define

$$\Phi_L(x) := \sum_{j=1}^J \widetilde{\times}_{M,\epsilon} \left( \Lambda_j(x), \Phi_{j,L}(x) \right),$$

where $\widetilde{\times}_{M,\epsilon}$ denotes the multiplication network from Proposition D.4, with $M := 1 + \bar{C} \exp(-\bar{\gamma}) + \sup_{x \in [a,b]} |f(x)|$, and $\epsilon := J^{-1} \bar{C} \exp(-\bar{\gamma} L^{1/2})$. Then we have

$$\mathrm{depth}(\Phi_L) \le \mathrm{depth}(\widetilde{\times}_{M,\epsilon}) + \mathrm{depth}(\Lambda) + \max_{j=1,\ldots,J} \mathrm{depth}(\Phi_{j,L})$$
$$\le C' \log(M\epsilon^{-1}) + 1 + \bar{C} L$$
$$\le C(1 + L),$$

where the constant $C > 0$ on the last line depends on $\sup_{x \in [a,b]} |f(x)|$, $\bar{\gamma}$ and on $\bar{C}$, but is independent of $L$. Similarly, we find that

$$\mathrm{width}(\Phi_L) \le \mathrm{width}(\Lambda) + \max_{j=1,\ldots,J} \mathrm{width}(\Phi_{j,L}) \le 4J + \bar{C},$$

is bounded independently of $L$. After potentially enlarging the constant $C > 0$, we can thus ensure that

$$\mathrm{depth}(\Phi_L) \le CL, \quad \mathrm{width}(\Phi_L) \le C,$$

with a constant $C > 0$ that depends only on $f$, but is independent of $L$. Finally, we note that

$$|\Phi_L(x) - f(x)| \le \sum_{j=1}^J \left| \widetilde{\times}_{M,\epsilon} \left( \Lambda_j(x), \Phi_{j,L}(x) \right) - \Lambda_j(x) f(x) \right|$$
$$\le \sum_{j=1}^J \left| \widetilde{\times}_{M,\epsilon} \left( \Lambda_j(x), \Phi_{j,L}(x) \right) - \Lambda_j(x) \Phi_{j,L}(x) \right|$$
$$+ \sum_{j=1}^J \Lambda_j(x) |\Phi_{j,L}(x) - f(x)|.$$

By construction of $\widetilde{\times}_{M,\epsilon}$, and since $|\Phi_{j,L}(x)| \le M$, $\Lambda_j \le M$, the first sum can be bounded by $J\epsilon = \bar{C} \exp(-\bar{\gamma} L^{1/2})$. Furthermore, each term in the second sum is bounded by $\Lambda_j(x) \bar{C} \exp(-\bar{\gamma} L^{1/2})$, and hence

$$\sum_{j=1}^J \Lambda_j(x) |\Phi_{j,L}(x) - f(x)| \le \left( \sum_{j=1}^J \Lambda_j(x) \right) \bar{C} \exp(-\bar{\gamma} L^{1/2}) = \bar{C} \exp(-\bar{\gamma} L^{1/2}),$$

for all $x \in [a, b]$. We conclude that $\sup_{x \in [a,b]} |\Phi_L(x) - f(x)| \leq 2\bar{C} \exp(-\bar{\gamma} L^{1/2})$, with constants $\bar{C}, \bar{\gamma} > 0$ independent of $L$. Setting $\gamma := \bar{\gamma}$ and after potentially enlarging the constant $C > 0$ further, we thus conclude: there exist $C, \gamma > 0$, such that for any $L \in \mathbb{N}$, there exists a neural network $\Phi_L : \mathbb{R} \to \mathbb{R}$ with $\mathrm{depth}(\Phi_L) \leq CL$, $\mathrm{width}(\Phi_L) \leq C$, such that

$$\sup_{x \in [a,b]} |\Phi_L(x) - f(x)| \leq C \exp(-\gamma L^{1/2}).$$

This conclude the proof of Theorem D.6 $\qquad\qquad\qquad\qquad\qquad\qquad\qquad\qquad \square$

By combining a suitable ReLU neural network approximation of division $a \mapsto 1/a$ based on Theorem D.6 (division is an analytic function away from 0), and the approximation of multiplication by Yarotsky (2017) (cp. Proposition D.4, above), we can also state the following result:

***Proposition* D.7** (Division). Let $0 < a \leq b$ be given. Then there exists $C = C(a, b) > 0$, such that for any $\epsilon \in (0, \frac{1}{2}]$, there exists a ReLU network $\widetilde{\div}_{a,b,\epsilon} : \mathbb{R} \times \mathbb{R} \to \mathbb{R}$, with

$$\mathrm{depth}(\widetilde{\div}_{a,b,\epsilon}) \leq C \log\left(\epsilon^{-1}\right)^2, \quad \mathrm{width}(\widetilde{\div}_{a,b,\epsilon}) \leq C, \quad \mathrm{size}(\widetilde{\div}_{a,b,\epsilon}) \leq C \log\left(\epsilon^{-1}\right)^2,$$

satisfying

$$\sup_{x,y \in [a,b]} \left| \widetilde{\div}_{a,b,\epsilon}(x; y) - \frac{x}{y} \right| \leq \epsilon.$$

We end this section with the following result.

***Lemma* D.8.** There exists a constant $C > 0$, such that for any $\epsilon > 0$, there exists a neural network $\Xi_\epsilon : \mathbb{R} \to \mathbb{R}$, such that

$$\sup_{\xi \in [0, 2\pi - \epsilon]} |\xi - \Xi_\epsilon(\cos(\xi), \sin(\xi))| \leq \epsilon,$$

with $\Xi_\epsilon(\cos(\xi), \sin(\xi)) \in [0, 2\pi]$ for all $\xi \in [0, 2\pi]$, and such that

$$\mathrm{depth}(\Xi_\epsilon) \leq C \log(\epsilon^{-1})^2, \quad \mathrm{width}(\Xi_\epsilon) \leq C.$$

*Sketch of proof.* We can divide up the unit circle $\{(\cos(\xi), \sin(\xi)) \,|\, \xi \in [0, 2\pi]\}$ into 5 subsets, where

$$\begin{cases} \xi \in [0, \pi/4], & \Longleftrightarrow \ x \geq 1/\sqrt{2}, \ y \geq 0, \\ \xi \in (\pi/4, 3\pi/4], & \Longleftrightarrow \ y > 1/\sqrt{2}, \\ \xi \in [3\pi/4, 5\pi/4], & \Longleftrightarrow \ x \leq -1/\sqrt{2}, \\ \xi \in (5\pi/4, 7\pi/4), & \Longleftrightarrow \ y < -1/\sqrt{2}, \\ \xi \in [7\pi/4, 2\pi), & \Longleftrightarrow \ x \leq -1/\sqrt{2}, \ y < 0, \end{cases}$$

On each of these subsets, one of the mappings

$$x \in \left[ -\frac{1}{\sqrt{2}}, \frac{1}{\sqrt{2}} \right] \to \mathbb{R}, \quad x = \cos(\xi) \mapsto \xi,$$

or

$$y \in \left[ -\frac{1}{\sqrt{2}}, \frac{1}{\sqrt{2}} \right] \to \mathbb{R}, \quad y = \sin(\xi) \mapsto \xi,$$

is well-defined and possesses an analytic, invertible extension to the open interval $(-1, 1)$ (with analytic inverse). By Theorem D.6, it follows that for any $\epsilon > 0$, we can find neural networks $\Phi_1, \ldots, \Phi_5$, such that $|\Phi_j(\cos(\xi), \sin(\xi)) - \xi| \leq \epsilon$ on an open set containing the corresponding domain, and

$$\mathrm{depth}(\Phi_j) \leq C \log(\epsilon^{-1})^2, \quad \mathrm{width}(\Phi_j) \leq C.$$

By a straight-forward partition of unity argument based on Proposition D.3, we can combine these mappings to a global map,[2] which is represented by a neural network $\Xi_\epsilon : \mathbb{R}^2 \to [0, 2\pi]$, such that

$$\sup_{\xi \in [0, 2\pi - \epsilon]} |\Xi_\epsilon(\cos(\xi), \sin(\xi)) - \xi| \leq \epsilon,$$

---

[2]this step is where the $\epsilon$-gap at the right boundary $2\pi - \epsilon$ is needed, as the points at angles $\xi = 0$ and $\xi = 2\pi$ are identical on the circle.

and such that

$$\text{depth}(\Xi_\epsilon) \leq C \log(\epsilon^{-1})^2, \quad \text{width}(\Xi_\epsilon) \leq C.$$

$\square$

### D.2 PROOF OF THEOREM 3.1

The proof of this theorem follows from the following two propositions,

***Proposition* D.9** (Lower bound in $p$). Consider the solution operator $\mathcal{G}_{\text{adv}} : L^1 \cap L^\infty(\mathbb{T}) \to L^1 \cap L^\infty(\mathbb{T})$ of the linear advection equation, with input measure $\mu$ given as the random law of box functions of height $h \in [\underline{h}, \overline{h}]$, width $w \in [\underline{w}, \overline{w}]$ and shift $\xi \in [0, 2\pi]$. Let $M > 0$. There exists a constant $C = C(M, \mu) > 0$, depending only on $\mu$ and $M$, with the following property: If $\mathcal{N}(\bar{u}) = \sum_{k=1}^p \beta_k(\bar{u})\tau_k$ is any operator approximation with linear reconstruction dimension $p$, such that $\sup_{\bar{u}\sim\mu} \|\mathcal{N}(\bar{u})\|_{L^\infty} \leq M < \infty$, then

$$\mathbb{E}_{\bar{u}\sim\mu}[\|\mathcal{N}(\bar{u}) - \mathcal{G}_{\text{adv}}(u)\|_{L^1}] \geq \frac{C}{p}.$$

*Sketch of proof.* The argument is an almost exact repetition of the lower bound derived in Lanthaler et al. (2022), and therefore we will only outline the main steps of the argument, here: Since the measure $\mu$ is translation-invariant, it can be shown that the optimal PCA eigenbasis with respect to the $L^2(\mathbb{T})$-norm (cp. **SM** A) is the Fourier basis. Consider the (complex) Fourier basis $\{e^{ikx}\}_{k\in\mathbb{Z}}$, and denote the corresponding eigenvalues $\{\widetilde{\lambda}_k\}_{k\in\mathbb{Z}}$. The $k$-th eigenvalue $\widetilde{\lambda}_k$ of the covariance-operator $\Gamma_\mu = \mathbb{E}_{u\sim\mu}[(u \otimes u)]$ satisfies

$$\Gamma_\mu \left(e^{-ikx}\right) = \widetilde{\lambda}_k e^{-ikx}.$$

A short calculation, as in (Lanthaler et al., 2022, Proof of Lemma 4.14), then shows that

$$\widetilde{\lambda}_k = \int_{\underline{h}}^{\overline{h}} \int_{\underline{w}}^{\overline{w}} h^2 |\widehat{\psi}_w(k)|^2 \frac{dw}{\Delta w} \frac{dh}{\Delta h} \geq \underline{h}^2 \int_{\underline{w}}^{\overline{w}} |\widehat{\psi}_w(k)|^2 \frac{dw}{\Delta w},$$

where $\widehat{\psi}_w(k)$ denotes the $k$-th Fourier coefficient of $\psi_w(x) := 1_{[-w/2, w/2]}(x)$. Since $\psi_w(x)$ has a jump discontinuity of size 1 for any $w > 0$, it follows from basic Fourier analysis, that the asymptotic decay of $|\widehat{\psi}_w(k)| \sim C/|k|$, and hence, there exists a constant $C = C(\underline{h}, \underline{w}, \overline{w}) > 0$, such that

$$\widetilde{\lambda}_k \geq C|k|^{-2}.$$

Re-ordering these eigenvalues $\widetilde{\lambda}_k$ in descending order (and renaming), $\lambda_1 \geq \lambda_2 \geq \ldots$, it follows that for some constant $C = C(\underline{h}, \underline{w}, \overline{w}) > 0$, we have

$$\sum_{j>p} \lambda_j \geq C \sum_{j>p} j^{-2} \geq Cp^{-1}.$$

By Theorem 2.1, this implies that (independently of the choice of the functionals $\beta_k(\bar{u})$!), in the Hilbert space $L^2(\mathbb{T})$, we have

$$\mathbb{E}_{\bar{u}\sim\mu} \left[\|\mathcal{N}(\bar{u}) - \mathcal{G}(\bar{u})\|_{L^2}^2\right] \geq \sum_{j>p} \lambda_j \geq \frac{C}{p},$$

for a constant $C > 0$ that depends only on $\mu$, but is independent of $p$. To obtain a corresponding estimate with respect to the $L^1$-norm, we simply observe that the above lower bound on the $L^2$-norm together with the a priori bound $\sup_{\bar{u}\sim\mu} \|\mathcal{G}_{\text{adv}}(\bar{u})\|_{L^\infty} \leq \overline{h}$ on the underlying operator and the assumed $L^\infty$-bound $\sup_{\bar{u}\sim\mu} \|\mathcal{N}(\bar{u})\|_{L^\infty} \leq M$, imply

$$\frac{C}{p} \leq \mathbb{E}_{\bar{u}\sim\mu} \left[\|\mathcal{N}(\bar{u}) - \mathcal{G}(\bar{u})\|_{L^2}^2\right]$$

$$\leq \mathbb{E}_{\bar{u}\sim\mu} \left[\|\mathcal{N}(\bar{u}) - \mathcal{G}(\bar{u})\|_{L^\infty} \|\mathcal{N}(\bar{u}) - \mathcal{G}(\bar{u})\|_{L^1}\right]$$

$$\leq (M + \overline{h}) \mathbb{E}_{\bar{u}\sim\mu} \left[\|\mathcal{N}(\bar{u}) - \mathcal{G}(\bar{u})\|_{L^1}\right].$$

This immediately implies the claimed lower bound. $\square$

***Proposition* D.10** (Lower bound in $m$)**.** Consider the solution operator $\mathcal{G}_{\mathrm{adv}} : L^1(\mathbb{T}) \cap L^\infty(\mathbb{T}) \to L^1(\mathbb{T}) \cap L^\infty(\mathbb{T})$ of the linear advection equation, with input measure $\mu$ given as the law of random box functions of height $h \in [\underline{h}, \overline{h}]$, width $w \in [\underline{w}, \overline{w}]$ and shift $\xi \in [0, 2\pi]$. There exists an absolute constant $C > 0$ with the following property: If $\mathcal{N}^{\mathrm{DON}}$ is a DeepONet approximation with $m$ sensor points, then

$$\mathbb{E}_{\bar{u} \sim \mu} \left[ \| \mathcal{N}(\bar{u}) - \mathcal{G}_{\mathrm{adv}}(\bar{u}) \|_{L^1} \right] \geq \frac{C}{m}.$$

*Proof.* We recall that the initial data $\bar{u}$ is a randomly shifted box function, of the form

$$\bar{u}(x) = h 1_{[-w/2, +w/2]}(x - \xi),$$

where $\xi \in [0, 2\pi]$, $h \in [\underline{h}, \overline{h}]$ and $w \in [\underline{w}, \overline{w}]$ are independent, uniformly distributed random variables.

Let $x_1, \ldots, x_m \in (0, 2\pi]$ be an arbitrary choice of $m$ sensor points. In the following, we denote $\bar{u}(\boldsymbol{X}) := (\bar{u}(x_1), \ldots, \bar{u}(x_m)) \in \mathbb{R}^m$. Let now $(x, \bar{u}(\boldsymbol{X})) \mapsto \Phi(x; \bar{u}(\boldsymbol{X}))$ be *any* mapping, such that $x \mapsto \Phi(x; \bar{u}(\boldsymbol{X})) \in L^1(\mathbb{T})$ for all possible random choices of $\bar{u}$ (i.e. for all $\bar{u} \sim \mu$). Then we claim that

$$\mathbb{E}_{\bar{u} \sim \mu}[\| \mathcal{G}_{\mathrm{adv}}(\bar{u}) - \Phi(\,\cdot\,; \bar{u}(\boldsymbol{X})) \|_{L^1}] \geq \frac{C}{m}, \tag{D.1}$$

for a constant $C = C(\underline{h}, \overline{w}) > 0$, holds for all $m \in \mathbb{N}$. Clearly, the lower bound (D.1) immediately implies the statement of Proposition D.10, upon making the particular choice

$$\Phi(x; \bar{u}(\boldsymbol{X})) = \mathcal{N}(\bar{u})(x) \equiv \sum_{k=1}^p \beta_k(\bar{u}(x_1), \ldots, \bar{u}(x_m)) \tau_k(x).$$

To prove (D.1), we first recall that $\bar{u}(x) = \bar{u}(x; h, w, \xi)$ depends on three parameters $h$, $w$ and $\xi$, and the expectation over $\bar{u} \sim \mu$ in (D.1) amounts to averaging over $h \in [\underline{h}, \overline{h}]$, $w \in [\underline{w}, \overline{w}]$ and $\xi \in [0, 2\pi]$. In the following, we fix $w$ and $h$, and only consider the average over $\xi$. Suppressing the dependence on the fixed parameters, we will prove that

$$\frac{1}{2\pi} \int_0^{2\pi} \| \bar{u}(x; \xi) - \Phi(x; \bar{u}(\boldsymbol{X}; \xi)) \|_{L^1} \, d\xi \geq \frac{C}{m}, \tag{D.2}$$

with a constant that only depends on $\underline{h}, \overline{w}$. This clearly implies (D.1).

To prove (D.2), we first introduce two mappings $\xi \mapsto I(\xi)$ and $\xi \mapsto J(\xi)$, by

$$I(\xi) = i \Leftrightarrow \xi - \frac{w}{2} \in [x_i, x_{i+1}), \qquad J(\xi) = j \Leftrightarrow \xi + \frac{w}{2} \in [x_j, x_{j+1}),$$

where we make the natural identifications on the periodic torus (e.g. $x_{m+1}$ is identified with $x_1$ and $\xi \pm w/2$ is evaluated modulo $2\pi$). We observe that both mappings $\xi \mapsto I(\xi), J(\xi)$ cycle exactly once through the entire index set $\{1, \ldots, m\}$ as $\xi$ varies from $0$ to $2\pi$. Next, we introduce

$$A_{ij} := \{\xi \in [0, 2\pi] \, | \, I(\xi) = i, J(\xi) = j\}, \quad \forall i, j \in \{1, \ldots, m\}.$$

Clearly, each $\xi \in [0, 2\pi)$ belongs to only one of these sets $A_{ij}$. Since $\xi \mapsto I(\xi)$ and $\xi \mapsto J(\xi)$ have $m$ jumps on $[0, 2\pi)$, it follows that the mapping $\xi \mapsto (I(\xi), J(\xi))$ can have at most $2m$ jumps. In particular, this implies that there are at most $2m$ non-empty sets $A_{ij} \neq \emptyset$ (these are all sets of the form $A_{I(\xi), J(\xi)}, \xi \in [0, 2\pi)$), i.e.

$$\#\{A_{ij} \neq \emptyset \, | \, i, j \in \{1, \ldots, m\}\} \leq 2m. \tag{D.3}$$

Since $\bar{u}(x; \xi) = h \, 1_{[-w/2, w/2]}(x - \xi)$, one readily sees that when $\xi$ varies in the interior of $A_{ij}$, then all sensor point values $\bar{u}(\boldsymbol{X}; \xi)$ remain constant, i.e. the mapping

$$\mathrm{interior}(A_{ij}) \mapsto \mathbb{R}^m, \quad \xi \mapsto \bar{u}(\boldsymbol{X}; \xi) = \mathrm{const.}$$

We also note that $A_{ij} = [x_i + w/2, x_{i+1} + w/2) \cap [x_j - w/2, x_{j+1} - w/2)$ is in fact an interval. Fix $i, j$ such that $A_{ij} \neq \emptyset$ for the moment. We can write $A_{ij} = [a - \Delta, a + \Delta)$ for some $a, \Delta \in \mathbb{T}$,

and there exists a constant $\bar{U} \in \mathbb{R}^m$ such that $\bar{U} \equiv \bar{u}(\boldsymbol{X}; \xi)$ for $\xi \in [a - \Delta, a + \Delta)$. It follows from the triangle inequality that

$$
\int_{A_{ij}} \|\bar{u}(x; \xi) - \Phi(x; \bar{u}(\boldsymbol{X}; \xi))\|_{L^1} \, d\xi = \int_{a-\Delta}^{a+\Delta} \|\bar{u}(x; \xi) - \Phi(x; \bar{U})\|_{L^1} \, d\xi
$$

$$
= \int_0^\Delta \|\bar{u}(x; a - \xi') - \Phi(x; \bar{U})\|_{L^1} \, d\xi'
$$

$$
+ \int_0^\Delta \|\bar{u}(x; a + \xi') - \Phi(x; \bar{U})\|_{L^1} \, d\xi'
$$

$$
\geq \int_0^\Delta \|\bar{u}(x; a + \xi') - \bar{u}(x; a - \xi')\|_{L^1} \, d\xi'.
$$

Since $\bar{u}(x; \xi) = h \, 1_{[-w/2, w/2]}(x - \xi)$, we have, by a simple change of variables

$$
\|\bar{u}(x; a + \xi') - \bar{u}(x; a - \xi')\|_{L^1} = h \int_{\mathbb{T}} |1_{[-w/2, w/2]}(x) - 1_{[-w/2, w/2]}(x + 2\xi')| \, dx.
$$

The last expression is of order $\xi'$, provided that $\xi'$ is small enough to avoid overlap with a periodic shift (recall that we are on working on the torus, and $1_{[-w/2, w/2]}(x)$ is identified with its periodic extension). To avoid such issues related to periodicity, we first note that $\xi' \leq \Delta \leq \pi$, and then we choose a (large) constant $C_0 = C_0(\overline{w})$, such that for any $\xi' \leq \pi/C_0$ and $w \leq \overline{w}$, we have

$$
\int_{\mathbb{T}} |1_{[-w/2, w/2]}(x) - 1_{[-w/2, w/2]}(x + 2\xi')| \, dx = \int_{-w/2-2\xi'}^{-w/2} 1 \, dx + \int_{w/2-2\xi'}^{w/2} 1 \, dx = 4\xi'.
$$

From the above, we can now estimate

$$
\int_{A_{ij}} \|\bar{u}(x; \xi) - \Phi(x; \bar{u}(\boldsymbol{X}; \xi))\|_{L^1} \, d\xi \geq \int_0^\Delta \|\bar{u}(x; a + \xi') - \bar{u}(x; a - \xi')\|_{L^1} \, d\xi'
$$

$$
\geq \int_0^{\Delta/C_0} \|\bar{u}(x; a + \xi') - \bar{u}(x; a - \xi')\|_{L^1} \, d\xi'
$$

$$
\geq \underline{h} \int_0^{\Delta/C_0} 4\xi' \, d\xi'
$$

$$
= 2\underline{h} \frac{\Delta^2}{C_0^2} \geq C |A_{ij}|^2,
$$

where $C = C(\underline{h}, \overline{w})$ is a constant only depending on the fixed parameters $\underline{h}, \overline{w}$.

Summing over all $A_{ij} \neq \emptyset$, we obtain the lower bound

$$
\int_0^{2\pi} \|\bar{u}(x; \xi) - \Phi(x; \bar{u}(\boldsymbol{X}; \xi))\|_{L^1} \, d\xi = \sum_{A_{ij} \neq \emptyset} \int_{A_{ij}} \|\bar{u}(x; \xi) - \Phi(x; \bar{u}(\boldsymbol{X}; \xi))\|_{L^1} \, d\xi
$$

$$
\geq C \sum_{A_{ij} \neq \emptyset} |A_{ij}|^2.
$$

We observe that $[0, 2\pi) = \bigcup A_{ij}$ is a disjoint union, and hence $\sum_{A_{ij} \neq \emptyset} |A_{ij}| = 2\pi$. Furthermore, as observed above, there are at most $2m$ non-zero summands $|A_{ij}| \neq 0$. To finish the proof, we claim that the functional $\sum_{k=1}^{2m} |\alpha_k|^2$ is minimized among all $\alpha_1, \ldots, \alpha_{2m}$ satisfying the constraint $\sum_{k=1}^{2m} |\alpha_k| = 2\pi$ if, and only if, $|\alpha_1| = \cdots = |\alpha_{2m}| = \pi/m$. Given this fact, it then immediately follows from the above estimate that

$$
\frac{1}{2\pi} \int_0^{2\pi} \|\bar{u}(x; \xi) - \Phi(x; \bar{u}(\boldsymbol{X}; \xi))\|_{L^1} \, d\xi \geq C \sum_{A_{ij} \neq \emptyset} |A_{ij}|^2 \geq \frac{2C\pi^2}{m}.
$$

where $C = C(\underline{h}, \overline{w}) > 0$ is independent of the values of $w \in [\underline{w}, \overline{w}]$ and $h \in [\underline{h}, \overline{h}]$. This suffices to conclude the claim of Proposition D.10.

It remains to prove the claim: We argue by contradiction. Let $\alpha_1, \ldots, \alpha_{2m}$ be a minimizer of $\sum_k |\alpha_k|^2$ under the constraint $\sum_k |\alpha_k| = 2\pi$. Clearly, we can wlog assume that $0 \leq \alpha_1 \leq \cdots \leq \alpha_{2m}$ are non-negative numbers. If the claim does not hold, then there exists a minimizer, such that $\alpha_1 < \alpha_{2m}$. Given $\delta > 0$ to be determined below, we define $\beta_k$ by

$$\beta_1 = \alpha_1 + \delta, \quad \beta_{2m} = \alpha_{2m} - \delta,$$

and $\beta_k = \alpha_k$, for all other indices. Then, by a simple computation, we observe that

$$\sum_k \alpha_k^2 - \sum_k \beta_k^2 = 2\delta(\alpha_{2m} - \alpha_1 - \delta).$$

Choosing $\delta > 0$ sufficiently small, we can ensure that the last quantity is $> 0$, while keeping $\beta_k \geq 0$ for all $k$. In particular, it follows that $\sum_k |\beta_k| = \sum_k |\alpha_k| = 2\pi$, but

$$\sum_k \alpha_k^2 > \sum_k \beta_k^2,$$

in contradiction to the assumption that $\alpha_1, \ldots, \alpha_{2m}$ minimize the last expression. Hence, any minimizer must satisfy $|\alpha_1| = \cdots = |\alpha_{2m}| = \pi/m$. $\qquad\square$

### D.3 PROOF OF THEOREM 3.2

*Proof.* We choose equidistant grid points $x_1, \ldots, x_m$ for the construction of a shift-DeepONet approximation to $\mathcal{G}_{\mathrm{adv}}$. We may wlog assume that the grid distance $\Delta x = x_2 - x_1 < \underline{w}$, as the statement is asymptotic in $m \to \infty$. We note the following points:

**Step 1:** We show that $h$ can be efficiently determined by max-pooling. First, we observe that for any two numbers $a, b$, the mapping

$$\begin{pmatrix} a \\ b \end{pmatrix} \mapsto \begin{pmatrix} \max(0, a-b) \\ \max(0, b) \\ \max(0, -b) \end{pmatrix} \mapsto \max(0, a-b) + \max(0, b) - \max(0, -b) \equiv \max(a, b),$$

is exactly represented by a ReLU neural network $\widetilde{\max}(a, b)$ of width 3, with a single hidden layer. Given $k$ inputs $a_1, \ldots, a_k$, we can parallelize $O(k/2)$ copies of $\widetilde{\max}$, to obtain a ReLU network of width $\leq 3k$ and with a single hidden layer, which maps

$$\begin{pmatrix} a_1 \\ a_2 \\ \vdots \\ a_{k-1} \\ a_k \end{pmatrix} \mapsto \begin{pmatrix} \max(a_1, a_2) \\ \vdots \\ \max(a_{k-1}, a_k) \end{pmatrix}.$$

Concatenation of $O(\log_2(k))$ such ReLU layers with decreasing input sizes $k, \lceil k/2 \rceil, \lceil k/4 \rceil, \ldots, 1$, provides a ReLU representation of max-pooling

$$\begin{pmatrix} a_1 \\ \vdots \\ a_k \end{pmatrix} \mapsto \begin{pmatrix} \max(a_1, a_2) \\ \vdots \\ \max(a_{k-1}, a_k) \end{pmatrix} \mapsto \begin{pmatrix} \max(a_1, a_2, a_3, a_4) \\ \vdots \\ \max(a_{k-3}, a_{k-2}, a_{k-1}, a_k) \end{pmatrix} \mapsto \cdots \mapsto \max(a_1, \ldots, a_k).$$

This concatenated ReLU network maxpool : $\mathbb{R}^k \to \mathbb{R}$ has width $\leq 3k$, depth $O(\log(k))$, and size $O(k \log(k))$.

Our goal is to apply the above network maxpool to the shift-DeepONet input $u(x_1), \ldots, u(x_m)$ to determine the height $h$. To this end, we first choose $\ell_1, \ldots, \ell_k \in \{1, \ldots, m\}$, such that $x_{\ell_{j+1}} - x_{\ell_j} \leq \underline{w}$, with $k \in \mathbb{N}$ minimal. Note that $k$ is uniformly bounded, with a bound that only depends on $\underline{w}$ (not on $m$). Applying the maxpool construction above the $u(x_{\ell_1}), \ldots, u(x_{\ell_k})$, we obtain a mapping

$$\begin{pmatrix} u(x_1) \\ \vdots \\ u(x_m) \end{pmatrix} \mapsto \begin{pmatrix} u(x_{\ell_1}) \\ \vdots \\ u(x_{\ell_k}) \end{pmatrix} \mapsto \mathrm{maxpool}(u(x_{\ell_1}), \ldots, u(x_{\ell_k})) = h.$$

This mapping can be represented by $O(\log(k))$ ReLU layers, with width $\leq 3k$ and total (fully connected) size $O(k^2 \log(k))$. In particular, since $k$ only depends on $\underline{w}$, we conclude that there exists $C = C(\underline{w}) > 0$ and a neural network $\widetilde{h} : \mathbb{R}^m \to \mathbb{R}$ with

$$\text{depth}(\widetilde{h}) \leq C, \quad \text{width}(\widetilde{h}) \leq C, \quad \text{size}(\widetilde{h}) \leq C, \tag{D.4}$$

such that

$$\widetilde{h}(\bar{u}(\boldsymbol{X})) = h, \tag{D.5}$$

for any initial data of the form $\bar{u}(x) = h 1_{[-w/2, w/2]}(x - \xi)$, where $h \in [\underline{h}, \overline{h}]$, $w \in [\underline{w}, \overline{w}]$, and $\xi \in [0, 2\pi]$.

**Step 2:** To determine the width $w$, we can consider a linear layer (of size $m$), followed by an approximation of division, $\widetilde{\div}(a; b) \approx a/b$ (cp. Proposition D.7):

$$\bar{u}(\boldsymbol{X}) \mapsto \Delta x \sum_{j=1}^{m} \bar{u}(x_j) \mapsto \widetilde{\div}\left( \Delta x \sum_{j=1}^{m} \bar{u}(x_j); \widetilde{h}(\bar{u}(\boldsymbol{X})) \right)$$

Denote this by $\widetilde{w}(\bar{u}(\boldsymbol{X}))$. Then

$$|w - \widetilde{w}| = \left| \frac{1}{h} \int_0^{2\pi} \bar{u}(x)\, dx - \frac{\Delta x}{h} \sum_j \bar{u}(x_j) \right|$$

$$+ \left| \frac{\Delta x}{h} \sum_j \bar{u}(x_j) - \widetilde{\div}_{\underline{h}, \overline{h}, \epsilon}\left( \Delta x \sum_{j=1}^{m} \bar{u}(x_j); h \right) \right|$$

$$\leq \frac{2\pi}{m} + \epsilon.$$

And we have $\text{depth}(\widetilde{w}) \leq C \log(\epsilon^{-1})^2$, $\text{width}(\widetilde{w}) \leq C$, $\text{size}(\widetilde{w}) \leq C\left(m + \log(\epsilon^{-1})^2\right)$, by the complexity estimate of Proposition D.7.

**Step 3:** To determine the shift $\xi \in [0, 2\pi]$, we note that

$$\Delta x \sum_{j=1}^{m} \bar{u}(x_j) e^{-i x_j} = \int_0^{2\pi} \bar{u}(x) e^{-ix}\, dx + O\left( \frac{1}{m} \right)$$

$$= 2 \sin(w/2) e^{-i\xi} + O\left( \frac{1}{m} \right).$$

Using the result of Lemma D.8, combined with the approximation of division of Proposition D.7, and the observation that $w \in (\underline{w}, \pi)$ implies that $\sin(w/2) \geq \sin(\underline{w}/2) > 0$ is uniformly bounded from below for all $w \in [\underline{w}, \overline{w}]$, it follows that for all $\epsilon \in (0, \frac{1}{2}]$, there exists a neural network $\widetilde{\xi} : \mathbb{R}^m \to [0, 2\pi]$, of the form

$$\widetilde{\xi}(\bar{u}(\boldsymbol{X})) = \Xi_\epsilon\left[ \widetilde{\div}\left( \Delta x \sum_{j=1}^{m} u(x_j) e^{-i x_j}; \widetilde{\times}_{M, \epsilon}\left( \widetilde{h}, 2\widetilde{\sin}(\widetilde{w}/2) \right) \right) \right]$$

such that

$$\left| \xi - \widetilde{\xi}(\bar{u}(\boldsymbol{X})) \right| \leq \epsilon,$$

for all $\xi \in [0, 2\pi - \epsilon)$, and

$$\text{depth}(\widetilde{\xi}) \leq C \log(\epsilon^{-1})^2, \quad \text{width}(\widetilde{\xi}) \leq C, \quad \text{size}(\widetilde{\xi}) \leq C\left(m + \log(\epsilon^{-1})^2\right).$$

**Step 4:** Combining the above three ingredients (Steps 1–3), and given the fixed advection velocity $a \in \mathbb{R}$ and fixed time $t$, we define a shift-DeepONet with $p = 6$, scale-net $\mathcal{A}_k \equiv 1$, and shift-net $\gamma$

with output $\gamma_k(\bar{u}) \equiv \widetilde{\xi} + at$, as follows:

$$\mathcal{N}^{\text{sDON}}(\bar{u}) = \sum_{k=1}^{p} \beta_k(\bar{u})\tau_k(x - \gamma_k(\bar{u}))$$

$$:= \sum_{j=-1}^{1} \widetilde{h}\,\widetilde{1}_{[0,\infty)}^{\epsilon}\left(x - \widetilde{\xi} - at + \widetilde{w}/2 + 2\pi j\right)$$

$$- \sum_{j=-1}^{1} \widetilde{h}\,\widetilde{1}_{[0,\infty)}^{\epsilon}\left(x - \widetilde{\xi} - at - \widetilde{w}/2 + 2\pi j\right),$$

where $\widetilde{h} = \widetilde{h}(\bar{u}(\boldsymbol{X}))$, $\widetilde{w} = \widetilde{w}(\bar{u}(\boldsymbol{X}))$, and $\widetilde{\xi} = \widetilde{\xi}(\bar{u}(\boldsymbol{X}))$, and where $\widetilde{1}_{[0,\infty)}^{\epsilon}$ is a sufficiently accurate $L^1$-approximation of the indicator function $1_{[0,\infty)}(x)$ (cp. Proposition D.2). To estimate the approximation error, we denote $\widetilde{1}_{[-\widetilde{w}/2,\widetilde{w}/2]}^{\epsilon}(x) := \widetilde{1}_{[0,\infty)}^{\epsilon}(x + \widetilde{w}/2) - \widetilde{1}_{[0,\infty)}^{\epsilon}(x - \widetilde{w}/2)$ and identify it with it's periodic extension to $\mathbb{T}$, so that we can more simply write

$$\mathcal{N}^{\text{sDON}}(\bar{u})(x) = \widetilde{h}\,\widetilde{1}_{[-\widetilde{w}/2,\widetilde{w}/2]}^{\epsilon}(x - \widetilde{\xi} - at).$$

We also recall that the solution $u(x,t)$ of the linear advection equation $\partial_t u + a\partial_x u = 0$, with initial data $u(x,0) = \bar{u}(x)$ is given by $u(x,t) = \bar{u}(x - at)$, where $at$ is a fixed constant, independent of the input $\bar{u}$. Thus, we have

$$\mathcal{G}_{\text{adv}}(\bar{u})(x) = \bar{u}(x - at) = h\,1_{[-w/2,w/2]}(x - \xi - at).$$

We can now write

$$|\mathcal{G}_{\text{adv}}(\bar{u})(x) - \mathcal{N}^{\text{sDON}}(\bar{u})(x)| = \left|h\,1_{[-w/2,w/2]}(x - \xi - at) - \widetilde{h}\,\widetilde{1}_{[-\widetilde{w}/2,\widetilde{w}/2]}^{\epsilon}(x - \widetilde{\xi} - at)\right|.$$

We next recall that by the construction of Step 1, we have $\widetilde{h}(\bar{u}) \equiv h$ for all inputs $\bar{u}$. Furthermore, upon integration over $x$, we can clearly get rid of the constant shift $at$ by a change of variables. Hence, we can estimate

$$\|\mathcal{G}_{\text{adv}}(\bar{u}) - \mathcal{N}^{\text{sDON}}(\bar{u})\|_{L^1} \leq \overline{h}\int_{\mathbb{T}}\left|1_{[-w/2,w/2]}(x - \xi) - \widetilde{1}_{[-\widetilde{w}/2,\widetilde{w}/2]}^{\epsilon}(x - \widetilde{\xi})\right|dx. \tag{D.6}$$

Using the straight-forward bound

$$\left|1_{[-w/2,w/2]}(x - \xi) - \widetilde{1}_{[-\widetilde{w}/2,\widetilde{w}/2]}^{\epsilon}(x - \widetilde{\xi})\right| \leq \left|1_{[-w/2,w/2]}(x - \xi) - 1_{[-w/2,w/2]}(x - \widetilde{\xi})\right|$$

$$+ \left|1_{[-w/2,w/2]}(x - \widetilde{\xi}) - 1_{[-\widetilde{w}/2,\widetilde{w}/2]}(x - \widetilde{\xi})\right|$$

$$+ \left|1_{[-\widetilde{w}/2,\widetilde{w}/2]}(x - \widetilde{\xi}) - \widetilde{1}_{[-\widetilde{w}/2,\widetilde{w}/2]}^{\epsilon}(x - \widetilde{\xi})\right|,$$

one readily checks that, by Step 3, the integral over the first term is bounded by

$$\|(I)\|_{L^1} \leq C\int_0^{2\pi-\epsilon}|\xi - \widetilde{\xi}|\,d\xi + \int_{2\pi-\epsilon}^{2\pi} 2\,d\xi \leq (C+2)\epsilon.$$

where $C = C(\underline{w}, \overline{w}) > 0$. By Step 2, the integral over the second term can be bounded by

$$\|(II)\|_{L^1} \leq C|w - \widetilde{w}| \leq C\left(1/m + \epsilon\right).$$

Finally, by Proposition D.1, by choosing $\epsilon$ sufficiently small (recall also that the size of $\widetilde{1}^{\epsilon}$ is independent of $\epsilon$), we can ensure that

$$\|(III)\|_{L^1} = \|\widetilde{1}_{[-\widetilde{w}/2,\widetilde{w}/2]}^{\epsilon} - 1_{[-\widetilde{w}/2,\widetilde{w}/2]}\|_{L^1} \leq \epsilon,$$

holds uniformly for any $\widetilde{w}$. Hence, the right-hand side of (D.6) obeys an upper bound of the form

$$\mathbb{E}_{\bar{u}\sim\mu}\left[\|\mathcal{G}_{\text{adv}}(\bar{u}) - \mathcal{N}^{\text{sDON}}(\bar{u})\|_{L^1}\right] = \fint_{\underline{h}}^{\overline{h}}dh\fint_{\underline{w}}^{\overline{w}}dw\fint_{\mathbb{T}}d\xi\,\|\mathcal{G}_{\text{adv}}(\bar{u}) - \mathcal{N}^{\text{sDON}}(\bar{u})\|_{L^1}$$

$$\leq \frac{\overline{h}}{2\pi}\fint_{\underline{w}}^{\overline{w}}dw\int_0^{2\pi}d\xi\,\{\|(I)\|_{L^1} + \|(II)\|_{L^1} + \|(III)\|_{L^1}\}$$

$$\leq C\left(\epsilon + \frac{1}{m}\right),$$

for a constant $C = C(\underline{w}, \overline{w}, \overline{h}) > 0$. We also recall that by our construction,

$$\mathrm{depth}(\mathcal{N}^{\mathrm{sDON}}) \le C\log(\epsilon^{-1})^2, \quad \mathrm{width}(\mathcal{N}^{\mathrm{sDON}}) \le C, \quad \mathrm{size}(\mathcal{N}^{\mathrm{sDON}}) \le C\left(m + \log(\epsilon^{-1})^2\right).$$

Replacing $\epsilon$ by $\epsilon/2C$ and choosing $m \sim \epsilon^{-1}$, we obtain

$$\mathbb{E}_{\bar{u} \sim \mu}\left[\|\mathcal{G}_{\mathrm{adv}}(\bar{u}) - \mathcal{N}^{\mathrm{sDON}}(\bar{u})\|_{L^1}\right] \le \epsilon,$$

with

$$\mathrm{depth}(\mathcal{N}^{\mathrm{sDON}}) \le C\log(\epsilon^{-1})^2, \quad \mathrm{width}(\mathcal{N}^{\mathrm{sDON}}) \le C, \quad \mathrm{size}(\mathcal{N}^{\mathrm{sDON}}) \le C\epsilon^{-1},$$

where $C$ depends only on $\mu$, and is independent of $\epsilon$. This implies the claim of Theorem 3.2. $\qquad\square$

### D.4 PROOF OF THEOREM 3.3

For the proof of Theorem 3.3, we will need a few intermediate results:

**Lemma D.11.** Let $\bar{u} = h\,1_{[-w/2,w/2]}(x - \xi)$ and fix a constant $at \in \mathbb{R}$. There exists a constant $C > 0$, such that given $N$ grid points, there exists an FNO with

$$k_{\max} = 1, \quad d_v \le C, \quad \mathrm{depth} \le C, \quad \mathrm{size} \le C,$$

such that

$$\sup_{h,w,\xi} \left|\mathcal{N}^{\mathrm{FNO}}(\bar{u})(x) - \sin(w/2)\cos(x - \xi - at)\right| \le \frac{C}{N}.$$

*Proof.* We first note that there is a ReLU neural network $\Phi$ consisting of two hidden layers, such that

$$\Phi(\bar{u}(x)) = \min(1, \underline{h}^{-1}\bar{u}(x)) = \min\left(1, \underline{h}^{-1}h1_{[-w/2,w/2]}(x)\right) = 1_{[-w/2,w/2]}(x),$$

for all $h \in [\underline{h}, \overline{h}]$. Clearly, $\Phi$ can be represented by FNO layers where the convolution operator $K_\ell \equiv 0$.

Next, we note that the $k = 1$ Fourier coefficient of $\widetilde{u} := 1_{[-w/2,w/2]}(x - \xi)$ is given by

$$\begin{aligned}
\mathcal{F}_N\widetilde{u}(k = \pm 1) &= \frac{1}{N}\sum_{j=1}^{N} 1_{[-w/2,w/2]}(x_j - \xi)e^{\mp ix_j} \\
&= \frac{1}{2\pi}\int_0^{2\pi} 1_{[-w/2,w/2]}(x - \xi)e^{\mp ix}\,dx + O(N^{-1}) \\
&= \frac{\sin(w/2)e^{\mp i\xi}}{\pi} + O(N^{-1}),
\end{aligned}$$

where the $O(N^{-1})$ error is bounded uniformly in $\xi \in [0, 2\pi]$ and $w \in [\underline{w}, \overline{w}]$. It follows that the FNO $\mathcal{N}^{\mathrm{FNO}}$ defined by

$$\bar{u} \mapsto \Phi(\bar{u}) \mapsto \sigma\left(\mathcal{F}_N^{-1}P\mathcal{F}_N\Phi(\bar{u})\right) - \sigma\left(\mathcal{F}_N^{-1}(-P)\mathcal{F}_N\Phi(\bar{u})\right),$$

where $P$ implements a projection onto modes $|k| = 1$ and multiplication by $e^{\pm iat}\pi/2$ (the complex exponential introduces a phase-shift by $at$), satisfies

$$\sup_{x \in \mathbb{T}} \left|\mathcal{N}^{\mathrm{FNO}}(\bar{u})(x) - \sin(w/2)\cos(x - \xi - at)\right| \le \frac{C}{N},$$

where $C$ is independent of $N$, $w$, $h$ and $\xi$. $\qquad\square$

**Lemma D.12.** Fix $0 < \underline{h} < \overline{h}$ and $0 < \underline{w} < \overline{w}$. There exists a constant $C = C(\underline{h}, \overline{h}, \underline{w}, \overline{w}) > 0$ with the following property: For any input function $\bar{u}(x) = h\,1_{[-w/2,w/2]}(x - \xi)$ with $h \in [\underline{h}, \overline{h}]$ and $w \in [\underline{w}, \overline{w}]$, and given $N$ grid points, there exists an FNO with constant output function, such that

$$\sup_{h,w,\xi} \left|\mathcal{N}^{\mathrm{FNO}}(\bar{u})(x) - w\right| \le \frac{C}{N},$$

and with uniformly bounded size,

$$k_{\max} = 0, \quad d_v \le C, \quad \mathrm{depth} \le C, \quad \mathrm{size} \le C.$$

*Proof.* We can define an FNO mapping

$$\bar{u} \mapsto 1_{[-w/2,w/2]}(x) \mapsto \frac{2\pi}{N} \sum_{j=1}^{N} 1_{[-w/2,w/2]}(x_j) = w + O(N^{-1}),$$

where we observe that the first mapping is just $\bar{u} \mapsto \max(\underline{h}^{-1}\bar{u}(x), 1)$, which is easily represented by an ordinary ReLU NN of bounded size. The second mapping above is just projection onto the 0-th Fourier mode under the discrete Fourier transform. In particular, both of these mappings can be represented exactly by an FNO with $k_{\max} = 0$ and uniformly bounded $d_v$, depth and size. To conclude the argument, we observe that the error $O(N^{-1})$ depends only on the grid size and is independent of $w \in [\underline{w}, \overline{w}]$. $\qquad\square$

***Lemma* D.13.** Fix $0 < \underline{w} < \overline{w}$. There exists a constant $C = C(\underline{w}, \overline{w}) > 0$, such that for any $\epsilon > 0$, there exists an FNO such that for any constant input function $\bar{u}(x) \equiv w \in [\underline{w}, \overline{w}]$, we have

$$\left| \mathcal{N}^{\mathrm{FNO}}(\bar{u})(x) - \frac{1}{2}\sin(w) \right| \leq \epsilon, \quad \forall x \in [0, 2\pi],$$

and

$$k_{\max} = 0, \quad d_v \leq C, \quad \mathrm{depth} \leq C\log(\epsilon^{-1})^2, \quad \mathrm{size} \leq C\log(\epsilon^{-1})^2.$$

*Proof.* It follows e.g. from (Elbrächter et al., 2021, Thm. III.9) (or also Theorem D.6 above) that there exists a constant $C = C(\underline{w}, \overline{w}) > 0$, such that for any $\epsilon > 0$, there exists a ReLU neural network $S_\epsilon$ with $\mathrm{size}(S_\epsilon) \leq C\log(\epsilon^{-1})^2$, $\mathrm{depth}(S_\epsilon) \leq C\log(\epsilon^{-1})^2$ and $\mathrm{width}(S_\epsilon) \leq C$, such that

$$\sup_{w \in [\underline{w}, \overline{w}]} \left| S_\epsilon(w) - \frac{1}{2}\sin(w) \right| \leq \epsilon.$$

To finish the proof, we simply note that this ReLU neural network $S_\epsilon$ can be easily represented by an FNO $\mathcal{S}_\epsilon$ with $k_{\max} = 0$, $d_v \leq C$, $\mathrm{depth}(S_\epsilon) \leq C\log(\epsilon^{-1})^2$ and $\mathrm{size}(\mathcal{S}_\epsilon) \leq C\log(\epsilon^{-1})^2$; it suffices to copy the weight matrices $W_\ell$ of $S_\epsilon$, set the entries of the Fourier multiplier matrices $P_\ell(k) \equiv 0$, and choose constant bias functions $b_\ell(x) = \mathrm{const.}$ (with values given by the corresponding biases in the hidden layers of $S_\epsilon$). $\qquad\square$

***Lemma* D.14.** Let $\bar{u} = h\,1_{[-w/2,w/2]}(x - \xi)$. Assume that $2\pi/N \leq \underline{w}$. For any $\epsilon > 0$, there exists an FNO with constant output function, such that

$$\sup_{h,w,\xi} \left| \mathcal{N}^{\mathrm{FNO}}(\bar{u})(x) - h \right| \leq \epsilon,$$

and

$$k_{\max} = 0, \quad d_v \leq C, \quad \mathrm{depth} \leq C\log(\epsilon^{-1})^2.$$

*Proof.* The proof follows along similar lines as the proofs of the previous lemmas. In this case, we can define an FNO mapping

$$\bar{u} \mapsto \begin{bmatrix} h\,1_{[-w/2,w/2]}(x) \\ 1_{[-w/2,w/2]}(x) \end{bmatrix} \mapsto \begin{bmatrix} h\sum_{j=1}^{N} 1_{[-w/2,w/2]}(x_j) \\ \sum_{j=1}^{N} 1_{[-w/2,w/2]}(x_j) \end{bmatrix} \mapsto \widetilde{\div}_\epsilon \left( h\sum_{j=1}^{N} 1_{[-w/2,w/2]}(x_j), \sum_{j=1}^{N} 1_{[-w/2,w/2]}(x_j) \right).$$

The estimate on $k_{\max}$, $d_v$, depth follow from the construction of $\widetilde{\div}$ in Proposition D.7. $\qquad\square$

*Proof of Theorem 3.3.* We first note that (the $2\pi$-periodization of) $1_{[-w/2,w/2]}(x - \xi - at)$ is $= 1$ if, and only if

$$\cos(x - \xi - at) \geq \cos(w/2) \iff \sin(w/2)\cos(x - \xi - at) \geq \frac{1}{2}\sin(w). \tag{D.7}$$

The strategy of proof is as follows: Given the input function $\bar{u}(x) = h\,1_{[-w/2,w/2]}(x - \xi)$ with unknown $w \in [\underline{w}, \overline{w}]$, $\xi \in [0, 2\pi]$ and $h \in [\underline{h}, \overline{h}]$, and for given $a, t \in \mathbb{R}$ (these are fixed for this problem), we first construct an FNO which approximates the sequence of mappings

$$\bar{u} \mapsto \begin{bmatrix} h \\ w \\ \sin(w/2)\cos(x - \xi) \end{bmatrix} \mapsto \begin{bmatrix} h \\ \frac{1}{2}\sin(w) \\ \sin(w/2)\cos(x - \xi - at) \end{bmatrix} \mapsto \begin{bmatrix} h \\ \sin(w/2)\cos(x - \xi - at) - \frac{1}{2}\sin(w) \end{bmatrix}.$$

Then, according to (D.7), we can approximately reconstruct $1_{[-w/2,w/2]}(x-\xi-at)$ by approximating the identity

$$1_{[-w/2,w/2]}(x-\xi-at) = 1_{[0,\infty)}\left(\sin(w/2)\cos(x-\xi-at) - \frac{1}{2}\sin(w)\right),$$

where $1_{[0,\infty)}$ is the indicator function of $[0,\infty)$. Finally, we obtain $\mathcal{G}_{\mathrm{adv}}(\bar{u}) = h\,1_{[-w/2,w/2]}(x-\xi-at)$ by approximately multiplying this output by $h$. We fill in the details of this construction below.

**Step 1:** The first step is to construct approximations of the mappings above. We note that we can choose a (common) constant $C_0 = C_0(\underline{h}, \overline{h}, \underline{w}, \overline{w}) > 0$, depending only on the parameters $\underline{h}, \overline{h}, \underline{w}$ and $\overline{w}$, such that for any grid size $N \in \mathbb{N}$ all of the following hold:

1. There exists an FNO $\mathcal{H}_N$ with constant output (cp. Lemma D.14), such that for $\bar{u}(x) = h\,1_{[-w/2,w/2]}(x-\xi)$,

$$\sup_{w,h}|\mathcal{H}_N(\bar{u}) - h| \le \frac{1}{N}. \tag{D.8}$$

   and with

$$k_{\max} \le 1, \quad d_v \le C_0, \quad \mathrm{depth} \le C_0\log(N)^2, \quad \mathrm{size} \le C_0\log(N)^2.$$

2. Combining Lemma D.12 and D.13, we conclude that there exists an FNO $\mathcal{S}_N$ with constant output, such that for $\bar{u}(x) = h\,1_{[-w/2,w/2]}(x-\xi)$, we have

$$\sup_{w\in[\underline{w}-1,\overline{w}+1]}\left|\mathcal{S}_N(\bar{u}) - \frac{1}{2}\sin(w)\right| \le \frac{C_0}{N}. \tag{D.9}$$

   and with

$$k_{\max} = 0, \quad d_v \le C_0, \quad \mathrm{depth} \le C_0\log(N)^2, \quad \mathrm{size} \le C_0\log(N)^2.$$

3. There exists an FNO $\mathcal{C}_N$ (cp. Lemma D.11), such that for $\bar{u} = h\,1_{[-w/2,w/2]}$,

$$\sup_{x,\xi,w}|\mathcal{C}_N(\bar{u})(x) - \sin(w/2)\cos(x-\xi-at)| \le \frac{C_0}{N}, \tag{D.10}$$

   where the supremum is over $x,\xi \in [0,2\pi]$ and $w \in [\underline{w},\overline{w}]$, and such that

$$k_{\max} = 1, \quad d_v \le C_0, \quad \mathrm{depth} \le C_0, \quad \mathrm{size} \le C_0.$$

4. There exists a ReLU neural network $\widetilde{1}^N_{[0,\infty)}$ (cp. Proposition D.1), such that

$$\|\widetilde{1}^N_{[0,\infty)}\|_{L^\infty} \le 1, \quad \widetilde{1}^N_{[0,\infty)}(z) = \begin{cases} 0, & (x < 0), \\ 1, & (x \ge \frac{1}{N}). \end{cases} \tag{D.11}$$

   with

$$\mathrm{width}(\widetilde{1}^N_{[0,\infty)}) \le C_0, \quad \mathrm{depth}(\widetilde{1}^N_{[0,\infty)}) \le C_0.$$

5. there exists a ReLU neural network $\widetilde{\times}_N$ (cp. Proposition D.4), such that

$$\sup_{a,b}|\widetilde{\times}_N(a,b) - ab| \le \frac{1}{N}, \tag{D.12}$$

   where the supremum is over all $|a|, |b| \le \overline{h}+1$, and

$$\mathrm{width}(\widetilde{\times}_N) \le C_0, \quad \mathrm{depth}(\widetilde{\times}_N) \le C_0\log(N).$$

Based on the above FNO constructions, we define

$$\mathcal{N}^{\mathrm{FNO}}(\bar{u}) := \widetilde{\times}_N\left(\mathcal{H}_N(\bar{u}), \widetilde{1}^N_{[0,\infty)}\left(\mathcal{C}_N(\bar{u}) - \mathcal{S}_N(\bar{u})\right)\right). \tag{D.13}$$

Taking into account the size estimates from points 1–5 above, as well as the general FNO size estimate (B.2), it follows that $\mathcal{N}^{\mathrm{FNO}}$ can be represented by an FNO with

$$k_{\max} = 1, \quad d_v \leq C, \quad \text{depth} \leq C \log(N)^2, \quad \text{size} \leq C \log(N)^2. \tag{D.14}$$

To finish the proof of Theorem 3.3, it suffices to show that $\mathcal{N}^{\mathrm{FNO}}$ satisfies an estimate

$$\sup_{\bar{u} \sim \mu} \|\mathcal{N}^{\mathrm{FNO}}(\bar{u}) - \mathcal{G}_{\mathrm{adv}}(\bar{u})\|_{L^2} \leq \frac{C}{N},$$

with $C > 0$ independent of $N$.

**Step 2:** We claim that if $x \in [0, 2\pi]$ is such that

$$\left| \sin(w/2) \cos(x - \xi) - \frac{1}{2} \sin(w) \right| \geq \frac{2C_0 + 1}{N},$$

with $C_0$ the constant of Step 1, then

$$\widetilde{1}_{[0,\infty)}^N (\mathcal{C}_N(\bar{u})(x) - \mathcal{S}_N(\bar{u})) = 1_{[-w/2, w/2]}(x - \xi).$$

To see this, we first assume that

$$\sin(w/2) \cos(x - \xi) - \frac{1}{2} \sin(w) \geq \frac{2C_0 + 1}{N} > 0.$$

Then

$$\mathcal{C}_N(\bar{u})(x) - \mathcal{S}_N(\bar{u}) \geq \sin(w/2) \cos(x - \xi) - \frac{1}{2} \sin(w)$$

$$- |\mathcal{C}_N(\bar{u})(x) - \sin(w/2) \cos(x - \xi)| - \left| \mathcal{S}_N(\bar{u}) - \frac{1}{2} \sin(w) \right|$$

$$\geq \frac{2C_0 + 1}{N} - \frac{C_0}{N} - \frac{C_0}{N} = \frac{1}{N} > 0.$$

Hence, it follows from (D.11) that

$$\widetilde{1}_{[0,\infty)}^N (\mathcal{C}_N(\bar{u})(x) - \mathcal{S}_N(\bar{u})) = 1$$

$$= 1_{[0,\infty)} \left( \sin(w/2) \cos(x - \xi) - \frac{1}{2} \sin(w) \right)$$

$$= 1_{[-w/2, w/2]}(x).$$

The other case,

$$\sin(w/2) \cos(x - \xi) - \frac{1}{2} \sin(w) \leq -\frac{2C_0 + 1}{N},$$

is shown similarly.

**Step 3:** We note that there exists $C = C(\underline{w}, \overline{w}) > 0$, such that for any $\delta > 0$, the Lebesgue measure

$$\mathrm{meas}\left\{ x \in [0, 2\pi] \,\middle|\, \left| \sin(w/2) \cos(x - \xi) - \frac{1}{2} \sin(w) \right| < \delta \right\} \leq C\delta.$$

**Step 4:** Given the previous steps, we now write

$$\mathcal{G}_{\mathrm{adv}}(\bar{u}) - \mathcal{N}^{\mathrm{FNO}}(\bar{u}) = h \, 1_{[-w/2, w/2]}(x) - \widetilde{\times}_N \left( \mathcal{H}_N, \widetilde{1}_{[0,\infty)}^N \left( \mathcal{C}_N - \mathcal{S}_N(\bar{u}) \right) \right)$$

$$= \left[ h \, 1_{[-w/2, w/2]}(x) - h\widetilde{1}_{[0,\infty)}^N (\mathcal{C}_N - \mathcal{S}_N) \right]$$

$$+ (h - \mathcal{H}_N)\widetilde{1}_{[0,\infty)}^N (\mathcal{C}_N - \mathcal{S}_N)$$

$$+ \mathcal{H}_N \widetilde{1}_{[0,\infty)}^N \left( \mathcal{C}_N - \mathcal{S}_N(\bar{u}) \right) - \widetilde{\times}_N \left( \mathcal{H}_N, \widetilde{1}_{[0,\infty)}^N \left( \mathcal{C}_N - \mathcal{S}_N(\bar{u}) \right) \right)$$

$$=: (I) + (II) + (III).$$

The second $(II)$ and third $(III)$ terms are uniformly bounded by $N^{-1}$, by the construction of $\widetilde{\times}_N$ and $\mathcal{H}_N$. By Steps 2 and 3 (with $\delta = (2C_0 + 1)/N$), we can estimate the $L^1$-norm of the first term as

$$\|(I)\|_{L^1} \leq 2\overline{h} \operatorname{meas}\{\left|\sin(w/2)\cos(x-\xi) - 2^{-1}\sin(w)\right| < \delta\} \leq C/N,$$

where the constant $C$ is independent of $N$, and only depends on the parameters $\underline{h}, \overline{h}, \underline{w}$ and $\overline{w}$. Hence, $\mathcal{N}^{\mathrm{FNO}}$ satisfies

$$\sup_{\bar{u} \sim \mu} \|\mathcal{N}^{\mathrm{FNO}}(\bar{u}) - \mathcal{G}_{\mathrm{adv}}(\bar{u})\|_{L^1} \leq \frac{C}{N},$$

for a constant $C > 0$ independent of $N$, and where we recall (cp. (D.14) above):

$$k_{\max} = 1, \quad d_v \leq C, \quad \operatorname{depth}(\mathcal{N}^{\mathrm{FNO}}) \leq C\log(N)^2, \quad \operatorname{size}(\mathcal{N}^{\mathrm{FNO}}) \leq C\log(N)^2.$$

The claimed error and complexity bounds of Theorem 3.3 are now immediate upon choosing $N \sim \epsilon^{-1}$. $\qquad \square$

### D.5 PROOF OF THEOREM 3.5

To motivate the proof, we first consider the Burgers' equation with the particular initial data $\bar{u}(x) = -\sin(x)$, with periodic boundary conditions on the interval $x \in [0, 2\pi]$. The solution for this initial datum can be constructed via the well-known method of characteristics; we observe that the solution $u(x,t)$ with initial data $\bar{u}(x)$ is smooth for time $t \in [0,1)$, develops a shock discontinuity at $x = 0$ (and $x = 2\pi$) for $t \geq 1$, but remains otherwise smooth on the interval $x \in (0, 2\pi)$ for all times. In fact, fixing a time $t \geq 0$, the solution $u(x,t)$ can be written down explicitly in terms of the bijective mapping (cp. Figure 4)

$$\Psi_t : [x_t, 2\pi - x_t] \to [0, 2\pi], \quad \Psi_t(x_0) = x_0 - t\sin(x_0),$$

where

$$\begin{cases} x_t = 0, & \text{for } t \leq 1, \\ x_t > 0 \text{ is the unique solution of } x_t = t\sin(x_t), & \text{for } t > 1. \end{cases} \tag{D.15}$$

We note that for given $x_0$, the curve $t \mapsto \Psi_t(x_0)$ traces out the characteristic curve for the Burgers' equation, starting at $x_0$ (and until it collides with the shock). Following the method of characteristics, the solution $u(x,t)$ is then given in terms of $\Psi_t$, by

$$u(x,t) = -\sin\left(\Psi_t^{-1}(x)\right), \quad \text{for } x \in [0, 2\pi]. \tag{D.16}$$

We are ultimately interested in solutions for more general periodic initial data of the form $\bar{u}(x) = -\sin(x - \xi)$; these can easily be obtained from the particular solution (D.16) via a shift. We summarize this observation in the following lemma:

**Lemma D.15.** Let $\xi \in [0, 2\pi)$ be given, fix a time $t \geq 0$. Consider the initial data $\bar{u}(x) = -\sin(x-\xi)$. Then the entropy solution $u(x,t)$ of the Burgers' equations with initial data $\bar{u}$ is given by

$$u(x,t) = \begin{cases} -\sin(\Psi_t^{-1}(x - \xi + 2\pi)), & (x < \xi), \\ -\sin(\Psi_t^{-1}(x - \xi)), & (x \geq \xi), \end{cases} \tag{D.17}$$

for $x \in [0, 2\pi]$, $t \geq 0$.

**Lemma D.16.** Let $t > 1$, and define $U : [0, 2\pi] \to \mathbb{R}$ by $U(x) := -\sin(\Psi_t^{-1}(x))$. There exists $\Delta_t > 0$ (depending on $t$), such that $x \mapsto U(x)$ can be extended to an analytic function $\bar{U} : (-\Delta_t, 2\pi + \Delta_t) \to \mathbb{R}$, $x \mapsto \bar{U}(x)$; i.e., such that $\bar{U}(x) = U(x)$ for all $x \in [0, 2\pi]$.

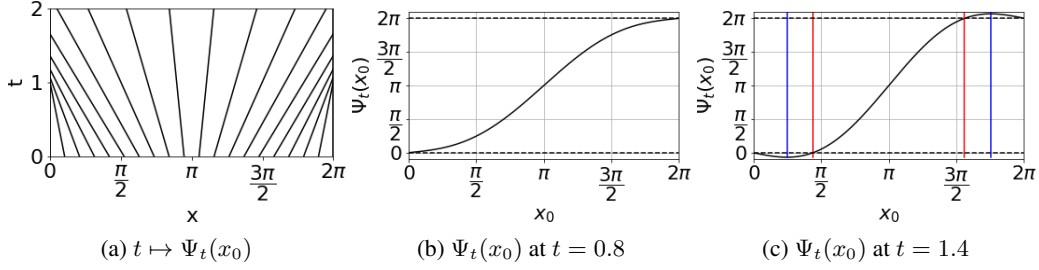

(a) $t \mapsto \Psi_t(x_0)$     (b) $\Psi_t(x_0)$ at $t = 0.8$     (c) $\Psi_t(x_0)$ at $t = 1.4$

Figure 4: Illustration of $\Psi_t(x_0)$: (a) characteristics traced out by $t \mapsto \Psi_t(x_0)$ (until collision with shock), (b) $\Psi_t(x_0)$ before shock formation, (c) $\Psi_t(x_0)$ after shock formation, including the interval $[x_t, 2\pi - x_t]$ (red limits) and the larger domain (blue limits) allowing for bijective analytic continuation, $(\Delta_t, 2\pi - \Delta_t)$.

***Corollary* D.17.** Let $U : [0, 2\pi] \to \mathbb{R}$ be defined as in Lemma D.16. There exists a constant $C > 0$, such that for any $\epsilon > 0$, there exists a ReLU neural network $\Phi_\epsilon : \mathbb{R} \to \mathbb{R}$, such that

$$\sup_{x \in [0, 2\pi]} |\Phi_\epsilon(x) - U(x)| \leq \epsilon,$$

and

$$\mathrm{depth}(\Phi_\epsilon) \leq C \log(\epsilon^{-1})^2, \quad \mathrm{width}(\Phi_\epsilon) \leq C,$$

*Proof.* By Lemma D.16, $U$ can be extended to an analytic function $\bar{U} : (-\Delta_t, 2\pi + \Delta_t) \to \mathbb{R}$. Thus, by Theorem D.6, there exist constants $C, \gamma > 0$, such that for any $L \in \mathbb{N}$, there exists a neural network

$$\widetilde{\Phi}_L : \mathbb{R} \to \mathbb{R},$$

such that

$$\sup_{x \in [0, 2\pi]} |U(x) - \widetilde{\Phi}_L(x)| \leq C \exp(-\gamma L^{1/2}),$$

$$\mathrm{depth}(\widetilde{\Phi}_L) \leq CL, \quad \mathrm{width}(\widetilde{\Phi}_L) \leq C.$$

Given $\epsilon > 0$, we can thus choose $L \geq \gamma^{-2} \log(\epsilon^{-1})^2$, to obtain a neural network $\Phi_\epsilon := \widetilde{\Phi}_L$, such that $\sup_{x \in [0, 2\pi]} |U(x) - \Phi_\epsilon(x)| \leq \epsilon$, and

$$\mathrm{depth}(\Phi_\epsilon) \leq C \log(\epsilon^{-1})^2, \quad \mathrm{width}(\Phi_\epsilon) \leq C,$$

for a constant $C > 0$, independent of $\epsilon$. $\qquad\square$

***Lemma* D.18.** Let $t > 1$, and let $U : [-2\pi, 2\pi] \to \mathbb{R}$ be given by

$$U(x) := \begin{cases} -\sin(\Psi_t^{-1}(x + 2\pi)), & (x < 0), \\ -\sin(\Psi_t^{-1}(x)), & (x \geq 0). \end{cases}$$

There exists a constant $C = C(t) > 0$, depending only on $t$, such that for any $\epsilon \in (0, \frac{1}{2}]$, there exists a neural network $\Phi_\epsilon : \mathbb{R} \to \mathbb{R}$, such that

$$\mathrm{depth}(\Phi_\epsilon) \leq C \log(\epsilon^{-1})^2, \quad \mathrm{width}(\Phi_\epsilon) \leq C,$$

and such that

$$\|\Phi_\epsilon - U\|_{L^1([-2\pi, 2\pi])} \leq \epsilon.$$

*Proof.* By Corollary D.17, there exists a constant $C > 0$, such that for any $\epsilon > 0$, there exist neural networks $\Phi^-, \Phi^+ : \mathbb{R} \to \mathbb{R}$, such that

$$\mathrm{depth}(\Phi^\pm) \leq C \log(\epsilon^{-1})^2, \quad \mathrm{width}(\Phi^\pm) \leq C,$$

and

$$\|\Phi^-(x) - U(x)\|_{L^\infty([-2\pi, 0])} \leq \epsilon, \quad \|\Phi^+(x) - U(x)\|_{L^\infty([0, 2\pi])} \leq \epsilon.$$

This implies that

$$\left\| U - \left[ 1_{[-2\pi,0)}\Phi^- + 1_{[0,2\pi]}\Phi^+ \right] \right\|_{L^\infty([-2\pi,2\pi])} \le \epsilon.$$

By Proposition D.2 (approximation of indicator functions), there exist neural networks $\chi_\epsilon^\pm : \mathbb{R} \to \mathbb{R}$ with uniformly bounded width and depth, such that

$$\left\| \chi_\epsilon^- - 1_{[-2\pi,0]} \right\|_{L^1} \le \epsilon, \quad \left\| \chi_\epsilon^+ - 1_{[0,2\pi]} \right\|_{L^1} \le \epsilon.$$

and $\|\chi_\epsilon^\pm(x)\|_{L^\infty} \le 1$. Combining this with Proposition D.4 (approximation of multiplication), it follows that there exists a neural network

$$\Phi_\epsilon : \mathbb{R} \to \mathbb{R}, \quad \Phi_\epsilon(x) = \widetilde{\times}_\epsilon(\chi_\epsilon^+, \Phi^+) + \widetilde{\times}_\epsilon(\chi_\epsilon^-, \Phi^-),$$

such that

$$\left\| \Phi_\epsilon - \left[ 1_{[-2\pi,0)}\Phi^- + 1_{[0,2\pi]}\Phi^+ \right] \right\|_{L^1([-2\pi,2\pi])}$$
$$\le \left\| \widetilde{\times}_{M,\epsilon}(\chi^+, \Phi^+) - 1_{[0,2\pi]}\Phi^+ \right\|_{L^1([-2\pi,2\pi])}$$
$$+ \left\| \widetilde{\times}_{M,\epsilon}(\chi^-, \Phi^-) - 1_{[-2\pi,0)}\Phi^- \right\|_{L^1([-2\pi,2\pi])}$$

By construction of $\widetilde{\times}_{M,\epsilon}$, we have

$$\left\| \widetilde{\times}_{M,\epsilon}(\chi^+, \Phi^+) - 1_{[0,2\pi]}\Phi^+ \right\|_{L^1} \le \left\| \widetilde{\times}_{M,\epsilon}(\chi^+, \Phi^+) - \chi^+\Phi^+ \right\|_{L^1}$$
$$+ \left\| \left( \chi^+ - 1_{[0,2\pi]} \right)\Phi^+ \right\|_{L^1}$$
$$\le 4\pi\epsilon + \|\chi^+ - 1_{[0,2\pi]}\|_{L^1}\|\Phi^+\|_{L^\infty}$$
$$\le (4\pi + 2)\,\epsilon.$$

And similarly for the other term. Thus, it follows that

$$\left\| \Phi_\epsilon - \left[ 1_{[-2\pi,0)}\Phi^- + 1_{[0,2\pi]}\Phi^+ \right] \right\|_{L^1([-2\pi,2\pi])} \le 2(4\pi + 2)\epsilon,$$

and finally,

$$\|U - \Phi_\epsilon\|_{L^1([-2\pi,2\pi])} \le 4\pi \left\| U - \left[ 1_{[-2\pi,0)}\Phi^- + 1_{[0,2\pi]}\Phi^+ \right] \right\|_{L^\infty([-2\pi,2\pi])}$$
$$+ \left\| \left[ 1_{[-2\pi,0)}\Phi^- + 1_{[0,2\pi]}\Phi^+ \right] - \Phi_\epsilon \right\|_{L^1([-2\pi,2\pi])}$$
$$\le 4\pi\epsilon + 2(4\pi + 2)\epsilon = (12\pi + 4)\epsilon.$$

for a neural network $\Phi_\epsilon : \mathbb{R} \to \mathbb{R}$ of size:

$$\mathrm{depth}(\Phi_\epsilon) \le C\log(\epsilon^{-1})^2, \quad \mathrm{width}(\Phi_\epsilon) \le C.$$

Replacing $\epsilon$ with $\widetilde{\epsilon} = \epsilon/(12\pi + 4)$ yields the claimed estimate for $\Phi_{\widetilde{\epsilon}}$. $\qquad\square$

Based on the above results, we can now prove the claimed error and complexity estimate for the shift-DeepONet approximation of the Burgers' equation, Theorem 3.5.

*Proof of Theorem 3.5.* Fix $t > 1$. Let $U : [-2\pi, 2\pi] \to \mathbb{R}$ be the function from Lemma D.18. By Lemma D.15, the exact solution of the Burgers' equation with initial data $\bar{u}(x) = -\sin(x - \xi)$ at time $t$, is given by

$$u(x, t) = U(x - \xi), \quad \forall\, x \in [0, 2\pi].$$

From Lemma D.18 (note that $x - \xi \in [-2\pi, 2\pi]$), it follows that there exists a constant $C > 0$, such that for any $\epsilon > 0$, there exists a neural network $\Phi_\epsilon : \mathbb{R} \to \mathbb{R}$, such that

$$\|u(\,\cdot\,, t) - \Phi_\epsilon(\,\cdot\, - \xi)\|_{L^1([0,2\pi])} \le \epsilon,$$

and

$$\mathrm{depth}(\Phi_\epsilon) \le C\log(\epsilon^{-1})^2, \quad \mathrm{width}(\Phi_\epsilon) \le C.$$

We finally observe that for equidistant sensor points $x_0, x_1, x_2 \in [0, 2\pi]$, $x_j = 2\pi j/3$, there exists a matrix $A \in \mathbb{R}^{2 \times 3}$, which for any function of the form $g(x) = \alpha_0 + \alpha_1 \sin(x) + \alpha_2 \cos(x)$ maps

$$[g(x_0), g(x_1), g(x_2)]^T \mapsto A \cdot [g(x_0), g(x_1), g(x_2)]^T = [-\alpha_1, \alpha_2]^T.$$

Clearly, the considered initial data $\bar{u}(x) = -\sin(x - \xi)$ is of this form, for any $\xi \in [0, 2\pi)$, or more precisely, we have

$$\bar{u}(x) = -\sin(x - \xi) = -\cos(\xi)\sin(x) - \sin(\xi)\cos(x),$$

so that

$$A \cdot [\bar{u}(x_0), \bar{u}(x_1), \bar{u}(x_2)]^T = [\cos(\xi), \sin(\xi)].$$

As a next step, we recall that there exists $C > 0$, such that for any $\epsilon > 0$, there exists a neural network $\Xi_\epsilon : \mathbb{R}^2 \to [0, 2\pi]$ (cp. Lemma D.8), such that

$$\sup_{\xi \in [0, 2\pi - \epsilon)} |\xi - \Xi_\epsilon(\cos(\xi), \sin(\xi))| < \epsilon,$$

such that $\Xi_\epsilon(\cos(\xi), \sin(\xi)) \in [0, 2\pi]$ for all $\xi$, and

$$\mathrm{depth}(\Xi_\epsilon) \le C \log(\epsilon^{-1})^2, \quad \mathrm{width}(\Xi_\epsilon) \le C.$$

Based on this network $\Xi_\epsilon$, we can now define a shift-DeepONet approximation of $\mathcal{N}^{\mathrm{sDON}}(\bar{u}) \approx \mathcal{G}_{\mathrm{Burg}}(\bar{u})$ of size:

$$\mathrm{depth}(\mathcal{N}^{\mathrm{sDON}}) \le C \log(\epsilon^{-1})^2, \quad \mathrm{width}(\mathcal{N}^{\mathrm{sDON}}) \le C,$$

by the composition

$$\mathcal{N}^{\mathrm{sDON}}(\bar{u})(x) := \Phi_\epsilon(x - \Xi_\epsilon(A \cdot \bar{u}(\boldsymbol{X}))), \tag{D.18}$$

where $\bar{u}(\boldsymbol{X}) := [\bar{u}(x_0), \bar{u}(x_1), \bar{u}(x_2)]^T$, and we note that (denoting $\Xi_\epsilon := \Xi_\epsilon(A \cdot \bar{u}(\boldsymbol{X}))$), we have for $\xi \in [0, 2\pi - \epsilon]$:

$$
\begin{aligned}
\|\mathcal{G}_{\mathrm{Burg}}(\bar{u}) - \mathcal{N}^{\mathrm{sDON}}(\bar{u})\|_{L^1} &= \|U(\cdot - \xi) - \Phi_\epsilon(\cdot - \Xi_\epsilon)\|_{L^1} \\
&\le \|U(\cdot - \xi) - U(\cdot - \Xi_\epsilon)\|_{L^1} \\
&\quad + \|U(\cdot - \Xi_\epsilon) - \Phi_\epsilon(x - \Xi_\epsilon)\|_{L^1} \\
&\le C|\xi - \Xi_\epsilon| + \epsilon \\
&\le (C + 1)\epsilon,
\end{aligned}
$$

where $C$ only depends on $U$, and is independent of $\epsilon > 0$. On the other hand, for $\xi > 2\pi - \epsilon$, we have

$$
\begin{aligned}
\|\mathcal{G}_{\mathrm{Burg}}(\bar{u}) - \mathcal{N}^{\mathrm{sDON}}(\bar{u})\|_{L^1} &= \|U(\cdot - \xi) - \Phi_\epsilon(\cdot - \Xi_\epsilon)\|_{L^1} \\
&\le \|U(\cdot - \xi)\|_{L^1} + \|\Phi_\epsilon(\cdot - \Xi_\epsilon)\|_{L^1} \\
&\le 2\pi \left( \|U(\cdot - \xi)\|_{L^\infty} + \|\Phi_\epsilon(\cdot - \Xi_\epsilon)\|_{L^\infty} \right) \\
&\le 6\pi,
\end{aligned}
$$

is uniformly bounded. It follows that

$$
\begin{aligned}
\mathbb{E}_{\bar{u} \sim \mu}\left[ \|\mathcal{G}_{\mathrm{Burg}}(\bar{u}) - \mathcal{N}^{\mathrm{sDON}}(\bar{u})\|_{L^1} \right] &= \int_0^{2\pi - \epsilon} + \int_{2\pi - \epsilon}^{2\pi} \|\mathcal{G}_{\mathrm{Burg}}(\bar{u}) - \mathcal{N}^{\mathrm{sDON}}(\bar{u})\|_{L^1} \, d\xi \\
&\le 2\pi(C + 1)\epsilon + 6\pi\epsilon,
\end{aligned}
$$

with a constant $C > 0$, independent of $\epsilon > 0$. Replacing $\epsilon$ by $\widetilde{\epsilon} = \epsilon/C'$ for a sufficiently large constant $C' > 0$ (depending only on the constants in the last estimate above), one readily sees that there exists a shift-DeepONet $\mathcal{N}^{\mathrm{sDON}}$, such that

$$\mathscr{E}(\mathcal{N}^{\mathrm{sDON}}) = \mathbb{E}_{\bar{u} \sim \mu}\left[ \|\mathcal{G}_{\mathrm{Burg}}(\bar{u}) - \mathcal{N}^{\mathrm{sDON}}(\bar{u})\|_{L^1} \right] \le \epsilon,$$

and such that

$$\mathrm{width}(\mathcal{N}^{\mathrm{sDON}}) \le C, \quad \mathrm{depth}(\mathcal{N}^{\mathrm{sDON}}) \le C \log(\epsilon^{-1})^2,$$

and $\mathrm{size}(\mathcal{N}^{\mathrm{sDON}}) \le C \log(\epsilon^{-1})^2$, for a constant $C > 0$, independent of $\epsilon > 0$. This concludes our proof. $\qquad \square$

### D.6 Proof of Theorem 3.6

*Proof.* **Step 1:** Assume that the grid size is $N \geq 3$. Then there exists an FNO $\mathcal{N}_1$, such that

$$\mathcal{N}_1(\bar{u}) = [\cos(\xi), \sin(\xi)],$$

and $\mathrm{depth}(\mathcal{N}_1) \leq C$, $d_v \leq C$, $k_{\max} = 1$.

To see this, we note that for any $\xi \in [0, 2\pi]$, the input function $\bar{u}(x) = -\sin(x - \xi) = -\cos(\xi)\sin(x) - \sin(\xi)\cos(x)$ can be written in terms of a sine/cosine expansion with coefficients $\cos(\xi)$ and $\sin(\xi)$. For $N \geq 3$ grid points, these coefficients can be retrieved exactly by a discrete Fourier transform. Therefore, combining a suitable lifting to $d_v = 2$ with a Fourier multiplier matrix $P$, we can exactly represent the mapping

$$\bar{u} \mapsto \mathcal{F}_N^{-1}(P \cdot \mathcal{F}_N(R \cdot \bar{u})) = \begin{bmatrix} \cos(\xi)\sin(x) \\ \sin(\xi)\cos(x) \end{bmatrix},$$

by a linear Fourier layer. Adding a suitable bias function $b(x) = [\sin(x), \cos(x)]^T$, and composing with an additional nonlinear layer, it is then straightforward to check that there exists a (ReLU-)FNO, such that

$$\bar{u} \mapsto \begin{bmatrix} \cos(\xi)\sin(x) + \sin(x) \\ \sin(\xi)\cos(x) + \cos(x) \end{bmatrix} = \begin{bmatrix} (\cos(\xi) + 1)\sin(x) \\ (\sin(\xi) + 1)\cos(x) \end{bmatrix}$$

$$\mapsto \begin{bmatrix} |\cos(\xi) + 1||\sin(x)| \\ |\sin(\xi) + 1||\cos(x)| \end{bmatrix}$$

$$\mapsto \begin{bmatrix} |\cos(\xi) + 1| \sum_{j=1}^{N} |\sin(x_j)| \\ |\sin(\xi) + 1| \sum_{j=1}^{N} |\cos(x_j)| \end{bmatrix}$$

$$\mapsto \begin{bmatrix} |\cos(\xi) + 1| - 1 \\ |\sin(\xi) + 1| - 1 \end{bmatrix} = \begin{bmatrix} \cos(\xi) \\ \sin(\xi) \end{bmatrix}.$$

**Step 2:** Given this construction of $\mathcal{N}_1$, the remainder of the proof follows essentially the same argument as in the proof D.5 of Theorem 3.5: We again observe that the solution $u(x, t)$ with initial data $\bar{u}(x) = -\sin(x - \xi)$ is well approximated by the composition

$$\mathcal{N}^{\mathrm{FNO}}(\bar{u})(x) := \Phi_\epsilon(x - \Xi_\epsilon(\mathcal{N}_1(\bar{u}))),$$

such that (by verbatim repetition of the calculations after (D.18) for shift-DeepONets)

$$\mathscr{E}(\mathcal{N}^{\mathrm{FNO}}) = \mathbb{E}_{\bar{u} \sim \mu} \left[ \|\mathcal{G}_{\mathrm{Burg}}(\bar{u}) - \mathcal{N}^{\mathrm{FNO}}(\bar{u})\|_{L^1} \right] \leq \epsilon,$$

and where $\Phi_\epsilon : \mathbb{R} \to \mathbb{R}$ is a ReLU neural network of width

$$\mathrm{depth}(\Phi_\epsilon) \leq C \log(\epsilon^{-1})^2, \quad \mathrm{width}(\Phi_\epsilon) \leq C,$$

and $\Xi_\epsilon : \mathbb{R}^2 \to [0, 2\pi]$ is an ReLU network with

$$\mathrm{depth}(\Xi_\epsilon) \leq C \log(\epsilon^{-1})^2, \quad \mathrm{width}(\Xi_\epsilon) \leq C.$$

Being the composition of an FNO $\mathcal{N}_1$ satisfying $k_{\max} = 1$, $d_v \leq C$, $\mathrm{depth}(\mathcal{N}_1) \leq C$ with the two ordinary neural networks $\Phi_\epsilon$ and $\Xi_\epsilon$, it follows that $\mathcal{N}^{\mathrm{FNO}}$ can itself be represented by an FNO with $k_{\max} = 1$, $d_v \leq C$ and $\mathrm{depth}(\mathcal{N}^{\mathrm{FNO}}) \leq C \log(\epsilon^{-1})^2$. By the general complexity estimate (B.2),

$$\mathrm{size}(\mathcal{N}^{\mathrm{FNO}}) \lesssim d_v^2 k_{\max}^d \mathrm{depth}(\mathcal{N}^{\mathrm{FNO}}),$$

we also obtain the claimed an upper complexity bound $\mathrm{size}(\mathcal{N}^{\mathrm{FNO}}) \leq C \log(\epsilon^{-1})^2$. $\qquad\square$

## E Details of Numerical Experiments and Further Experimental Results.

### E.1 Training and Architecture Details

Below, details concerning the model architectures and training are discussed.

### E.1.1 FEED FORWARD DENSE NEURAL NETWORKS

Given an input $y \in \mathbb{R}^m$, a feedforward neural network (also termed as a multi-layer perceptron), transforms it to an output, through a layer of units (neurons) which compose of either affine-linear maps between units (in successive layers) or scalar nonlinear activation functions within units Goodfellow et al. (2016), resulting in the representation,

$$u_\theta(y) = C_L \circ \sigma \circ C_{L-1} \ldots \circ \sigma \circ C_2 \circ \sigma \circ C_1(y). \tag{E.1}$$

Here, $\circ$ refers to the composition of functions and $\sigma$ is a scalar (nonlinear) activation function. For any $1 \le \ell \le L$, we define

$$C_\ell z_\ell = W_\ell z_\ell + b_\ell, \text{ for } W_\ell \in \mathbb{R}^{d_{\ell+1} \times d_\ell}, z_\ell \in \mathbb{R}^{d_\ell}, b_\ell \in \mathbb{R}^{d_{\ell+1}}., \tag{E.2}$$

and denote,

$$\theta = \{W_\ell, b_\ell\}_{\ell=1}^L, \tag{E.3}$$

to be the concatenated set of (tunable) weights for the network. Thus in the terminology of machine learning, a feed forward neural network (E.1) consists of an input layer, an output layer, and $L$ hidden layers with $d_\ell$ neurons, $1 < \ell < L$. In all numerical experiments, we consider a uniform number of neurons across all the layer of the network $d_\ell = d_{\ell-1} = d$, $1 < \ell < L$. The number of layers $L$, neurons $d$ and the activation function $\sigma$ are chosen though cross-validation.

### E.1.2 RESNET

A residual neural network consists of *residual blocks* which use *skip or shortcut connections* to facilitate the training procedure of deep networks He et al. (2016). A residual block spanning $k$ layers is defined as follows,

$$r(z_\ell, z_{\ell-k}) = \sigma(W_\ell z_\ell + b_\ell) + z_{\ell-k}. \tag{E.4}$$

In all numerical experiments we set $k = 2$.

The residual network takes as input a sample function $\bar{u} \in \mathcal{X}$, encoded at the *Cartesian grid* points $(x_1, \ldots, x_m)$, $\mathcal{E}(\bar{u}) = (\bar{u}(x_1), \ldots, \bar{u}(x_m)) \in \mathbb{R}^m$, and outputs the output sample $\mathcal{G}(\bar{u}) \in \mathcal{Y}$ encoded at the same set of points, $\mathcal{E}(\mathcal{G}(\bar{u})) = (\mathcal{G}(\bar{u})(x_1), \ldots, \mathcal{G}(\bar{u})(x_m)) \in \mathbb{R}^m$. For the compressible Euler equations the encoded input is defined as

$$\mathcal{E}(\bar{u}) = \Big(\rho_0(x_1), \ldots, \rho_0(x_m), \rho_0(x_1)u_0(x_1), \ldots, \rho_0(x_m)u_0(x_m), E_0(x_1), \ldots, E_0(x_m)\Big) \in \mathbb{R}^{3m}$$

$$\mathcal{E}(\bar{u}) = \Big(\rho_0(x_1), \ldots, \rho_0(x_{m^2}), \rho_0(x_1)u_0(x_1), \ldots, \rho_0(x_{m^2})u_0(x_{m^2}),$$

$$\rho_0(x_1)v_0(x_1), \ldots, \rho_0(x_{m^2})v_0(x_{m^2})E_0(x_1), \ldots, E_0(x_{m^2})\Big) \in \mathbb{R}^{4m^2}$$

$$\tag{E.5}$$

for the 1d and 2d problem, respectively.

### E.1.3 FULLY CONVOLUTIONAL NEURAL NETWORK

Fully convolutional neural networks are a special class of convolutional networks which are independent of the input resolution. The networks consist of an *encoder* and *decoder*, both defined by a composition of linear and nonlinear transformations:

$$\begin{aligned} E_{\theta_e}(y) &= C_L^e \circ \sigma \circ C_{L-1}^e \ldots \circ \sigma \circ C_2^e \circ \sigma \circ C_1^e(y), \\ D_{\theta_d}(z) &= C_L^d \circ \sigma \circ C_{L-1}^d \ldots \circ \sigma \circ C_2^d \circ \sigma \circ C_1^d(z), \\ u_\theta(y) &= D_{\theta_d} \circ E_{\theta_e}(y). \end{aligned} \tag{E.6}$$

The affine transformation $C_\ell$ commonly corresponds to a *convolution* operation in the encoder, and *transposed convolution* (also know as *deconvolution*), in the decoder. The latter can also be performed with a simple *linear* (or *bilinear*) *upsampling* and a convolution operation, similar to the encoder.

The (de)convolution is performed with a kernel $W_\ell \in \mathbb{R}^{k_\ell}$ (for 1d-problems, and $W_\ell \in \mathbb{R}^{k_\ell \times k_\ell}$ for 2d-problems), stride $s$ and padding $p$. It takes as input a tensor $z_\ell \in \mathbb{R}^{w_\ell \times c_\ell}$ (for 1d-problems, and $z_\ell \in \mathbb{R}^{w_\ell \times h_\ell \times c_\ell}$ for 2d-problems), with $c_\ell$ being the number of input channels, and computes

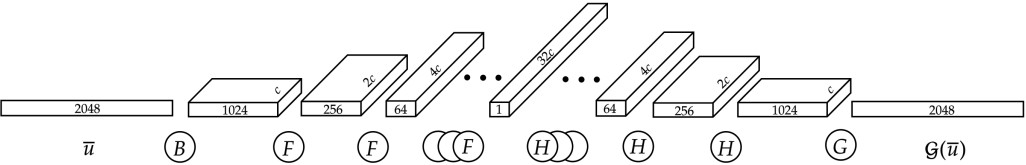

Figure 5: Fully convolutional neural network architecture for the linear advection equation and shock tube benchmarks. $B(z) = BN \circ \sigma \circ C^e(z)$, with $BN$ denoting a batch normalization and $C^e$ a convolution defined by the tuple $(3, 2, 1, c_{in}, c)$, with $c_{in} = 1$ for the advection equation and $c_{in} = 3$ shocktube benchmarks. $F(z) = BN \circ \sigma \circ C_4^e \circ BN \circ \sigma \circ C_3^e \circ BN \circ \sigma \circ C_2^e \circ BN \circ \sigma \circ C_1^e(z)$, with $C_1^e$, $C_2^e$, $C_3^e$, $C_4^e$ identified with $(3, 2, 1, c_{in}, 2c_{in})$, $(1, 1, 0, 2c_{in}, 2c_{in})$, $(1, 1, 0, 2c_{in}, 2c_{in})$, $(3, 2, 1, 2c_{in}, 2c_{in})$. $H(z) = BN \circ \sigma \circ C_4^d \circ BN \circ \sigma \circ C_3^d \circ BN \circ \sigma \circ C_2^d \circ BN \circ \sigma \circ C_1^d(z)$, with $C_1^d$, $C_2^d$, $C_3^d$, $C_4^d$ transposed convolutions defined by $(3, 2, 1, c_{in}, c_{in})$, $(1, 1, 0, c_{in}, c_{in})$, $(1, 1, 0, c_{in}, c_{in})$, $(3, 2, 1, c_{in}, 0.5c_{in})$. $G$ is a transposed convolution defined by the tuple $(3, 2, 1, c, 1)$.

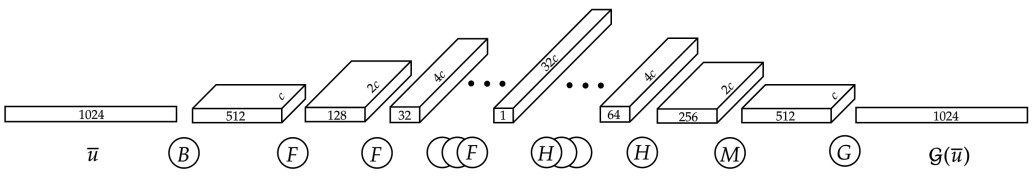

Figure 6: Fully convolutional neural network architecture for the Burgers' equation benchmark: $B, F$, $G$ follow the same definition as in the caption of figure 5. $H(z) = \sigma \circ BN \circ C_4^e \circ UP \circ \sigma \circ BN \circ C_3^e \circ UP \circ \sigma \circ BN \circ C_2^e \circ UP \circ \sigma \circ BN \circ UP \circ C_1^e(z)$, with $C_1^e, C_2^e, C_3^e, C_4^e$ being standard convolutions defied by the tuples $(3, 2, 1, c_{in}, c_{in})$, $(1, 1, 0, c_{in}, c_{in})$, $(1, 1, 0, c_{in}, c_{in})$, $(3, 2, 1, c_{in}, 0.5c_{in})$, and $UP$ denoting up-sampling operation with scaling factor 2.

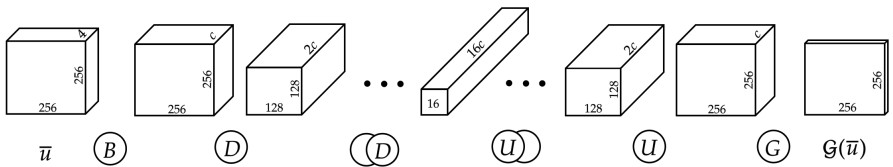

Figure 7: Fully convolutional neural network architecture for the 2-dimensional Riemann problem. $B(z) = \sigma \circ BN \circ C_2^e \circ \sigma \circ BN \circ C_1^e$, with $C_1^e$ and $C_2^e$ being standard convolutions defined by the tuples $(3, 1, 0, c_{in}, 2c_{in})$, $(3, 1, 0, 2c_{in}, 2c_{in})$. $D(z) = B(z) \circ MP(z)$, with MP being a max pool with kernel size 2. $U(Z) = B(z) \circ UP(z)$, where $UP$ denotes an up-sampling with scale factor 2. $G$ is a convolution defined by $(1, 1, 0, c, 1)$

$z_{\ell+1} \in \mathbb{R}^{w_{\ell+1} \times c_{\ell+1}}$ (for 1d-problems, and $z_{\ell+1} \in \mathbb{R}^{w_{\ell+1} \times h_{\ell+1} \times c_{\ell+1}}$ for 2d-problems). Therefore, a (de)convolutional affine transformation can be uniquely identified with the tuple $(k_\ell, s, p, c_\ell, c_{\ell+1})$.

The main difference between the encoder's and decoder's transformation is that, for the encoder $h_{\ell+1} < h_\ell, w_{\ell+1} < w_\ell, c_{\ell+1} > c_\ell$ and for the decoder $h_{\ell+1} > h_\ell, w_{\ell+1} > w_\ell, c_{\ell+1} < c_\ell$.

For the linear advection equation and the Burgers' equation we employ the same variable encoding of the input and output samples as ResNet. On the other hand, for the compressible Euler equations, each input variable is embedded in an individual channel. More precisely, $\mathcal{E}(\bar{u}) \in \mathbb{R}^{m \times 3}$ for the shock-tube problem, and $\mathcal{E}(\bar{u}) \in \mathbb{R}^{m \times m \times 4}$ for the 2d Riemann problem. The architectures used in the benchmarks examples are shown in figures 5, 6, 7.

In the experiments, the number of channel $c$ (see figures 5, 6, 7 for an explanation of its meaning) and the activation function $\sigma$ are selected with cross-validation.

### E.1.4 DEEPONET AND SHIFT-DEEPONET

The architectures of branch and trunk are chosen according to the benchmark addressed. In particular, for the first two numerical experiments, we employ standard feed-forward neural networks for both branch and trunk-net, with a skip connection between the first and the last hidden layer in the branch.

On the other hand, for the compressible Euler equation we use a convolutional network obtained as a composition of $L$ blocks, each defined as:

$$B(z_\ell) = BN \circ \sigma \circ C_\ell(z_\ell), \quad 1 < \ell < L, \tag{E.7}$$

with $BN$ denoting a batch normalization. The convolution operation is instead defined by $k_\ell = 3$, $s = 2$, $p = 1$, $c_\ell = c_{\ell+1} = d$, for all $1 < \ell < L - 1$. The output is then flattened and forwarded through a multi layer perceptron with 2 layer with 256 neurons and activation function $\sigma$.

For the *shift* and *scale*-nets of shift-DeepONet, we use the same architecture as the branch.

Differently from the rest of the models, the training samples for DeepONet and shift-DeepONet are encoded at $m$ and $n$ *uniformly distributed random* points, respectively. Specifically, the encoding points represent a randomly picked subset of the grid points used for the other models. The number of encoding points $m$ and $n$, together with the number of layers $L$, units $d$ and activation function of trunk and branch-nets, are chosen through cross-validation.

### E.1.5 FOURIER NEURAL OPERATOR

We use the implementation of the FNO model provided by the authors of Li et al. (2021a). Specifically, the lifting $R$ is defined by a linear transformation from $\mathbb{R}^{d_u \times m}$ to $\mathbb{R}^{d_v \times m}$, where $d_u$ is the number of inputs, and the projection $Q$ to the target space performed by a neural network with a single hidden layer with 128 neurons and $GeLU$ activation function. The same activation function is used for all the Fourier layers, as well. Moreover, the weight matrix $W_\ell$ used in the residual connection derives from a convolutional layer defined by $(k_\ell = 1, s = 1, p = 0, c_\ell = d_v, c_{\ell+1} = d_v)$, for all $1 < \ell < L - 1$. We use the same samples encoding employed for the fully convolutional models. The lifting dimension $d_v$, the number of Fourier layers $L$ and $k_{max}$, defined in 2, are the only objectives of cross-validation.

### E.1.6 TRAINING DETAILS

For all the benchmarks, a training set with 1024 samples, and a validation and test set each with 128 samples, are used. The training is performed with the ADAM optimizer, with learning rate $5 \cdot 10^{-4}$ for 10000 epochs and minimizing the $L^1$-loss function. We use the learning rate schedulers defined in table 2. We train the models in mini-batches of size 10. A weight decay of $10^{-6}$ is used for ResNet (all numerical experiments), DON and sDON (linear advection equation, Burgers' equation, and shock-tube problem). On the other hand, no weight decay is employed for remaining experiments and models. At every epoch the relative $L^1$-error is computed on the validation set, and the set of trainable parameters resulting in the lowest error during the entire process saved for testing. Therefore, no early stopping is used. The models hyperparameters are selected by running grid searches over a range of hyperparameter values and selecting the configuration realizing the lowest relative $L^1$-error on the validation set. For instance, the model size (minimum and maximum number of trainable parameters) that are covered in this grid search are reported in Table 3.

The results of the grid search i.e., the best performing hyperparameter configurations for each model and each benchmark, are reported in tables 4, 5, 6, 7 and 8.

### E.2 FURTHER EXPERIMENTAL RESULTS.

In this section, we present some further experimental results which supplement the results presented in Table 1 of the main text. We start by presenting more statistical information about the median errors shown in Table 1. To this end, in Table 9, we show the errors, for each model on each benchmark, corresponding to the $0.25$ and $0.75$ *quantiles*, within the test set. We observe from this table that the same trend, as seen in Table 1, also holds for the statistical spread. In particular, FNO and shift-DeepONet outperform DeepONet and the other baselines on every experiment. Similarly, FNO

| | ResNet | FCNN | DeepONet | Shift - DeepONet | FNO |
|---|---|---|---|---|---|
| **Advection Equation** | Step-wise decay 100 steps, $\gamma = 0.999$ | Step-wise Decay 50 steps, $\gamma = 0.99$ | Step-wise decay 100 steps, $\gamma = 0.999$ | Exponential decay $\gamma = 0.999$ | None |
| **Burgers' Equation** | Step-wise decay 100 steps, $\gamma = 0.999$ | Step-wise Decay 50 steps, $\gamma = 0.99$ | Step-wise decay 100 steps, $\gamma = 0.999$ | Exponential decay $\gamma = 0.999$ | None |
| **Shocktube Problem** | Step-wise decay 100 steps, $\gamma = 0.999$ | Step-wise Decay 50 steps, $\gamma = 0.99$ | Step-wise decay 100 steps, $\gamma = 0.999$ | Exponential decay $\gamma = 0.999$ | None |
| **2D Riemann Problem** | Step-wise decay 100 steps, $\gamma = 0.999$ | Step-wise Decay 50 steps, $\gamma = 0.99$ | Exponential decay $\gamma = 0.999$ | Exponential decay $\gamma = 0.999$ | None |

Table 2: Learning rate scheduler for different benchmarks and different models: $\gamma$ denotes the learning rate decay factor

| | ResNet | FCNN | DeepONet | Shift - DeepONet | FNO |
|---|---|---|---|---|---|
| **Advection Equation** | 576,768 1,515,008 | 1,156,449 1,8240,545 | 519,781 892,561 | 1,018,825 1,835,297 | 22,945 352,961 |
| **Burgers' Equation** | 313,600 989,696 | 1,155,025 18,219,489 | 519,781 892,561 | 1,018,825 1,835,297 | 22,945 352,961 |
| **Shocktube Problem** | 1,101,056 2,563,584 | 1,156,497 18,240,737 | 1,344,357 3,190,673 | 3,492,553 8,729,633 | 23,009 353,089 |
| **2D Riemann Problem** | 42,059,008 84,416,000 | 442,985 7,066,529 | 361,157 2,573,513 | 821,961 7,082,785 | 268,833 13,132,737 |

Table 3: Minimum (Top sub-row) and maximum (Bottom sub-row) number of trainable parameters among the grid-search hyperparameters configurations.

| | Advection Equation | Burgers' Equation | Shocktube Problem | 2D Riemann Problem |
|---|---|---|---|---|
| $\sigma$ | Leaky $ReLU$ | Leaky $ReLU$ | Leaky $ReLU$ | Leaky $ReLU$ |
| $L$ | 4 | 8 | 8 | 8 |
| $d$ | 128 | 256 | 256 | 256 |
| Trainable Params | 576,768 | 989,696 | 2,563,584 | 84,416,000 |

Table 4: ResNet best performing hyperparameters configuration for different benchmark problems.

| | Advection Equation | Burgers' Equation | Shocktube Problem | 2D Riemann Problem |
|---|---|---|---|---|
| $\sigma$ | Leaky $ReLU$ | Leaky $ReLU$ | Leaky $ReLU$ | $ReLU$ |
| $c$ | 8 | 16 | 16 | 16 |
| Trainable Params | 1,156,449 | 4,576,033 | 4,581,537 | 1,768,401 |

Table 5: Fully convolutional neural network best performing hyperparameters configuration for different benchmark problems.

| | Advection Equation | Burgers' Equation | Shocktube Problem | 2D Riemann Problem |
|---|---|---|---|---|
| $m$ | 256 | 512 | 512 | $64^2$ |
| $n$ | 512 | 256 | 256 | $256^2$ |
| $p$ | 200 | 50 | 200 | 100 |
| $L_{branch}$ | 3 | 3 | 3 | 3 |
| $L_{trunk}$ | 6 | 4 | 6 | 6 |
| $d_{branch}$ | 256 | 256 | 256 | 32 |
| $d_{trunk}$ | 256 | 256 | 256 | 256 |
| $\sigma_{branch}$ | Leaky $ReLU$ | Leaky $ReLU$ | Leaky $ReLU$ | $SoftSign$ |
| $\sigma_{trunk}$ | Leaky $ReLU$ | Leaky $ReLU$ | Leaky $ReLU$ | Leaky $ReLU$ |
| Trainable Params | 761,233 | 618,085 | 3,190,673 | 607,433 |

Table 6: DeepONet best performing hyperparameters configuration for different benchmark problems.

| | Advection Equation | Burgers' Equation | Shocktube Problem | 2D Riemann Problem |
|---|---|---|---|---|
| $m$ | 512 | 512 | 512 | $256^2$ |
| $n$ | 512 | 512 | 512 | $128^2$ |
| $p$ | 50 | 200 | 100 | 50 |
| $L_{branch}$ | 3 | 4 | 4 | 3 |
| $L_{trunk}$ | 6 | 6 | 6 | 6 |
| $d_{branch}$ | 256 | 256 | 256 | 32 |
| $d_{trunk}$ | 256 | 256 | 256 | 256 |
| $\sigma_{branch}$ | Leaky $ReLU$ | Leaky $ReLU$ | Leaky $ReLU$ | Leaky $ReLU$ |
| $\sigma_{trunk}$ | Leaky $ReLU$ | Leaky $ReLU$ | Leaky $ReLU$ | Leaky $ReLU$ |
| Trainable Params | 1,445,321 | 1,835,297 | 6,047,633 | 6,851,785 |

Table 7: Shift DeepONet best performing hyperparameters configuration for different benchmark problems.

| | Advection Equation | Burgers' Equation | Shocktube Problem | 2D Riemann Problem |
|---|---|---|---|---|
| $k_{max}$ | 15 | 19 | 7 | 15 |
| $d_v$ | 64 | 32 | 32 | 64 |
| $L$ | 2 | 4 | 4 | 4 |
| Trainable Params | 148,033 | 90,593 | 41,505 | 8,414,145 |

Table 8: Fourier neural operator best performing hyperparameters configuration for different benchmark problems.

| | ResNet | FCNN | DeepONet | Shift - DeepONet | FNO |
|---|---|---|---|---|---|
| **Advection Equation** | 10.1% - 22.8% | $8.2\% - 17.3\%$ | $4.9\% - 13.8\%$ | $1.4\% - 5.4\%$ | $0.35\% - 1.25\%$ |
| **Burgers' Equation** | $18.8\% - 22\%$ | $20.3\% - 25.7\%$ | $25.4\% - 32.4\%$ | $6.7\% - 9.6\%$ | $1.3\% - 1.9\%$ |
| **Shocktube Problem** | $3.6\% - 5.6\%$ | $7.0\% - 10.25\%$ | $3.4\% - 5.4\%$ | $2.0\% - 3.75\%$ | $1.2\% - 2.1\%$ |
| **2D Riemann Problem** | $2.4\% - 2.9\%$ | $0.17\% - 0.21\%$ | $0.77\% - 1.1\%$ | $0.10\% - 0.15\%$ | $0.10\% - 0.14\%$ |

Table 9: 0.25 and 0.75 quantile of the relative $L^1$ error computed over 128 testing samples for different benchmarks with different models.

| Test Samples | ResNet | FCNN | DeepONet | Shift - DeepONet | FNO |
|---|---|---|---|---|---|
| 128 | $10.1, 14.8, 22.8$ | $8.2, 11.6, 17.3$ | $4.9, 8.0, 13.8$ | $1.4, 2.8, 5.4$ | $0.25, 0.71, 1.25$ |
| 256 | $11.1, 15.5, 22.8$ | $8.0, 11.6, 16.8$ | $5.4, 9.2, 15.1$ | $1.5, 2.8, 5.1$ | $0.36, 0.63, 1.3$ |
| 512 | $11.3, 15.9, 23.6$ | $8.3, 11.6, 16.8$ | $5.4, 9.6, 15.2$ | $1.4, 2.8, 6.5$ | $0.36, 0.65, 1.36$ |
| 1024 | $11.0, 15.25, 23.1$ | $8.0, 11.35, 16.0$ | $5.4, 9.1, 15.4$ | $1.4, 2.8, 5.9$ | $0.35, 0.62, 1.29$ |
| 2048 | $11.31, 15.7, 24.0$ | $8.2, 11.5, 16.8$ | $5.4, 9.3, 15.8$ | $1.5, 2.9, 6.6$ | $0.36, 0.67, 1.37$ |

Table 10: 0.25, 0.5 and 0.75 quantile of the relative $L^1$ error computed over 128, 256, 512, 1024 and 2048 testing samples for the linear advection equation with different models.

| $k_{max}$ | $L$ | $d_v$ | **Trainable Params** | **FNO - Median Testing $L^1$-error** | **FFT - Median $L^1$-error** |
|---|---|---|---|---|---|
| 0 | 3 | 192 | 247,169 | 2.30% | 164.2 % |
| 1 | 3 | 160 | 252,097 | 1.21 % | 137.9 % |
| 3 | 3 | 128 | 263,169 | 0.48 % | 63.3 % |
| 7 | 3 | 92 | 241,113 | 0.54 % | 38.9 % |

Table 11: Testing error obtained by training FNO with different number of modes for the linear advection problem and corresponding error obtained with linear Fourier projection.

| $k_{max}$ | $L$ | $d_v$ | **Trainable Params** | **FNO - Median Training $L^1$-error** | **FNO - Median Testing $L^1$-error** | **FFT - Median $L^1$-error** |
|---|---|---|---|---|---|---|
| 0 | 3 | 64 | 33,409 | 2.2% | 2.38% | 164.2 % |
| 1 | 3 | 64 | 45,697 | 0.70% | 0.80 % | 137.9 % |
| 3 | 3 | 64 | 70,273 | 0.54% | 0.55 % | 63.3 % |
| 7 | 3 | 64 | 119,425 | 0.42% | 0.46 % | 38.9 % |

Table 12: Training and testing error obtained by training FNO with different number of modes for the linear advection problem and corresponding error obtained with linear Fourier projection.

| $p$ | Trainable Parameters DON / SDON | DON - Median Training $L^1$-error | SDON - Median Training $L^1$-error | DON - Median Testing $L^1$-error | SDON - Median Testing $L^1$-error |
|---|---|---|---|---|---|
| 1 | 658947 / 1054213 | 100.1% | 4.1% | 100.1% | 13.1% |
| 2 | 659461 / 1055497 | 100.0% | 3.5% | 100.0% | 5.0% |
| 4 | 660489 / 1058065 | 40.9% | 2.5% | 37.5% | 5.0% |
| 8 | 662545 / 1063201 | 26.6% | 1.6% | 29.7% | 4.1% |
| 16 | 666657 / 1073473 | 16.5% | 3.1% | 15.6% | 4.1% |
| 32 | 674881 / 1094017 | 10.9% | 3.0% | 9.8% | 4.6% |
| 64 | 691329 / 1135105 | 10.8% | 2.2% | 11.1% | 3.9% |

Table 13: Training and testing error obtained by training DON and SDON for the linear advection problem with different number of basis functions.

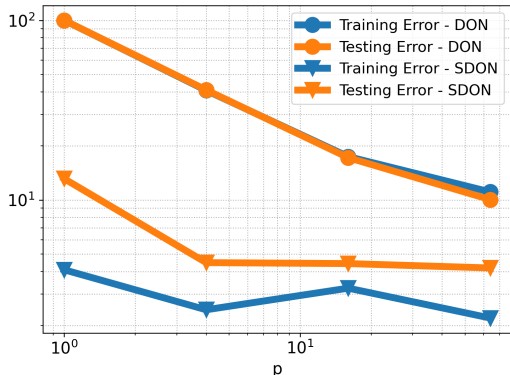

Figure 8: Testing error obtained by training DON and SDON with different number of basis function $p$ for the linear advection problem.

outperforms shift-DeepONet handily on each experiment, except the four-quadrant Riemann problem associated with the Euler equations of gas dynamics.

The results presented in Table 9 further demonstrate that FNO is the best performing model on all the benchmarks. In order to further understand the superior performance of FNOs, we consider the linear advection equation. As stated in the main text, given the linear nature of the underlying operator, a single FNO Fourier layer suffices to represent this operator. However, the layer width will grow linearly with decreasing error, requiring $k_{\max} \sim \epsilon^{-1}$ Fourier modes. Hence, it is imperative to use the nonlinear reconstruction, as seen in the proof of Theorem 3.3, to obtain good performance. To empirically illustrate this, we compare FNO with different $k_{\max}$ (number of Fourier modes) with corresponding error obtained by simply projecting the outputs of the operator into the linear space spanned by the corresponding number of Fourier modes. This projection onto Fourier space amounts to the action of a linear version of FNO. The errors, presented in Table 11 (keeping the total number of degrees of freedom roughly constant) and Table 12 (keeping the lifting dimension $d_v = 64$ fixed), clearly show that as predicted by the theory, very few Fourier modes with a $k_{\max} = 1$ are enough to obtain an error of approximately $1\%$. On the other hand, the corresponding error with just the linear projection is *two orders of magnitude* higher. In fact, one needs to project onto approximately 500 Fourier modes to obtain an error of approximately $1\%$ with this linear method. This stark contrast is further illustrated in Figure 9, which shows that FNOs relying on a *nonlinear* reconstruction mechanism can outperform a competing method based on linear Fourier reconstruction (Fourier linear projection) by one to two orders of magnitude in terms of approximation accuracy. This experiment clearly brings out the role of the nonlinearities in FNO in enhancing its expressive power on advection-dominated problems.

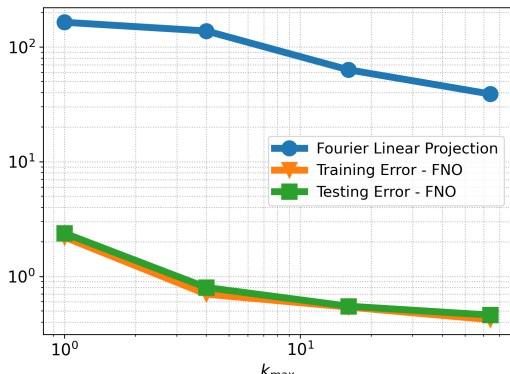

Figure 9: Testing error obtained by training FNO with different number of modes for the linear advection problem and corresponding error obtained with linear Fourier projection.

Besides our analysis of FNOs, we have shown by means of approximation theoretical arguments that shift-DeepONets, which are based on a nonlinear reconstruction, can overcome the fundamental limitations faced by DeepONets. Clearly, the approximation theoretic arguments employed in this work neglect some aspects of the practical training of neural operators, most notably possible errors due to a finite number of training samples and errors due to numerical optimization procedure (e.g. convergence to sub-optimal local minima, etc.). In order to further understand the superior performance of shift-DeepONets over DeepONets, we carry out a scan in the number of basis functions $p$.

For the linear advection problem, we first recall that the fundamental limitation of **DeepONets** is due to the lower bound of Theorem 3.1, which shows that DeepONets can achieve *at-best* a linear decay error $\sim 1/p$ in terms of the number of DeepONet basis functions $p$. To compare this theoretical prediction with empirical observation, we carry out a scan over different choices of $p = 1, 2, 4, \ldots, 64$ in Table 13. The table clearly shows that the DeepONet training/testing errors both decrease with increasing $p$. Furthermore, inspection of the corresponding Figure 8 reveals that this decrease follows an approximately linear decay, which is only slightly worse than the (optimal) lower bound $\sim 1/p$, consistent with approximation theory.

In contrast, our approximation theoretic result shows that **shift-DeepONets** can overcome the lower bound, and achieve high accuracy even with *a bounded number* of basis functions $p \leq C$ (cp. Theorem 3.2). This is reflected by the *almost immediate saturation* of the error as a function of $p$, in Figure 8 (cf. also Table 13). Consistent with the theoretical insight, our numerical experiment demonstrates that (a) the error for shift-DeepONets (based on nonlinear reconstruction) is considerably smaller than the corresponding error for DeepONets (based on linear reconstruction), and (b) increasing the number of basis functions beyond very moderate values of $p \sim 4, 8$ does not improve the accuracy of shift-DeepONets, consistent with the fact that a uniformly bounded number $p \leq C$ is sufficient.

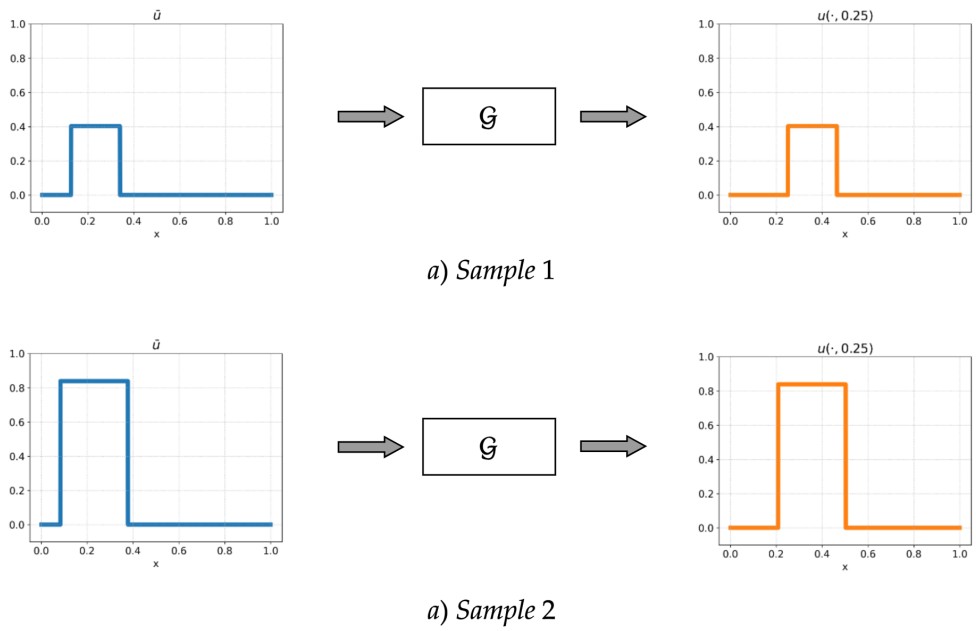

Figure 10: Illustration of two input (blue) and output (orange) samples for the advection equation.

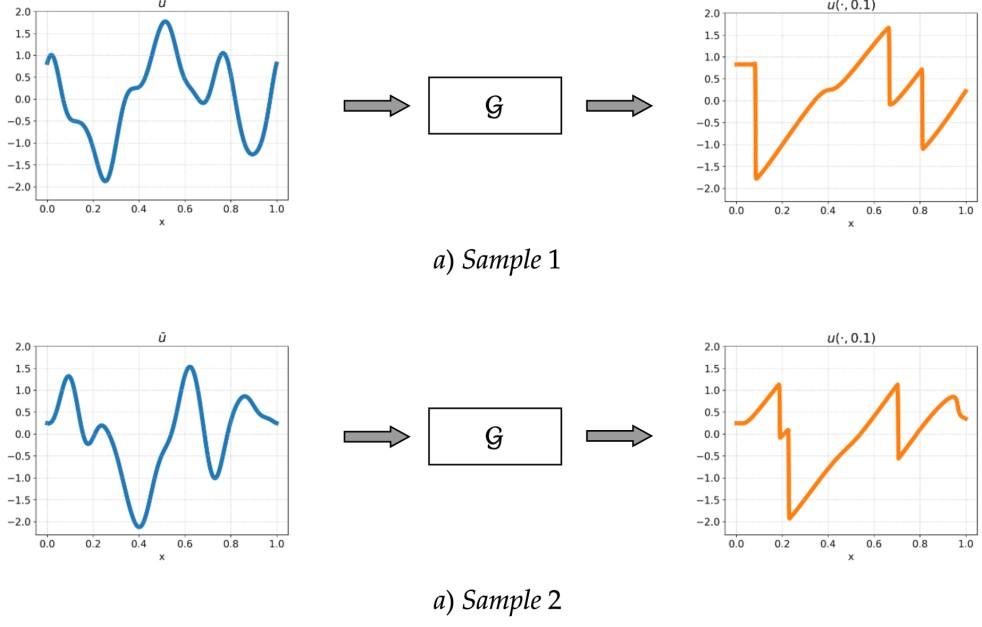

Figure 11: Illustration of two input (blue) and output (orange) samples for the Burgers' equation.

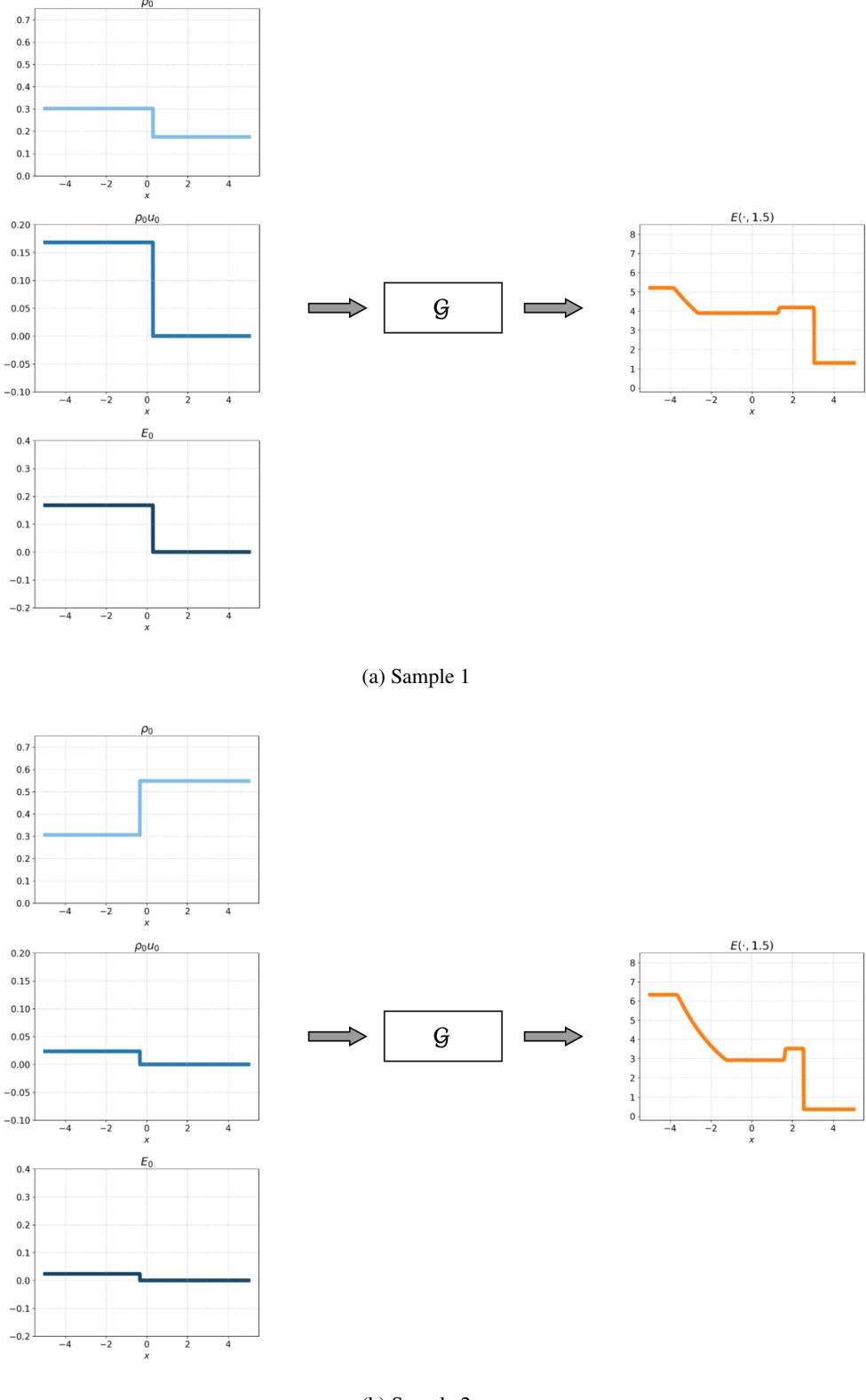

(a) Sample 1

(b) Sample 2

Figure 12: Illustration of two input (blue) and output (orange) samples for the shock-tube problem.

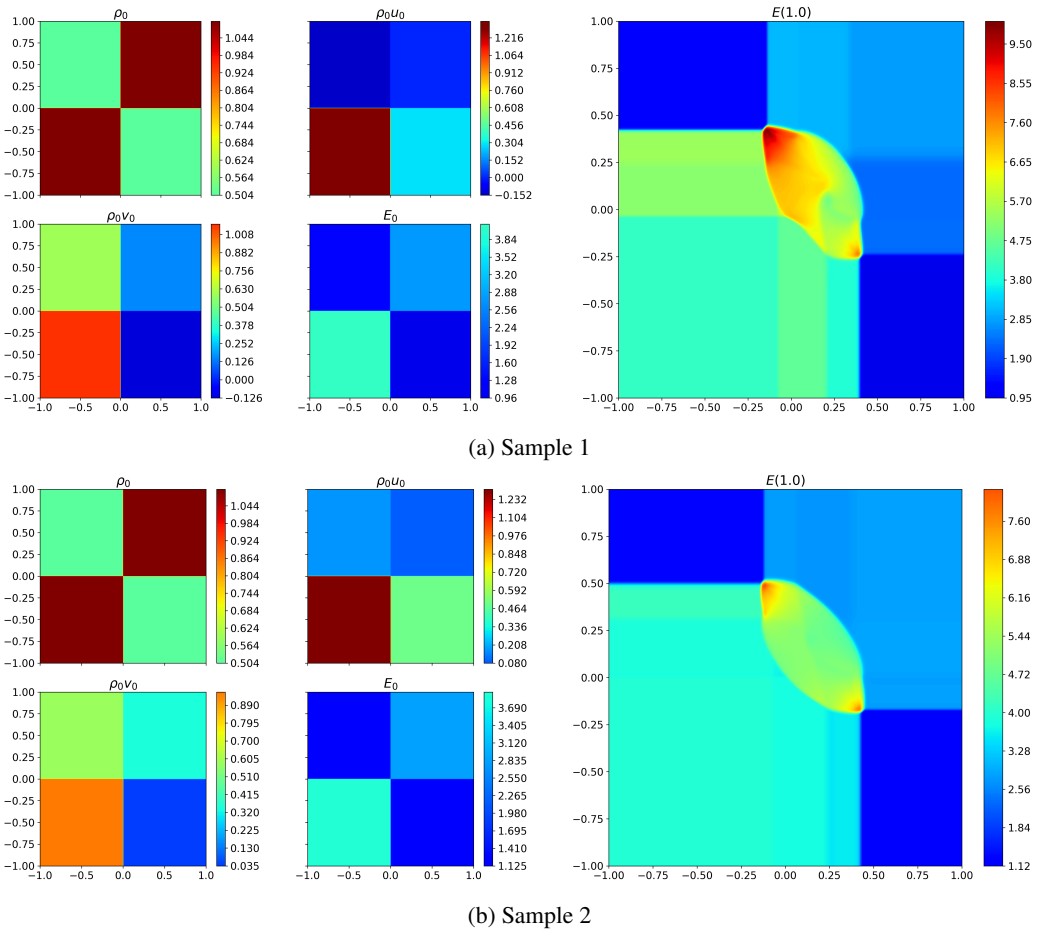

Figure 13: Illustration of two input (left) and output (right) samples for the 2-dimensional Riemann problem.

