# OpenReview forum: "Nonlinear Reconstruction for Operator Learning of PDEs with Discontinuities"
_ICLR.cc/2023/Conference — ICLR 2023 notable top 25%_

### Official Review · Reviewer_vjUn · 2022-10-23

**Confidence:** 5
**Correctness:** 4
**Technical Novelty And Significance:** 4
**Empirical Novelty And Significance:** 4
**Recommendation:** 8

**Clarity, Quality, Novelty And Reproducibility:**

Quality
Good: The paper appears to be technically sound. The proofs, if applicable, appear to be correct, but I have not carefully checked the details. The experimental evaluation, if applicable, is adequate, and the results convincingly support the main claims.

Clarity
Good: The paper is well organized but the presentation has minor details that could be improved.

Novelty
Good: The paper makes non-trivial advances over the current state-of-the-art.

Reproducibility
Good: key resources (e.g., proofs, code, data) are available and sufficient details (e.g., proofs, experimental setup) are described such that an expert should be able to reproduce the main results.


**Strength And Weaknesses:**

Strengths:
1.This paper is the first theoretical paper investigating the operators learning of PDEs with discontinuities.
2.The authors provide quantitative theoretical results and rigorous proof.
3.The paper is well-organized and well-written.
4.Extensive and supportive numerical experiments are provided.

Weakness:
Numerical examples do not illustrate the quantitative result, for example, the linear decay of model size of DeepONet, exponential decay rate of FNO and Shift-DeepONet.


**Summary Of The Paper:**

This paper provides rigorous analysis of operator learning of PDEs with discontinuities. The main contributions are two-fold. First, author prove that frameworks using linear reconstruction (DeepONet or PCA-Net) fail to efficiently capture the discontinuities by proving the lower bound of the approximation error decay merely linearly in terms of size of DeepONet. Second, the authors prove that two nonlinear reconstruction architectures, Shift-DeepONet and FNO, outperform the linear reconstruction architectures in the sense that much smaller model size is required. And the theoretical results are firmly supported by extensive numerical experiments.

**Summary Of The Review:**

This paper is the first paper rigorously investigate the error estimate of operator learning of PDEs with discontinuities. The authors carefully illustrate why linear reconstruction approaches fail to efficiently handle discontinuities, and how nonlinear reconstruction fix this theoretically and numerically. Additional numerical experiments that support the quantitative result (linear rate in DeepONet and exponential rate in FNO and Shift-DeepONet) are expected.

---

> ### Author Response · Authors · 2022-11-18
> **Response to reviewer vjUn - Part II**
>
> [...]
>
> 3. Finally, we consider Theorem 3.3 on the approximation error for FNO. A careful look at the statement of the theorem reveals that the key piece in the complexity bound is provided by the fact that very few Fourier modes i.e., $k_{max}=1$ suffice to obtain small errors. Given that the underlying operator is linear, as a comparison, we can switch off the nonlinear activations in FNO and it reduces to a linear Fourier projection. As the underlying solution is discontinuous, number of modes for this Fourier projection has to increase linearly in the error $\epsilon$ i.e., $k_{max} \sim \frac{1}{\epsilon}$. This provides the contrast to nonlinear FNO where, following Theorem 3.3, $k_{max}=1$ suffices to achieve error $\epsilon$. Thus, the theory suggests that the approximation error with FNO will be very small with $k_{max}=1$ and could saturate if the number of Fourier modes is increased. In contrast, the error with linear Fourier projection will only decay linearly with the number of Fourier modes. We test these theoretical predictions numerically and present the results in **SM** Figure 9. As seen from this figure and predicted by the theory, the error with linear Fourier projection decays linearly with number of modes. On the other hand, as predicted by the theory, both the training and test errors with FNO are very small with very few Fourier modes $k_{max}=1$ and saturate and do not decrease much with further increase in Fourier modes. Thus, only few Fourier modes are needed to obtain low errors with FNO. This fact is also reiterated in **SM** Table 11 where training and test errors, for different number of Fourier modes, are presented. Moreover, we observe that it is not possible to drive training errors (and consequently test errors) below $\approx 0.5\\%$, on account of training saturation due to local minima. Thus, we claim that **SM** Figure 9 and Table 11 illustrate Theorem 3.3 well.
>
> Given these additional numerical experiments, we can illustrate the predictions of our theorems with empirical evidence and thank the reviewer for pointing out this avenue for further improvement of our article. Moreover, we sincerely hope that we have addressed the concerns of the reviewer to your satisfaction.

---

> ### Author Response · Authors · 2022-11-18
> **Response to reviewer vjUn - Part I**
>
> We start by thanking the reviewer for your appreciation of the merits of our paper and your welcome suggestions to improve it. Below, we address the concerns raised by the reviewer and thank the reviewer in advance for their patience in reading our detailed reply.
>
> 1. We agree with the reviewer's comment that further numerical experimentation might be needed to illustrate the theoretical results in Section 3, in particular with respect to the complexity of the underlying operator learning frameworks. Following the reviewer's excellent suggestions in this regard, we have provided additional experimental evidence to support the theoretical conclusions. To this end, we start with the observation that the theorems in section 3 provide upper bounds for the approximation error of shift-DeepONets and FNOs. Other sources of error such as optimization error (due to convergence to local rather than global minima) and generalization error (due to finite number of training samples) are also present and will affect the training and test errors. With this caveat in mind, we focused on the linear advection equation with square wave initial data and attempted to further illustrate the theoretical results in Theorems 3.1, 3.2 and 3.3.
> In this context, we first consider Theorem 3.1 on the approximation error of DeepONets. A careful reading reveals that the key ingredient in the error lower bound will be provided by the fact that the error only decays linearly in the number of basis functions (branch and trunk nets) $p$ i.e., $\epsilon \sim \frac{1}{p}$. This prediction is tested numerically for the linear advection equation and the resulting training and test errors are presented in **SM** Figure 8. We observe from this figure that the DeepONet errors indeed decay linearly in the number of basis functions as predicted by the theory.
>
> 2. Next, we consider the approximation errors due to shift-DeepOnets, which are estimated in Theorem 3.2. We observe from this Theorem that the key difference in approximation between DeepONet and shift-DeepONet is provided by the fact that in contrast to DeepOnets, a constant number of basis functions ($p \leq C$, with $C$ independent of error $\epsilon$) suffices to obtain a desired approximation error.  We test this prediction numerically and present the resulting training and test errors for shift-DeepONet in **SM** Figure 8. We see from this figure that the error drops rapidly and a very small number of basis functions ($p \sim 4$) suffice in yielding low errors. After this and as predicted by the theory, the error does not decrease with increasing the number of basis functions. Thus and in contrast to DeepONets, a small number of basis functions is enough for low errors with shift-DeepOnets. Consequently, this figure illustrates the predictions of Theorem 3.2 very well.

---

### Official Review · Reviewer_koS7 · 2022-10-23

**Confidence:** 4
**Correctness:** 2
**Technical Novelty And Significance:** 2
**Empirical Novelty And Significance:** 1
**Recommendation:** 3

**Clarity, Quality, Novelty And Reproducibility:**

The problem looks very new, but there is not a very presentation. It looks like the problem formulation has not been presented clearly.  The dynamic process also has not been shown in the paper.  Whether it makes sense for the theoretical bound provided in this paper is unknown.

**Strength And Weaknesses:**

Strength: The topic of learning the solution of PDE, especially the discontinuous solution, is very interesting.

Weaknesses: (1) The results in all theorems, from Theorem 3.1 to Theorem 3.3,  look to learn the solution to the PDE at some time T. They
                            do not tell the readers what kind of numerical method to get G_adv = u(\cdot, T).

                       (2) The result is about the solution at some T. Does the approximation depend on the time T?

                       (3)  How does the dynamic approximation estimation evolve with time?


**Summary Of The Paper:**

This paper tries to investigate the operator learning of PDEs with discontinuous solutions. They prove that the linear method cannot approximate efficiently the solution operator. Furthermore, they propose a nonlinear reconstruction mechanism to overcome these fundamental difficulties.   And they propose new algorithms such as Fourier Neural Operators and a novel extension of DeepONet termed shift-DeepONet. Finally, they provide a rigorous theory for the empirical results.

**Summary Of The Review:**

Using the method of deep learning to solve PDE is a very interesting topic.   The basic numerical method for the nonlinear PDE is
 also unknown. Moreover, the motivation, the dynamic process and any kind of approximation still need to be presented clearly.

---

> ### Author Response · Authors · 2022-11-18
> **Response to reviewer koS7 - Part II**
>
> [...]
>
> 4. In response to the reviewer's claim that the problem formulation has not been presented clearly, we beg to differ. We have very precisely stated which operators we wish to learn. The rationale for learning these operator is quite evident as PDEs with discontinuous solutions are very prominent in applications, particularly in fluid dynamics. Moreover, there is no prior rigorous work on operator learning in this context. Hence, we provide the first theoretical results on an important topic in machine learning for science and engineering and also supplement these results with careful numerical experiments.
> 5. The reviewer has assessed that "several of the claims in the paper are either incorrect or not well-supported". We wish to reiterate that the paper consists of two parts. The first (and major) part is the theoretical section, which has theorems with precise statements and proofs. The reviewer does not point out any mistakes in them. The second part consists of numerical experiments. Again, all the test cases, datasets and models are clearly defined. All the hyperparameters are specified and training and architecture details are provided in the **SM** Section D. Finally, the code is provided to ensure reproducibility of the results. Again, the reviewer does not specify which of the results in the numerical section are incorrect or not well-supported. We would be happy to address these concerns on correctness if the reviewer could be more specific.
> 6. Similarly, the reviewer has assessed that "novelty of the paper is either marginal or non-existent". We politely disagree with this characterization. As stated earlier, this is the first article that analyzes operator learning of PDEs with discontinuities, proves theorems about the approximation and provides numerical evidence to support the theoretical conclusions. This aspect was also appreciated by other reviewers. Finally, in their evaluation of the strengths of this article, the reviewer has acknowledged that this is a very interesting topic to explore. Therefore, we request the reviewer to reconsider their assessment on the novelty of our article.
>
> We sincerely hope that we have addressed all your concerns, particularly about the dependence of our results on the time variable, on the approximation of the time history (dynamic evolution) and on the novelty of the article. Hence, we kindly request the reviewer to update your assessment accordingly.

---

> ### Author Response · Authors · 2022-11-18
> **Response to reviewer koS7 - Part I**
>
> We start by thanking the reviewer for reading our paper and for your comments. Below, we address the concerns raised by the reviewer and thank the reviewer in advance for their patience in reading our detailed reply.
>
> 1. We start with the reviewer's comment that *we do not tell the reader what kind of numerical method to get $G_{adv}$.* In response, we would like to point out that $G_{adv}$ is the solution operator for the linear advection equation which has been very clearly defined on Page 4, in the sentence just below Eqn. (3.1). As it is the solution operator, its definition does not require the specification of any numerical method. Similarly, all the operators that are studied in the paper are precisely defined, for instance $G_{burg}$ is defined on Page 6, in the sentence below Eqn. (3.5) and the operators for the one-dimensional and two-dimensional versions of the compressible Euler equations are defined on page 8. These operators are also illustrated in **SM** Figures 10-13.
> Thus, all the operators studied here are defined based only on the underlying PDE and input measure and independent of any numerical methods. However to generate training and test data for operator learning, one has to use a numerical approximation of the underlying PDE when analytical solution formulas are not available. Again, we very clearly state what numerical methods are used to generate the training and test data. In particular, these data sets for all the nonlinear equations (Burgers', 1-D Euler and 2-D Euler) are generated with the open source ALSVINN code of Lye 2020, which is based on a high-resolution finite-volume scheme. The precise resolutions of the numerical approximation is also provided for each example on pages 7 and 8. Finally, we also provide the code, so that our results can be reproduced.
> 2. The reviewer's question on whether our derived bounds depend on the final time $T$ is very pertinent. We are happy to report that the bounds derived in Theorems 3.1 to 3.6 are in fact *independent of the final time T* and holds for any time T. This fact is now clearly mentioned in the corresponding theorems.
> 3. We interpret the reviewer's comment on *dynamic estimation* as whether the operator learning frameworks considered here (shift-DeepONets, FNOs) can also approximate the time history (trajectory) of the solution and not just the solution at any given arbitrary time. In response, we can point out that as our error bounds do not depend on the underlying time, we can indeed readily extend the operator learning frameworks to learn the time evolution of the system. This can be done in a sequential manner that has now been described in **SM** Section C.1. for the linear advection equation. Bounds, analogous to those in Theorems 3.1 to 3.3 can be derived for this case and are mentioned therein. We thank the reviewer for outlining this avenue for further extension of our results.

---

> ### Comment · Area_Chair_3s1b · 2022-12-04
> **Please provide feedback.**
>
> Dear reviewer koS7  you have the lowest score among all of the reviewers. Can you check the feedback of the authors, and also other reviews?

---

### Official Review · Reviewer_ZZzU · 2022-10-24

**Confidence:** 3
**Correctness:** 4
**Technical Novelty And Significance:** 4
**Empirical Novelty And Significance:** 4
**Recommendation:** 8

**Clarity, Quality, Novelty And Reproducibility:**

Clarity: The paper is clearly written and follows a clear structure

Quality: The paper presents a high quality mathematical analysis and uses a good experimental evaluation setup

Novelty: The authors present a novel and original method and analysis for an interesting and important problem in operator learning

Reproducibility: The results of the paper should be generally reproducibility with the information provided in the paper and the extensive supplemental material

**Strength And Weaknesses:**

Strength:
- Clear structure and motivation, original contribution of the shift idea to the DeepONet operator learning approach
- High quality mathematical analysis of the given example problems
- A comprehensive set of example PDEs / initial value instances for the experiments section

Weakness:
- Intuition of push forward covariance eigenvalue distribution does not become clear to someone who is not already familiar with what that looks like. Some explanation of what eigenvalues would look like
for a simple example of a solution operator for a smooth solution vs discontinuous solution would be helpful.
- In the experiments sections it's not entirely clear why only 128 test examples were used to compute relative errors (that would still allow for relatively high error variance). A training set of 1024 example also seems quite small. Also adding standard defiation bars around the errors would be helpful.

Other remarks:
- Typo: "In particular, this bound continuous to hold" -> should be "continues" instead?
- Typo: "described in detailed" -> "described in detail"
- Typo: "Riemman" -> Riemann
- I'm not sure if e.g. Theorem 3.2 should be a theorem or just a proposition as you're considering an example initial data measure and perhaps the generality is not as great as to warrant calling it a theorem. There are perhaps possibilities for achieving more generality by treating general piecewise constant solutions of scalar conservation laws (although I'm not sure having more generality is necessary for making the point that the paper makes).


**Summary Of The Paper:**

The authors consider the problem of learning the solution operator of PDEs with discontinuous solutions (such as hyperbolic conservation laws) from initial data/solution pairs with initial data draw from a given measure.
The authors consider the DeepONet approach for operator representation that uses a sort-of finite sum over basis functions leveraging a "branch-net" for learning the coefficients and a "trunk-net" for learning the "basis-functions".
Based on an existing lower bound on the expected error of DeepONet given the eigenvalues of the covariance of the push-forward measure obtained from the solution operator the authors motivate a variation of DeepONet called shift-DeepONet that through a shift-net enables learning shift relations between initial data and solution that are typical for hyperbolic conservation laws where shock discontinuities in the solution can shift as a result of perturbing the initial data.
The authors also consider Fourier neural operators (FNOs) as an alternative for better learning solution operators for discontinuous solutions because these also do not suffer from a similar dependence on the covariance spectrum of the push-forward.
In the theory section of the paper the authors prove that for a simple linear transport with initial data measure representing box-type initial data DeepONet size has to scale quadratically as error decays while their shift-DeepONet size only has to scale linearly and an FNO only has to scale logarithmically.
Furthermore, for a simple nonlinear scalar conservation law (Burgers' equations) with sinus initial data and solution after shock-formation the authors prove that DeepONet has to scale linearly, while shift DeepONet and FNO only have to scale logarithmically.
In the experiments section the authors again demostrate that both shift-DeepOnet and FNO have better relative approximation error than DeepONet and some other alternatives for a linear advection, a Burgers, an 1D Euler (shock tube) and a 2D Euler (Rieman) problem.

**Summary Of The Review:**

I recommend to accept the paper and would ask the authors to try to make it more accessible by providing a better intuition on the issue of the eigenspectrum decay for the covariance of the push-forward with discontinuous solutions. I like the combination of proposing the new shift-parameterization of the model which seems well motivated, providing a theoretical analysis and a comprehensive set of experiments. I'm not sure whether the ICLR audience is the right audience for this sort of problem but the fourier neural operator is an ICLR paper so it could be okay. I also feel like although the theoretical analysis is high quality and I like the insight it provides it is not all that general since it still only pertains a set of examples and does treat any more general class of operators.

---

> ### Author Response · Authors · 2022-11-18
> **Response to reviewer ZZzU - Part II**
>
> [...]
>
> 3. All the minor points raised by the reviewer have now been addressed in the revised version.
> 4. The reviewer is correct in pointing out that the theoretical analysis, presented in this paper, is provided for specific examples and might lack generality. In response, we start by saying that this is the first paper which addresses the issue of learning operators for PDEs with discontinuities rigorously. Given that the traditional numerical analysis of PDEs with discontinuous solutions is hard, it is quite standard to focus on model problems such as the linear advection equation and one-dimensional nonlinear Burgers' equation (see for instance Hesthaven 2018 and references therein). In fact, there is no satisfactory rigorous analysis of traditional numerical methods such as finite volume and finite element methods for systems of conservation laws such as the Euler equations, even in one space dimension. Thus, it would be very difficult to analyze machine learning algorithms for such problems. Given this context, we consider model problems here, where precise theoretical statements can be made. Within this constraint, we have obtained rather sharp bounds on the operator learning models. We can further generalize our results to examples of multi-dimensional linear advection (see **SM** Section C.2) and for approximating the entire time-trajectory of the solution (**SM** Section C.1). Extension to more general operators for scalar conservation laws should be possible with technical assumptions on the operator. However, we feel that this is well beyond the scope of this paper.
>
> We sincerely hope to have addressed your concerns, particularly about the intuition behind the spectral properties of the push-forward measure, and would kindly request the reviewer to update your assessment accordingly.

---

> > ### Comment · Reviewer_ZZzU · 2022-11-18
> > **Thanks for the response**
> >
> > Thank you for addressing my review so carefully in your revision. I have raised my recommendation to "8: accept, good paper" accordingly.

---

> > > ### Author Response · Authors · 2022-11-18
> > > **Thanking the Reviewer**
> > >
> > > We thank the reviewer for appreciating our revised version and rebuttal and for raising our score.

---

> ### Author Response · Authors · 2022-11-18
> **Response to reviewer ZZzU - Part I**
>
> We start by thanking the reviewer for your appreciation of the merits of our paper and your welcome suggestions to improve it. Below, we address the concerns raised by the reviewer and thank the reviewer in advance for their patience in reading our detailed reply.
>
> 1. The reviewer's comment about the possible lack of intuition of the readers on the distribution of eigenvalues of the covariance operator for the push-forward measure is very well-taken. Following your excellent suggestion, we have now added a new section in the **SM**, section A.1, where we illustrate this spectral distribution for a particular example. The operator that we consider stems from the linear advection equation where the initial data is transported to a final time $T=1$. We consider different random initial data, where the underlying measures differ in terms of the smoothness (sharpness) of the gradients of functions in their support. Some samples are illustrated in **SM** Figure 1 (a). We proceed to explicitly compute the eigenvalues of the covariance operator in terms of a representation formula (**SM** Eqn. (A.2)). This formula is numerically approximated to calculate the eigenvalue distribution which is presented in **SM** Figure 1(b). As seen from this figure, one observes how these eigenvalues decay with different levels of smoothness of the underlying functions. In particular, the slow decay of discontinuous functions is clearly seen and this underlies the slow convergence of DeepONets for this example. We hope that this new section will aid the reader in understanding the spectral distribution of the covariance operator associated with the push-forward measure and thank the reviewer for asking us to do so.
> 2. In response to the reviewer's pertinent question on the size of the test set being limited to 128, we would like to point that we generated a total of $1280$ samples from numerical simulations for each of the 4 test problems and split this set into training ($80\\%$), validation ($10\\%$) and test ($10\\%$) sets. Such splits are very common in the machine learning literature. We chose to have uniform number of training and test samples across all the benchmarks n order to better highlight the expressivity of the proposed models. As our test problems included a two-dimensional benchmark, a Riemann problem at a moderately fine resolution of $256^2$, generating more number of samples would have been very expensive. Moreover, in practice, generating anything larger than (approximately) $1000$ samples for problems of interest is infeasible. Given these considerations, we chose $128$ test samples for each learning task.
> However, it is very interesting to investigate how the size of the test set might influence the test errors. To this end, we follow the reviewer's suggestion and test all the models for the linear advection equation with increasing sizes of the test sets. In this particular problem, the exact solution can be readily evaluated to provide the necessary test data. We present the resulting test errors in a new table i.e., **SM** Table 10, where we report the median test errors for test set size that varies from $128$ to $2048$ samples. We see from this table, there is very little variation of the median error as the test set size is increased, particularly for shift-DeepONets and FNOs (and also FCNN). There is a slight increase in error for DeepONets and ResNets. This indicates to us that test set size of $128$ could also suffice for other experiments and we thank the reviewer for bringing this issue to our notice.
> Regarding the reviewer's question about error bars, we would like to point out that we had already provided some uncertainty measures for all the models in the **SM** Table 9 where the $0.25$ and $0.75$ quantiles of the test error were presented. We have also provided this information in newly added **SM** Table 10.

---

### Official Review · Reviewer_sk7n · 2022-10-24

**Confidence:** 3
**Correctness:** 3
**Technical Novelty And Significance:** 3
**Empirical Novelty And Significance:** 3
**Recommendation:** 8

**Clarity, Quality, Novelty And Reproducibility:**

There are a few places where some extra explanation could help the reader,

- Page 3, the authors mention the lower bound in equation 2.4 does not apply for shift DeepONet, why is that?
- The proof sketches provided after the theorems and lemmas are a bit hard to follow.
- For the lowerbound, i think the sketch pretty much follows the setup from Lanthanler et al.. While this fact is clearly mentioned in the appendix, it would be great that the authors could also point that out in the main paper.

**Strength And Weaknesses:**

The paper shows a useful results which delineates DeepONets are not as efficient as FNOs for approximating PDEs with discontinuous solution.
They introduce shift-DeepONet, a variant of deepOnet which performs as well as FNOs on pdes with discontinuous solutions.
The derive a lower bound (based similar proof techniques used in prior works) for the approximation error of DeepONet for advection equation and burgers equation (in one dimension) and show that FNO and shift DeepONet can upper bound the solution more efficiently. While I didn't go into the details of all the proofs, the overall sketch and techniques seem correct.

The authors back their theory with empirical results where they show that shift-DeepONet and FNO perform better than normal DeepOnet and resnets on PDEs like linear advection, burgers equation etc.

One small question i have is how would the upper bound change with increasing the dimension? A trivial way would be to treat each dimension independently, but I wonder if the authors have any other (more efficient way) that they have in mind?

**Summary Of The Paper:**

The main premise of the paper is that for hyperbolic conservation laws PDEs, operator learning frameworks like DeepOnets are inefficient as compared to counterparts like Neural Fourier Operators. This is because DeepOnets (and PCA-Nets) are linear reconstruction methodologies (i.e, the trunk and branch net outputs are linearly combined) which effects their representational capabilities, whereas FNO based methodology have a nonlinear reconstructions step.

The authors therefore extend the DeepOnet framework to a shift-deepOnet Framework with an extra shift and scale terms that are learned.

The proofs show that for one dimensional Linear Advection equation and inviscid Burgers equation, the size of the linear reconstruction methods scale at least quadratically in the reconstruction error whereas for the nonlinear reconstruction method size scales linearly only.

**Summary Of The Review:**

This paper theoretically and empirically shows that conditions under which DeepOnets are less efficient than methodologies like FNOs and also provides an extension of DeepONets that can make up for the difference.

---

> ### Author Response · Authors · 2022-11-16
> **Response to reviewer sk7n**
>
> We start by thanking the reviewer for your appreciation of the merits of our paper and your welcome suggestions to improve it. Below, we address the concerns raised by the reviewer and thank the reviewer in advance for their patience in reading our detailed reply.
>
> 1. Regarding the reviewer's question about how the bounds will change with dimension, as reviewer correctly intuits one option would be consider problems such as linear advection of a multi-dimensional box wave. In this case, our bounds can be readily extended dimension by dimension to obtain analogous complexity bounds for DeepONet, shift-DeepOnet and FNO. We have added a Section i.e., **SM**, Section C.2, where we provide the details of these results. We thank the reviewer for their excellent suggestion which allows to generalize our theoretical results further.
> 2. Regarding the reviewer's question on why the lower bound (2.4) does not hold for shift-DeepONet, we would like to answer that the linearity of the reconstruction procedure was essential in obtaining this lower bound. Hence, to overcome this lower bound, one needs to make the basis (trunk net) $\tau$ *depend on the input $u$*, making the basis adapt to the input in a non-linear way. This input dependence inhibits the derivation of the lower bound. As shift-DeepONets precisely build in this input dependence, the lower bound (2.4) will not hold for this model. We have now added a sentence to the first paragraph on Page 3 to explicitly point this out and thank the reviewer again for asking this pertinent question.
> 3. Regarding the reviewer's comment on the proof of the lower bound on DeepOnets, Theorem 3.1, indeed the proof follows some parts of earlier work of Lanthaler et al 2022. The proof in this work is based on two observations. The first, sketched out in Lanthaler et al 2022 provides a lower bound in the terms of the number of basis functions $p$. However, one also needs a lower bound in terms of the number of sensors $m$ and this is not considered in Lanthaler et al. 2022. We provide the corresponding argument here. These remarks are detailed in a paragraph (Page 5, first para) that immediately follows the statement of Theorem 3.1.
>
> We sincerely hope to have addressed the minor concerns to your satisfaction and thank the reviewer again for pointing them out to us.

---

> > ### Comment · Reviewer_sk7n · 2022-11-23
> > **Reply to the Rebuttal**
> >
> > I would like to thank the authors for their rebuttal. They have answered all my concerns (and even added the result for higher dimensions, which I really appreciate). Thank you for the insight on the lower bound!

---

> > > ### Author Response · Authors · 2022-12-04
> > > **Thanking the Reviewer**
> > >
> > > We thank the reviewer for appreciating the additions in the revised version and the clarifications in the rebuttal.

---

### Author Response · Authors · 2022-11-16
**General**

At the outset, we would like to thank all four reviewers for their thorough and patient reading of our article. Their fair criticism and constructive suggestions have enabled us to improve the quality of our article. A revised version of the article is uploaded. We proceed to answer the points raised by each of the reviewers individually, below.

We would also like to point out that all the references to page numbers, sections, figures, tables, equation numbers and references, refer to those in the revised version.

Yours sincerely'

Authors of "Nonlinear Reconstruction for Operator Learning of PDEs with Discontinuities."

---

### Public Comment · ~Shuhao_Cao1 · 2023-02-13
**A few questions and comments**

Wow, awesome work on getting the DeepONet working on the inviscid Burgers' with shocks and Euler's.

A few questions as a newcomer in this field:
- What type of finite volume schemes is used in generating data for the Burgers' and transport equation? I checked the reference Lye (2020) Section 6 on the implementation of ALSVINN, the description is rather brief. It would be appreciated if more details can be added? Judging by the nice discontinuous non-oscillatory figure of shocks in Fig. 11, something fancier than Lax–Friedrichs must be used.
- Particularly, what types of flux and/or jump conditions are used? There are quite a few entropy conditions for Burgers', by "entropy condition" I assume the jump condition in the FVM follows suit. So which one is used in the data generation?
- In the proof of Theorem 3.2 (transport eq) as well as Lemma D.11, what does it mean that "$at$ is a fixed constant" in the context of the solution represented by methods of characteristics?
- In the proofs of Theorem 3.3, does $k_{\mathrm{max}} = 0$ means one Fourier mode being retained, while $k_{\mathrm{max}} = 1$ means two? Moreover, if this is the case, should the DFT complexity be modified to big-O of $((k_{\mathrm{max}}+1)N)^d$?
- Appendix E.1.4: it says $m$ (number of sensors) is chosen according to the validation, so in Table 6 and Table 7 the values of $m$ are the optimal ones? Does shift-DeepONet's number of parameters depend on $m$?
- Will the code be publicly released on GitHub?


Comments:
- I think "linear-reconstruction based" should actually be "linear reconstruction-based".
- Some places use "nonlinear" while others use "non-linear".
- In several occasions, "a FNO" -> "an FNO".
- Earlier in describing the inviscid Burgers' eq, it says "$2\pi$-periodic domain $D=\mathbb{T}$", which I assume to be the spatiotemporal torus $\simeq S^1\times S^1$; later in the experiment section, it says $D=[0,1]$; in the Appendix it says $\mathbb{T} = [0, 2\pi]$ which I think is inaccurate.
- page 9: missing reference on "attention based framework", the first paper on attention and PDE operator learning is Cao, S. (2021). Choose a transformer: Fourier or Galerkin. *NeurIPS 2021*.
- page 9: "a operator learning manifold hypothesis".
- Lemma D.8: $\Xi_{\epsilon}$ should be from $\mathbb{R}\times \mathbb{R} \to \mathbb{R}$.
- Proposition D.10: the notation $\beta_k(\bar{u}(x_1),\dots, \bar{u}(x_m))$ uses a discretized vector as input, however, earlier it says $\beta_k: \mathcal{X}\to \mathbb{R}$.

---

> ### Author Response · Authors · 2023-02-14
> **Reply**
>
> Hi Shuhao,
>
> Thank you for your careful reading of the paper! Let me at least try to give you partial answers (I was mostly focused on the theory part):
>
> * I can't personally speak to the details of the implementation in ALSVINN -- but it definitely uses something more clever than the basic LxF; if you wanted to run it yourself, the code is open-source, and can be found on github: https://github.com/alsvinn/alsvinn
> * In the proof for the transport equation, the transport velocity $a\ne 0$ is assumed to be a fixed parameter of the underlying problem. We also fix the final evaluation time $t>0$, so their product $at$ is a constant that is independent of the input function.
> * Yes, $k_{max}=0$ means that only the zero-th Fourier mode is retained, while $k_{max}=1$ means that all Fourier modes with multi-indices $\max_{j=1,\dots, d} |k_j|\le 1$ are retained. In 1D, this corresponds to Fourier modes $k=-1,0,1$.
> * The code can be found in the supplementary material .zip file; don't know whether it will also be released on github.
> * By the one-dimensional torus $\mathbb{T}$, we just mean a domain that is topologically equivalent to a circle, i.e. $[0,2\pi]$ with endpoints identified (=periodic boundary conditions). Time is not assumed periodic. The spatio-temporal domain of interest is $[0,T]\times \mathbb T$ (periodic BC in space only). I guess the experiments rescaled the domain to $[0,1]$ and still with periodic boundary conditions -- to me that's only a cosmetic issue.

---

### Decision · Program_Chairs · 2023-01-20

**Decision:**

Accept: notable-top-25%

**Justification For Why Not Higher Score:**

I don't think this is an oral paper, since the result is novel, but there are many papers in this field and direction, and the current paper has an interesting result but for a particular type of equations. Also, it does not propose a significantly new or better architecture/method for learning the solution.

**Justification For Why Not Lower Score:**

There are no concerns that the results are fresh and important.

**Metareview: Summary, Strengths And Weaknesses:**

The paper highlights the problems of using DeepONets for the approximation of the solution of PDEs with discontinuous solutions.
Recent results show that the error of DeepONets can be bounded from below, which poses significant barrier for using such kind of models. This paper provides a first theoretical analysis of different methods for PDEs with discontinuous solutions.

Strengths:

The main strengths of the paper are very interesting theoretical results. As noted by the reviewers, In the theory section of the paper the authors prove that for a simple linear transport with initial data measure representing box-type initial data DeepONet size has to scale quadratically as error decays while their shift-DeepONet size only has to scale linearly and an FNO only has to scale logarithmically.
Furthermore, for a simple nonlinear scalar conservation law (Burgers' equations) with sinus initial data and solution after shock-formation the authors prove that DeepONet has to scale linearly, while shift DeepONet and FNO only have to scale logarithmically.

Weaknesses:
Experimental design and presentation of the result have been questioned (i.e. the number of train/test samples, meaning of the covariance matrix eigenvalues), but they have addressed. One of the reviewers had concerns about the weaknesses and "theoretical justification" but it looks just wrong to me, so it can not be considered as a weakness.

Overall, this is an interesting an important paper.

**Note From Pc:**

if the above contains the word "oral" or "spotlight" please see: "oral" presentation means -> notable-top-5% and "spotlight" means -> notable-top-25%. As stated in our emails, we are disassociating presentation type from AC recommendations